**TOOLS**

# Live imaging of transcription sites using an elongating RNA polymerase II–specific probe

Satoshi Uchino[1]* ⓘ, Yuma Ito[1]* ⓘ, Yuko Sato[1,2] ⓘ, Tetsuya Handa[2] ⓘ, Yasuyuki Ohkawa[3] ⓘ, Makio Tokunaga[1], and Hiroshi Kimura[1,2] ⓘ

In eukaryotic nuclei, most genes are transcribed by RNA polymerase II (RNAP2), whose regulation is a key to understanding the genome and cell function. RNAP2 has a long heptapeptide repeat (Tyr1-Ser2-Pro3-Thr4-Ser5-Pro6-Ser7), and Ser2 is phosphorylated on an elongation form. To detect RNAP2 Ser2 phosphorylation (RNAP2 Ser2ph) in living cells, we developed a genetically encoded modification-specific intracellular antibody (mintbody) probe. The RNAP2 Ser2ph-mintbody exhibited numerous foci, possibly representing transcription "factories," and foci were diminished during mitosis and in a Ser2 kinase inhibitor. An in vitro binding assay using phosphopeptides confirmed the mintbody's specificity. RNAP2 Ser2ph-mintbody foci were colocalized with proteins associated with elongating RNAP2 compared with factors involved in the initiation. These results support the view that mintbody localization represents the sites of RNAP2 Ser2ph in living cells. RNAP2 Ser2ph-mintbody foci showed constrained diffusional motion like chromatin, but they were more mobile than DNA replication domains and p300-enriched foci, suggesting that the elongating RNAP2 complexes are separated from more confined chromatin domains.

## Introduction

In eukaryotic nuclei, RNA polymerase II (RNAP2) transcribes most genes, and the regulation of RNAP2-mediated transcription is a key to determining the cell phenotype (Cramer, 2019; Reines, 2020; Vos, 2021). Human RNAP2 is a complex of 12 subunits, and the catalytic largest subunit (RPB1) contains 52 heptapeptide repeats (typically Tyr1-Ser2-Pro3-Thr4-Ser5-Pro6-Ser7, with variations) at the C-terminal domain (CTD). The site-specific Ser phosphorylation at the CTD is associated with the state of RNAP2 (Eick and Geyer, 2013; Schüller et al., 2016; Zaborowska et al., 2016; Harlen and Churchman, 2017). Un-phosphorylated forms of RNAP2 are assembled into preinitiation complexes at the gene promoters, and during the initiation of transcription, Ser5 at the CTD becomes phosphorylated by the cyclin-dependent kinase 7 (CDK7) in the transcription factor IIH complex. RNAP2 Ser5 phosphorylation (Ser5ph) is enriched at the transcription start sites (Komarnitsky et al., 2000) and remains during elongation to facilitate cotranscriptional splicing in mammalian cells (Nojima et al., 2018). After synthesizing a short stretch of RNA, RNAP2 is paused by negative elongation factor and 5,6-dichloro-1-β-ᴅ-ribofuranosylbenzimidazole sensitivity-inducing factor (Yamaguchi et al., 1999; Cramer, 2019; Reines, 2020). RNAP2 then undergoes elongation in association with Ser2 phosphorylation (Ser2ph) by positive

transcription elongation factor b (P-TEFb), which is a protein complex of CDK9 and cyclin T (Price, 2000). During elongation, Ser2ph is likely to be maintained by other CDKs, such as CDK12 and CDK13, each complexed with cyclin K, to facilitate transcription processivity (Fan et al., 2020; Tellier et al., 2020). RNAP2 Ser2ph can mediate the interaction with proteins involved in RNA processing, such as splicing and polyadenylation, as well as epigenetic regulation (Li et al., 2002; Hsin and Manley, 2012; Gu et al., 2013; Venkat Ramani et al., 2021). RNAP2 becomes dephosphorylated when transcription is terminated. On short genes, such as small nuclear RNAs and histones, Ser2 is not phosphorylated (Medlin et al., 2005). Transcription elongation is regulated by many proteins, including the polymerase-associated factor 1 (Paf1) complex that was initially identified as a protein complex interacting with RNAP2 in yeast, the depletion of which causes aberrant polyadenylation in mammalian cells (Hou et al., 2019; Francette et al., 2021).

The localization of RNAP2 and its phosphorylated forms in the cell nucleus has been analyzed by light and electron microscopy. RNAP2 Ser2ph is closely associated with nascent RNA labeled with bromo-UTP, which is consistent with this modification being a mark of elongating form (Iborra et al., 1996; Pombo et al., 1999). From the number of RNAP2 transcription

......................................................................................................................................................................

[1]School of Life Science and Technology, Tokyo Institute of Technology, Yokohama, Japan;   [2]Cell Biology Center, Institute of Innovative Research, Tokyo Institute of Technology, Yokohama, Japan;   [3]Division of Transcriptomics, Medical Institute of Bioregulation, Kyushu University, Fukuoka, Japan.

*S. Uchino and Y. Ito contributed equally to this paper;   Correspondence to Hiroshi Kimura: hkimura@bio.titech.ac.jp;   T. Handa's present address is Cancer Research UK Cambridge Institute, University of Cambridge, Cambridge, UK.

sites (~8,000) and actively transcribing molecules (~65,000) in a HeLa cell, it was estimated that one transcription site with ~80-nm diameter contains eight RNAP2 molecules, on average, to form transcription "factories" (Jackson et al., 1998; Cook, 1999). In other nontransformed cell types, a smaller number of factories with a larger diameter have been observed (Osborne et al., 2004; Eskiw and Fraser, 2011). In mouse fetal liver erythroblasts, for example, the average diameter of transcription sites is ~130 nm (Eskiw and Fraser, 2011). The presence of such "factories" can increase the local concentration of the factors involved in transcription, and genes regulated under the same set of transcription factors often share the same factories (Osborne et al., 2004; Xu and Cook, 2008; Schoenfelder et al., 2010; Eskiw and Fraser, 2011; Papantonis et al., 2012).

The dynamic behavior of RNAP2 in living cells has been analyzed using fluorescent protein (FP)–tagged RNAP2 molecules. As >100,000 RNAP2 molecules are present in a cell (Kimura et al., 1999; Stasevich et al., 2014), it has been difficult to resolve the small foci by standard confocal microscopy (Sugaya et al., 2000; Imada et al., 2021), but recent confocal and 3D stimulated emission depletion microscopy has enabled detection of single elongating RNAP2 foci in living cells (Li et al., 2019). On a transcriptionally activated gene array harboring >100 copies of transcription units, the accumulation of FP-tagged RNAP2 can be readily detected (Becker et al., 2002; Janicki et al., 2004). Fluorescence recovery after photobleaching (FRAP) revealed the kinetics of different RNAP2 fractions and the elongation periods (Becker et al., 2002; Kimura et al., 2002; Darzacq et al., 2007; Steurer et al., 2018). High-resolution single-molecule analyses using photoconvertible or photoactivatable RNAP2 have indicated the transient clustering of RNAP2 during initiation (Cisse et al., 2013; Cho et al., 2016; Boehning et al., 2018) in association with mediator condensates (Cho et al., 2018). CTD phosphorylation can promote a condensate preference switch from the mediator to splicing factor condensates (Guo et al., 2019). The dynamics of elongating RNAP2 have not been well documented, except RNAP2 foci on some specific genes, including *Nanog* in mouse embryonic stem cells (Li et al., 2019).

To detect the specifically modified forms of RNAP2 and histones in living cells, fluorescently labeled antigen-binding fragments have been used (Hayashi-Takanaka et al., 2011; Stasevich et al., 2014). Using a cell line that harbors a reporter gene array containing glucocorticoid response elements and that expresses glucocorticoid receptor tagged with GFP, the accumulation kinetics of RNAP2 Ser5ph and Ser2ph on the array upon glucocorticoid stimulation have been revealed (Stasevich et al., 2014). In this system, the target gene array was identifiable by glucocorticoid receptor–GFP accumulation in a large focus, but Ser5ph and Ser2ph on other single copy genes were not detected. Recently, more detailed kinetic analysis of RNAP2 Ser5 phosphorylation on a single copy gene has been performed, which has also enabled the visualization of the spatiotemporal organization of RNAP2 phosphorylation and mRNA synthesis (Forero-Quintero et al., 2021).

To visualize and track RNAP2 Ser2ph conveniently without protein loading, we developed a genetically encoded live-cell probe, which is a modification-specific intracellular antibody, or mintbody, that consists of the single-chain variable fragment (scFv) of the specific antibody and superfolder GFP (sfGFP; Sato et al., 2013, 2016; Tjalsma et al., 2021). Using high-resolution microscopy, we analyzed the relative localization of RNAP2 Ser2ph-mintbody with proteins involved in RNAP2 phosphorylation, elongation, and transcription activation, such as CDK9, CDK12, a Paf1 complex component LEO1, serine/arginine-rich splicing factor 1 (SRSF1; also known as SF2/ASF), bromodomain containing protein 4 (BRD4; Dey et al., 2003), and p300 histone acetyltransferase (Heintzman et al., 2009; Visel et al., 2009). RNAP2 Ser2ph showed more colocalization with CDK12 and LEO1 than CDK9, which facilitates elongation near the transcription start site (Price, 2000), and enhancer-associated proteins that can also facilitate elongation, such as BRD4 and p300 (Zhang et al., 2012; Li et al., 2019; Hsu et al., 2021). These proteins were more mobile in RNAP2 Ser2ph-enriched regions than elsewhere. RNAP2 Ser2ph foci as such were also more mobile than both euchromatic and heterochromatic DNA replication foci and p300-enriched foci. These results suggest that elongating RNAP2 foci are quite mobile compared with typical euchromatin and heterochromatin domains.

## Results

### Generation of the RNAP2 Ser2ph-specific mintbody

To generate a mintbody specific for RNAP2 Ser2ph (Fig. 1 A), we cloned the coding sequence of antibody heavy and light chains from 18 different hybridoma cells that produce Ser2ph-specific antibodies and transfected scFvs tagged with sfGFP into HeLa cells. Confocal microscopy revealed that a clone, 42B3, was most concentrated in the nucleus among the scFvs, and the others showed weak or little nuclear enrichment (Fig. S1 A). As the scFv-sfGFP does not harbor an NLS, this suggested that 42B3 scFv was functionally folded and bound to RNAP2 Ser2ph in the nucleus, whereas the other scFvs had a much lower antigen-binding affinity or failed to properly fold in the cytoplasm (Wörn and Plückthun, 2001; Sato et al., 2013, 2016; Zhao et al., 2019). In contrast to the sfGFP-tagged version, 42B3 scFv tagged with mCherry (42B3-mCherry) expressed in HeLa cells often exhibited cytoplasmic aggregations, and its nuclear enrichment was not as high as that of 42B3-sfGFP (Fig. 1 B, top; and Fig. S1 A). Because the folding of scFv is affected by the fusion partner protein or short peptide (Kabayama et al., 2020), 42B3 scFv folding can be assisted by sfGFP and/or be disturbed by mCherry, which could cause cytoplasmic aggregations of fusion proteins (Landgraf et al., 2012). Thus, 42B3 scFv did not appear to have a particularly stable framework, unlike another mintbody specific for histone H4 Lys20 monomethylation (H4K20me1), which is functional even when fused with mCherry and whose framework has been used to generate a stable chimeric scFv by implanting the complementary determining regions from a different antibody (Sato et al., 2016; Zhao et al., 2019; Liu et al., 2021).

To improve the folding and/or stability of 42B3 scFv, we introduced amino acid substitutions to the framework region by comparing with the H4K20me1-mintbody (Sato et al., 2016; Fig. 1

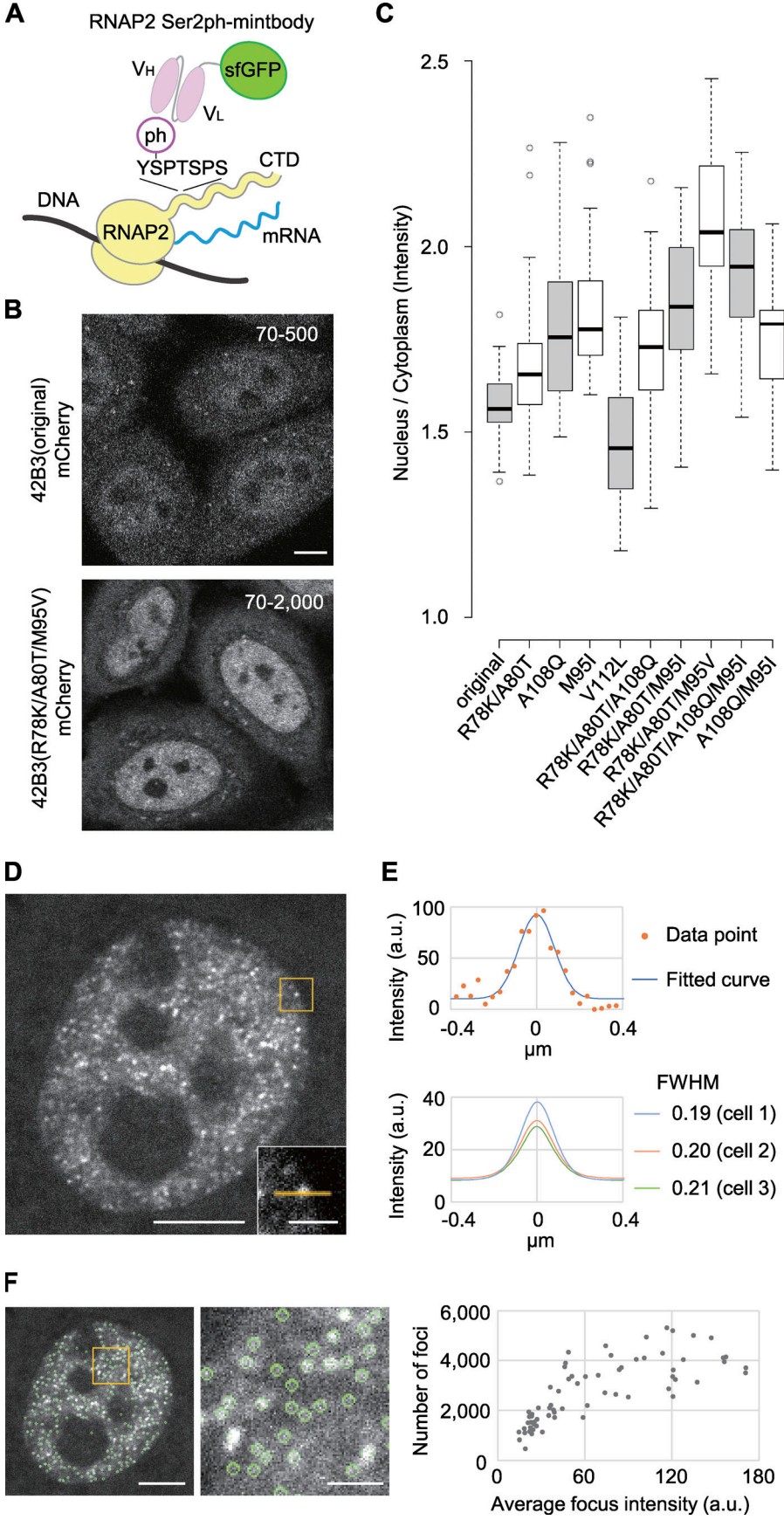

Figure 1. **Generation of the RNAP2 Ser2ph-specific mintbody and characterization of the mintbody foci in HeLa cells. (A)** Schematic diagram of the mintbody that binds to phosphorylated Ser2 on the CTD of RNAP2. **(B)** Single confocal sections of 42B3(original)-mCherry and 42B3(R78K/A80T/M95V)-mCherry, which were stably expressed in HeLa cells. Fluorescence images were acquired using a confocal microscope under the same settings, and image contrast was enhanced differently [70–500 for the original 42B3-mCherry and 70–2,000 for 42B3(R78K/A80T/M95V)-mCherry] because the intensity of the former was dimmer. **(C)** N/C ratios of 42B3 and mutants tagged with mCherry. HeLa cells expressing 42B3-mCherry and mutants were established. After confocal images were acquired, the N/C ratios of fluorescence were measured (*n* = 30). In the box plots, center lines show the medians; box limits indicate the 25th and 75th percentiles as determined by R software; whiskers extend 1.5 times the interquartile range from the 25th and 75th percentiles; data points are plotted as open circles. **(D–F)** RNAP2 Ser2ph-mintbody in living HeLa cells. **(D)** Using a high-resolution spinning-disk confocal system, the fluorescence image of RNAP2 Ser2ph-mintbody in a living HeLa cell was acquired. A single section of the confocal image is shown. Bottom right: An example of focus with a 1-μm line in the indicated area. **(E)** Estimating the size of the foci. Line intensity profiles of single fluorescence foci were plotted in a.u. Top: An example of the focus and its line intensity profile with the fitted curve. Top right: The intensity plot of the line and the fitted curve by the Gaussian distribution model. Bottom: Average fitted curves from 20 spots per cell are shown with an average full width at half maximum (FWHM) for three different cells. **(F)** Estimating the number of foci. Foci that were identified by a spot detection algorithm are indicated by green circles. A magnified view is shown in the middle. The total number of foci was counted by analyzing the z-sections covering the whole nucleus at 0.2-μm intervals. As a single focus can be detected in a maximum of three sections, the total counts were divided by 3. The numbers and average intensities (a.u.) of foci in 68 cells are plotted. Scale bars, 5 μm; 1 μm for magnified views.

C). Five amino acids in 42B3 were replaced with those in the H4K20me1-mintbody individually and in combinations (Fig. S1 B), and the mutant 42B3-mCherry proteins were stably expressed in HeLa cells. By evaluating the nuclear accumulation (i.e., the nucleus/cytoplasmic intensity [N/C] ratio) and the formation of cytoplasmic aggregations, we found that 42B3(R78K/A80T/M95I)-mCherry was the most improved, showing the highest N/C ratio with few cytoplasmic aggregates (Figs. 1 C and S1 C). Structural modeling showed that the R78K/A80T and M95I mutations likely contribute to intramolecular ionic and hydrophobic interactions, respectively (Fig. S1 D). As the hydrophobic cores of scFv are critical for folding and/or stability (Sato et al., 2016; Tjalsma et al., 2021), we examined another M95 mutant with Val substitution, and this 42B3(R78K/A80T/M95V)-mCherry exhibited the highest N/C ratio (Fig. 1, B and C). Therefore, we used this scFv for subsequent analyses. As described below, 42B3(R78K/A80T/M95V)-sfGFP specifically bound to RNAP2 Ser2ph, so we called this probe "RNAP2 Ser2ph-mintbody."

## RNAP2 Ser2ph-mintbody is concentrated in the foci in interphase nuclei in living HeLa cells

We established HeLa cells that stably express RNAP2 Ser2ph-mintbody and observed its distribution using a high-resolution spinning-disk confocal microscope (Fig. 1 D). RNAP2 Ser2ph-mintbody was concentrated in numerous foci over a diffuse background in the nucleus, suggesting that this probe highlights transcription elongation sites where RNAP2 Ser2ph is present. As one CTD harbors 52 repeats and more than half of Ser2 in the repeats can be phosphorylated (Schüller et al., 2016), a focus may be derived from a single RNAP2 molecule that is bound with multiple RNAP2 Ser2ph-mintbodies. However, it is also possible that a focus may contain multiple polymerases (Iborra et al., 1996). The size of the foci was estimated to be ∼200 nm (Fig. 1 E), which is the diffraction limit of confocal microscopy, suggesting that the actual focus size is <200 nm. The number of foci in a HeLa cell was estimated to be ∼500–5,000 (Fig. 1 F), depending on the mintbody brightness. The higher the mintbody fluorescence, the more foci were detected, because the signal-to-background ratio was low in weakly fluorescent cells. The number of foci, however, appeared to reach a plateau at ∼4,000–5,000, which is close to the number of factories in HeLa cells (Jackson et al., 1998; Pombo et al., 1999).

Most RNAP2 transcription is repressed during mitosis in mammalian cells by the release of the elongation complex (Prescott and Bender, 1962; Parsons and Spencer, 1997; Liang et al., 2015), although a subset of genes can remain to be transcribed (Palozola et al., 2017). By using HeLa cells that express both RNAP2 Ser2ph-mintbody and HaloTag-tagged histone H2B (H2B-Halo), which was labeled with HaloTag tetramethylrhodamine (TMR) ligand to visualize chromatin, we tracked the distribution of RNAP2 Ser2ph-mintbody during mitosis (Fig. 2; Videos 1 and 2). In cells that started chromosome condensation at the onset of prophase, RNAP2 Ser2ph-mintbody foci were observed around the edge or outside the condensing chromosomes (Fig. 2 A, 0 and 1 min; magnified views and line scan profiles), which is consistent with the results of a previous study using RNA in situ hybridization to detect nascent transcripts

(Liang et al., 2015). The number of foci decreased as the prophase progressed (Fig. 2 A, 2 and 3 min) and almost disappeared in the late prophase (Fig. 2 A, 4 min; Video 1). After the nuclear membrane broke down, the RNAP2 Ser2ph-mintbody diffused throughout the cytoplasm (Fig. 2 A, 5 min). During prometaphase to metaphase, the mintbody appeared to be largely excluded from condensed chromosomes (Fig. 2 B). During the telophase to $G_1$ phase after cytokinesis, RNAP2 Ser2ph-mintbody became concentrated in foci, and the number of foci gradually increased in early $G_1$ (Fig. 2 C; Video 2). These mitotic behaviors of RNAP2 Ser2ph-mintbody are consistent with previous observations using fixed cells (Parsons and Spencer, 1997; Liang et al., 2015).

## Mintbody foci depend on RNAP2 Ser2ph in living cells

To investigate whether RNAP2 Ser2ph-mintbody foci in nuclei depend on RNAP2 and its Ser2ph, cells were treated with triptolide, which inhibits transcription factor IIH ATPase activity (Titov et al., 2011) and also induces RNAP2 degradation (Wang et al., 2011; Forero-Quintero et al., 2021), and flavopiridol, which inhibits the RNAP2 Ser2 kinase P-TEFb (Chao and Price, 2001). As a control, we also treated cells with the vehicle DMSO. Time-lapse images were collected using a confocal microscope, and the numbers of the RNAP2 Ser2ph-mintbody foci in single sections were measured (Fig. 3, A and B). The number of RNAP2 Ser2ph-mintbody foci was not decreased in cells treated with DMSO for 2 h. In contrast, the inhibition of the transcription initiation and depletion of RNAP2 with 5 µM triptolide (Forero-Quintero et al., 2021) resulted in the rapid diminishment of nuclear foci. Flavopiridol also induced the diminishment of nuclear foci, but more mildly because only the productive elongation of newly initiated RNAP2 is inhibited, leaving elongating RNAP2 complexes until their termination (Jonkers et al., 2014); RNAP2 Ser2ph in elongating complexes has also been shown to remain in 5,6-dichloro-l-β-D-ribofuranosyl benzimidazole treatment, which also inhibits P-TEFb (Lavigne et al., 2017).

We also used FRAP to examine the kinetics of RNAP2 Ser2ph-mintbody in cells untreated or treated with triptolide and flavopiridol for 2–4 h (Fig. 3 C). After bleaching a 2-µm spot in the nucleoplasm, RNAP2 Ser2ph-mintbody signals recovered rapidly to >80% within 2 s in cells without inhibitors, indicating that this probe binds only transiently to Ser2ph, as do the other mintbodies and antigen-binding fragment fluorescent probes that have been used for tracking post-translational modifications in living cells without affecting cell division and animal development (Hayashi-Takanaka et al., 2009, 2011; Sato et al., 2013; Stasevich et al., 2014; Sato et al., 2016). In triptolide- and flavopiridol-treated cells, the recovery of RNAP2 Ser2ph-mintbody was faster, reaching >80% within 1 s after bleaching, which was consistent with the decreased level of the target RNAP2 Ser2ph. Thus, kinetic analysis supported the view that RNAP2 Ser2ph-mintbody repeatedly binds and unbinds to RNAP2 Ser2ph in living cells.

## Biochemical evaluation of RNAP2 Ser2ph-mintbody specificity

To biochemically characterize the specificity of RNAP2 Ser2ph-mintbody, we performed an in vitro ELISA using the purified

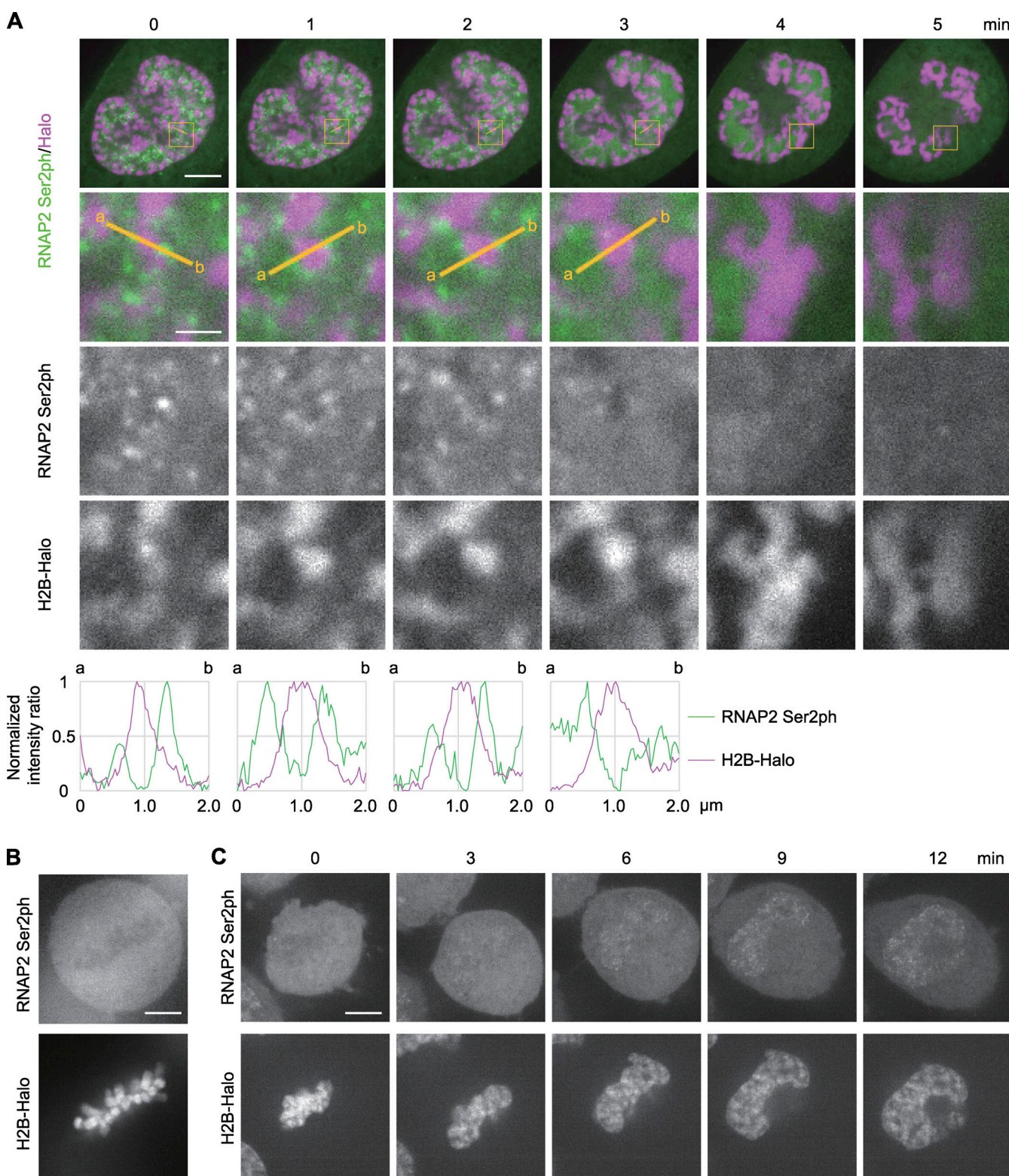

Figure 2. **RNAP2 Ser2ph-mintbody foci during the entry into and exit from mitosis in HeLa cells.** HeLa cells that expressed both RNAP2 Ser2ph-mintbody and H2B-Halo were established. Cells were stained with 50 nM of TMR-conjugated HaloTag ligand for 30 min before imaging using a high-resolution confocal microscope. **(A–C)** Single confocal sections of live-cell images during the prophase to prometaphase (A), metaphase (B), and telophase to $G_1$ (C) are shown. **(A and C)** The elapsed time (min) is indicated. **(A)** Merged images (top and second rows) and magnified views of indicated areas (the second to fourth rows) are shown. Line intensity profiles of 2-µm lines in the second row are shown (fifth row); the maximum and minimum intensities on the line are set to 1 and 0. See also Videos 1 and 2 for A and C, respectively. Scale bars, 5 µm; 1 µm for magnified views.

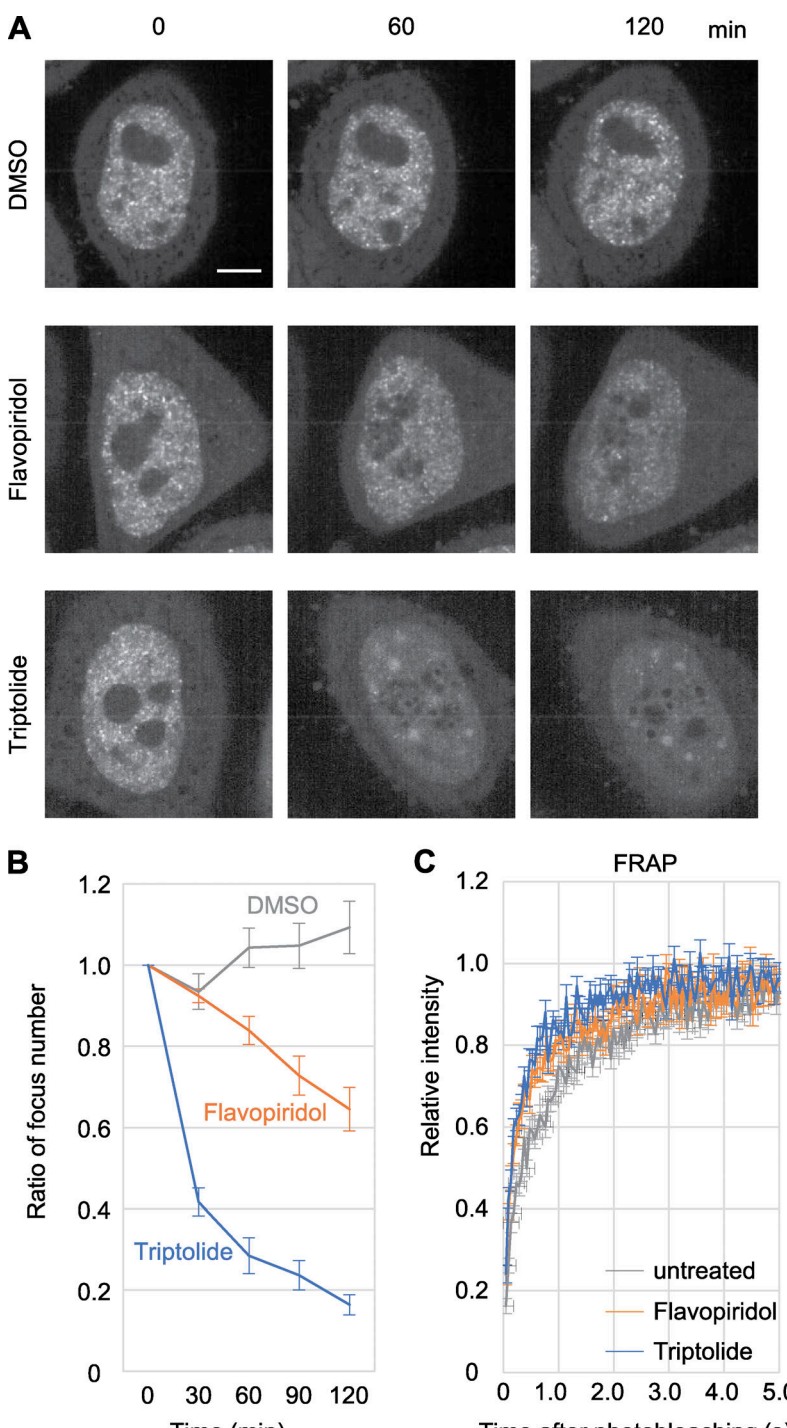

Figure 3. **Evaluating the specific binding of the RNAP2 Ser2ph-mintbody to RNAP2 Ser2ph in living cells. (A and B)** HeLa cells that stably express the RNAP2 Ser2ph-mintbody were treated with 1 µM flavopiridol, 5 µM triptolide, or vehicle (0.1% DMSO) for 2 h. High-resolution confocal images were acquired using a high-resolution confocal system. **(A)** Representative images at 0, 60, and 120 min. Scale bar, 5 µm. **(B)** Nuclear foci were selected using autothresholding, and the total area in the single nucleus was measured. The areas relative to time 0 were plotted with SEM ($n = 10$). The area of foci decreased by treatment with flavopiridol and triptolide. **(C)** FRAP. A 2-µm-diameter circle spot in a nucleus was bleached, and the fluorescence recovery was measured. RNAP2 Ser2ph-mintbody fluorescence recovered to 80% within 2 s in untreated cells, and the recovery speed increased (to <1 s) in cells treated with flavopiridol and triptolide for 2–4 h ($n = 35$).

protein. His-tagged RNAP2 Ser2ph-mintbody was expressed in *Escherichia coli* and purified using $Ni^{2+}$ beads. After removal of the His-tag by enterokinase, tag-free RNAP2 Ser2ph-mintbody was further purified (Fig. 4 A). ELISA plates were coated with RNAP2 CTD peptides that harbor phosphorylation at different Ser residues and incubated with RNAP2 Ser2ph-mintbody and control IgG antibodies, including the parental 42B3 and previously published Ser2ph (CMA602)- and Ser5ph (CMA603)-specific antibodies (Stasevich et al., 2014; Fig. 4 B). RNAP2 Ser2ph-mintbody bound to peptides harboring Ser2ph, but not those harboring only Ser5ph and Ser7ph, as did the parental 42B3 IgG and CMA602. Ser5ph-containing peptides were reacted with CMA603, assuring that the Ser5ph peptides that were not reacted with RNAP2 Ser2ph-mintbody were properly coated on the plates. These results indicate that RNAP2 Ser2ph-mintbody selectively recognizes Ser2ph at RNAP2 CTD without being particularly affected by Ser5ph and Ser7ph on the same repeat unit. Taken together with cell-based analyses, we concluded that RNAP2 Ser2ph-mintbody can be used as a probe to track RNAP2 Ser2ph in living cells.

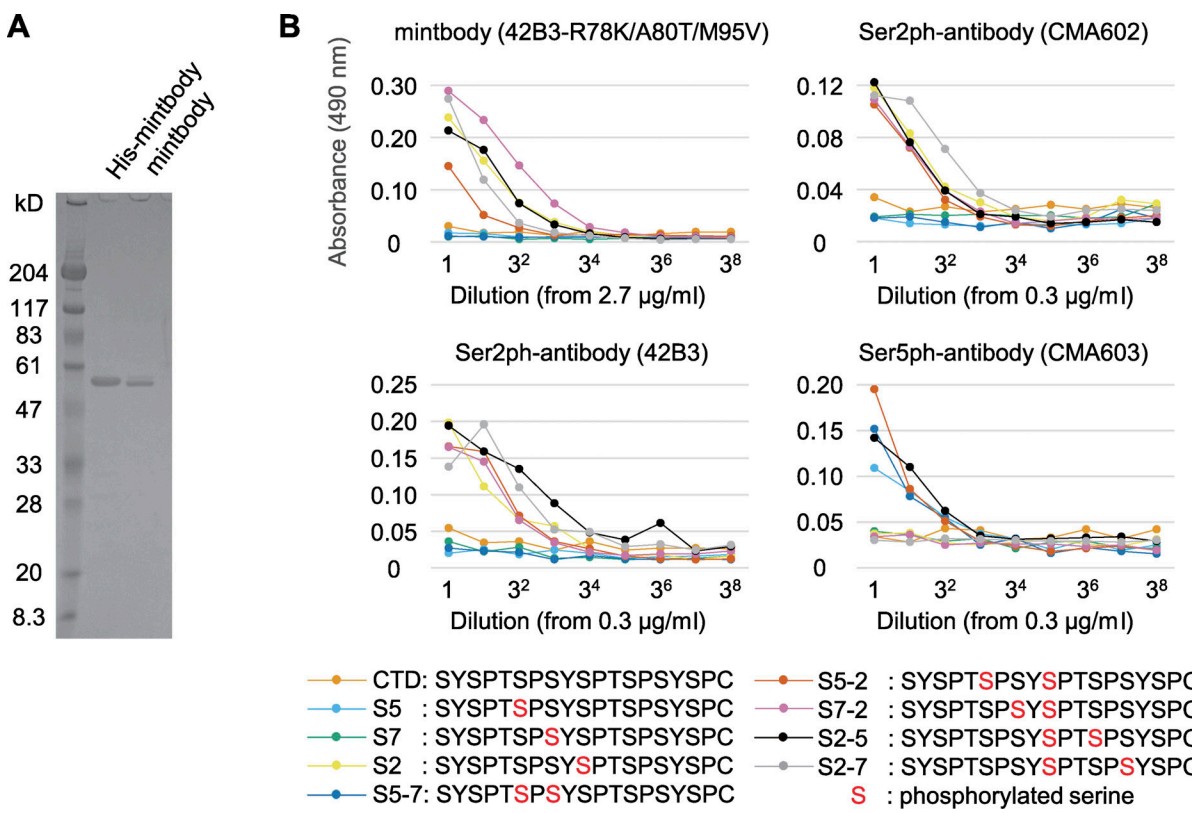

Figure 4. **Evaluating the specific binding of RNAP2 Ser2ph-mintbody to phosphopeptides in vitro.** RNAP2 Ser2ph-mintbody was expressed in *E. coli* as the His-tag form, purified through an Ni column, treated with enterokinase to remove His-tag, and further purified. **(A)** Purified proteins were separated on an SDS-polyacrylamide gel and stained with Coomassie Blue. Positions of size standards are indicated on the left. **(B)** ELISA plates that were coated with synthetic peptides conjugated with BSA, incubated with a dilution series of purified RNAP2 Ser2ph-mintbody, the parental 42B3 antibody (IgG), and control antibodies specific for RNAP2 Ser2ph (CMA602) and RNAP2 Ser5ph (CMA603). After incubations with peroxidase-conjugated anti-GFP (for RNAP2 Ser2ph-mintbody) or antimouse IgG (for mAb) and then with *o*-phenylenediamine, absorbance at 490 nm was measured. RNAP2 Ser2ph-mintbody reacted with peptides containing Ser2ph. Source data are available for this figure: SourceData F4.

## Localization of transcription-related factors with respect to RNAP2 Ser2ph foci

We compared the localization between RNAP2 Ser2ph-mintbody and proteins that are involved in transcription activation and elongation. An RNAP2 subunit, RPB3, histone H2B, and proliferating cell nuclear antigen (PCNA) were used as positive and negative controls. PCNA represents DNA replication foci, exhibiting various patterns depending on the cell cycle stages (Leonhardt et al., 2000; Fig. S2). HaloTag-tagged RPB3 (RPB3-Halo) labeled with TMR showed substantial overlap with RNAP2 Ser2ph-mintbody (Fig. 5, first column, merged image and line profile). To quantitatively evaluate the colocalization, we employed a cross-correlation function (CCF) analysis, which measures Pearson's correlations by shifting the image of one fluorescence channel to the x-direction (van Steensel et al., 1996; Fig. 5, bottom row). When foci in two fluorescence images overlap, a peak appears in the middle without shift. The steepness of the peak can also be an indication of a colocalized area. If the distribution of the two images is mutually exclusive, a dip in the middle is expected, and if the distribution is not correlated, either a peak or a dip would not be observed. In the case of RNAP2 Ser2ph-mintbody and RPB3-Halo, a narrow peak in CCF was observed (Fig. 5, first column, bottom), which was

consistent with the presence of Ser2ph on the RNAP2 complex that contains RPB3-Halo. The brightest foci of RPB3-Halo without RNAP2 Ser2ph-mintbody (Fig. 5, first column, top, arrowheads) probably represented large RNAP2 condensates, which are often associated with Cajal and histone locus bodies lacking Ser2ph (Xie and Pombo, 2006; Imada et al., 2021). By contrast, H2B-Halo exhibited a complementary pattern to RNAP2 Ser2ph-mintbody (Fig. 5, second column) showing a dip in CCF (Fig. 5, second column, bottom), which agrees with the decondensation of chromatin at transcribed regions. Halo-PCNA foci during the middle to late S phase when heterochromatin was replicated also showed anticorrelation with RNAP2 Ser2ph-mintbody (Fig. 5, third column). Thus, CCF analysis using live imaging data could indicate a relationship between a protein and RNAP2 Ser2ph-enriched transcription foci.

Several kinase complexes phosphorylate RNAP2 Ser2. CDK9 facilitates processive transcription by phosphorylating RNAP2 paused near the transcription start sites (Price, 2000), while CDK12 and CDK13 interact with RNAP2 during elongation (Fan et al., 2020; Tellier et al., 2020). RNAP2 Ser2ph-mintbody foci appeared to be only weakly associated with CDK9-Halo, with a small peak in CCF (Fig. 5, fourth column). A previous study has also shown infrequent overlapping between CDK9-mCherry and

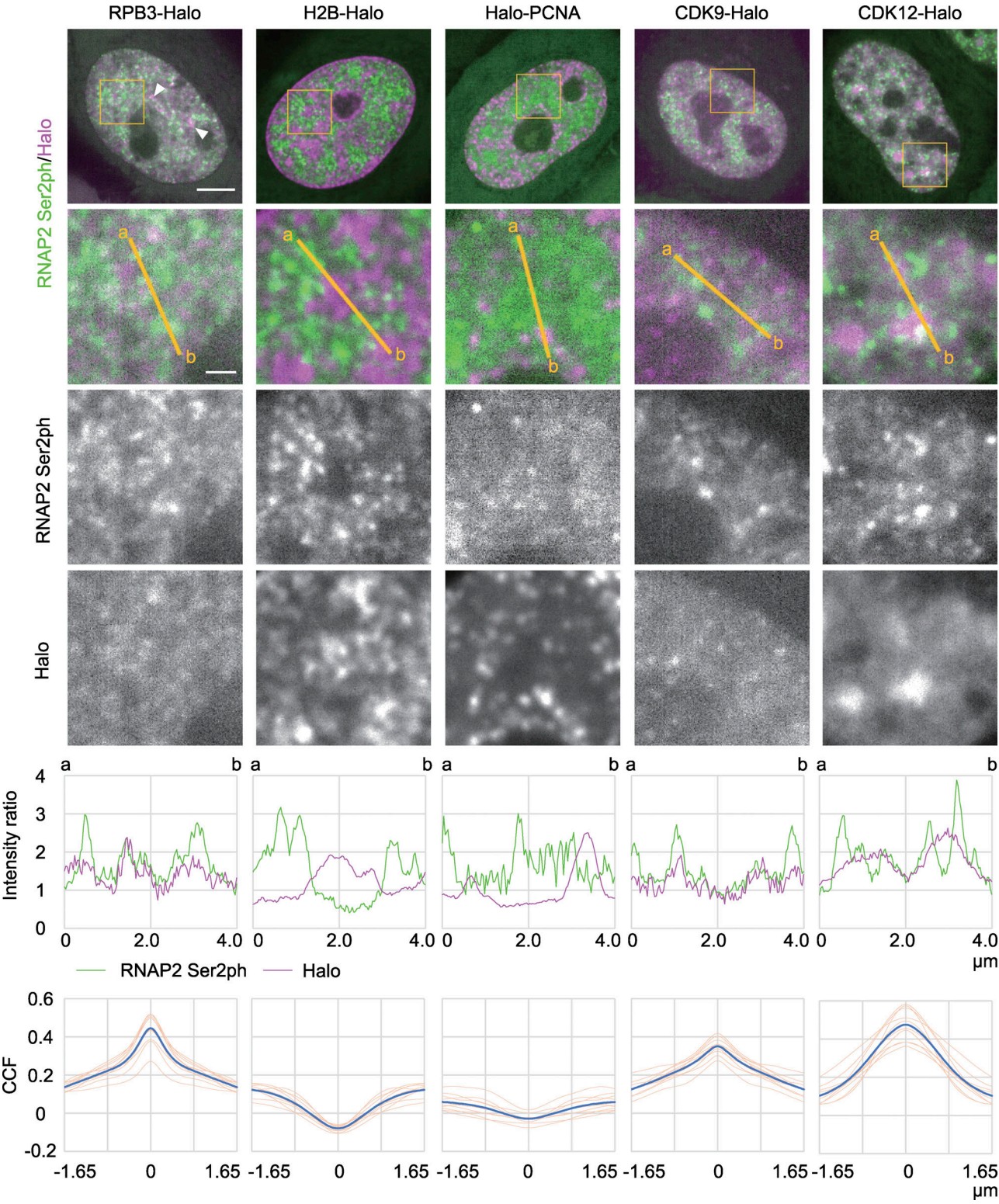

**Figure 5. Localization of RPB3, H2B, PCNA, CDK9, and CDK12 with respect to RNAP2 Ser2ph-mintbody.** HeLa cells stably expressing RNAP2 Ser2ph-mintbody were transfected with expression vectors for HaloTag-tagged proteins and stained with 50 nM HaloTag TMR ligand for 30 min before live-cell imaging using a high-resolution confocal system. The averaged images of 10 consecutive frames (551 ms/frame) are shown with magnified views of areas indicated in orange (middle three rows). Line intensity profiles of 4-μm lines in the second row are shown (fifth row) by normalizing against the average nuclear intensity to yield relative intensity ratios. The CCFs of 10 cells are shown at the bottom (blue, average; orange, individual cells). Scale bars, 5 μm; 1 μm for magnified views.

RNAP2 Ser2ph by immunofluorescence (Ghamari et al., 2013). RNAP2 Ser2ph-mintbody foci were rather associated with CDK12-Halo, although the distribution of CDK12-Halo was fuzzier than that of the mintbody, which resulted in a broad CCF peak (Fig. 5, fifth column). CDK12 was also concentrated in splicing factor speckle-like structures (Fig. 5, fifth column, top). A Paf1/RNAP2 complex component, LEO1, showed some overlapping with RNAP2 Ser2ph-mintbody with a CCF peak (Fig. 6, first column). These results suggested that CDK12 and the Paf1 complex are associated, but not always, with RNAP2 Ser2ph. A splicing factor, SRSF1, is recruited to the nascent RNA synthesis sites through the association with RNAP2 CTD phosphorylation (Misteli and Spector, 1999; Gu et al., 2013). RNAP2 Ser2ph-mintbody was often observed at the edge of SRSF1-enriched splicing speckles, which resulted in a broad peak in CCF (Fig. 6, second column).

Histone H3 acetylation is involved in transcription activation through chromatin decondensation and recruiting the co-activators and bromodomain proteins (Dhalluin et al., 1999; Tóth et al., 2004; Stasevich et al., 2014). We compared RNAP2 Ser2ph-mintbody localization with an acetyl reader protein, BRD4 (Fig. 6, third column), and an acetyltransferase, p300 (Fig. 6, fourth column; Fig. S3), which are associated with enhancers and promoters that harbor H3 Lys27 acetylation (Heintzman et al., 2009; Visel et al., 2009; Zhang et al., 2012; Kanno et al., 2014). Although the foci between RNAP2 Ser2ph-mintbody and BRD4 or p300 were sometimes closely associated, their overlapping was rare, and the correlations were low. Halo-p300 exhibited a number of small foci with a few bright condensates, as seen when monomeric enhanced GFP was knocked into the p300 locus (Ma et al., 2021). Although p300 could form larger condensates when highly expressed (McManus and Hendzel, 2003; Fig. S3 B), we only analyzed cells in which Halo-p300 was expressed at much lower levels than those with large condensates (Fig. S3 A). In these cells with low to moderate expression, CCF profiles of Halo-p300 with respect to RNAP Ser2ph-mintbody did not depend on the expression levels (Fig. S3 A). These observations for BRD4 and p300 were in good agreement with recent high-resolution analyses showing a separation of BRD4 and enhancers from nascent transcripts (Li et al., 2019, 2020).

### Mobility of proteins in RNAP2 Ser2ph-enriched regions

We next analyzed the mobility of single-protein molecules that were highly and lowly associated with RNAP2 Ser2ph-mintbody–enriched regions by labeling HaloTag-tagged proteins with a low concentration of the HaloTag TMR ligand using a highly inclined and laminated optical sheet (HILO) microscope (Tokunaga et al., 2008; Lim et al., 2018). As the individual RNAP2 Ser2ph-mintbody molecules bound to the target in less than a few seconds, we did not perform single-molecule analysis of the mintbody. Instead, we used the fluorescence intensity in the mintbody images to define RNAP2 Ser2ph-enriched regions using a much lower laser power than that used in the single-molecule analysis. Under the conditions used, individual RNAP2-mintbody foci were difficult to resolve. Thus, mintbody-enriched regions were defined by areas with locally high intensity after high-pass filtration (Fig. 7, A and B).

Single HaloTag-tagged proteins were simultaneously imaged with RNAP2 Ser2ph-mintbody fluorescence at 33.33 ms per frame for 16.67 s (Fig. 7, A and B). The diffusion coefficients ($D$) of the individual molecules were determined by linear fitting to the first six steps ($\sim$200 ms) of mean squared displacement (MSD; Fig. 7 C). The histograms of $D$ represent the bimodal distribution of slow and fast fractions (Fig. 7 D). As most H2B-Halo molecules were classified into slow fractions by setting a threshold of $D_{thr} = 0.065$ $\mu m^2/s$ (Fig. 7 D), this fraction represented the molecules bound to chromatin and/or other large structures. We called this the "bound fraction." Relatively large proportions of BRD4 and p300 were also found in the bound fraction compared with RPB3 and CDK9 (Figs. 7 D and S4 A). This probably reflects the small fraction of total cellular RNAP2 that is elongating at any given time (Kimura et al., 2002) and the transient interactions of CDK9 with RNAP2.

We then analyzed the mobility of bound molecules that were highly and lowly associated with RNAP2 Ser2ph-mintbody–enriched regions. As an example, shown in Fig. 7 C, the molecule tracked in (i) was located most (88%) of the time in an area of the nucleus that was enriched with RNAP2 Ser2ph-mintbody (Fig. 7 E, top). By contrast, the molecule tracked in (ii) was located exclusively in an area of the nucleus that was not enriched with RNAP2 Ser2ph-mintbody (Fig. 7 E, bottom). Molecules that went in and out of RNAP2 Ser2ph-mintbody–enriched regions were also observed (Fig. 7 F). After plotting $D$ and the underlying RNAP2 Ser2ph-mintbody intensity from each tracked molecule (Fig. 7 G, top; and Fig. S4 A), the bound fraction as determined above (Fig. 7 D) was subjected to further analyses (Fig. 7 G, bottom). Based on the RNAP2 Ser2ph-mintbody intensity, the top 25% and bottom 25% were classified as fractions that were highly and lowly associated with RNAP2 Ser2ph-mintbody–enriched regions (Fig. 7 G, bottom; and Fig. S4 A). Molecules in the top 25% and bottom 25% stayed in local RNAP2 Ser2ph-mintbody–enriched regions for, on average, 88% and 5% of the trajectory durations, respectively (Fig. 7 H). When we compared the $D$ values in the top (magenta) and bottom (green) fractions of bound molecules, all proteins except PCNA had higher $D$ values in the top 25% highly associated fraction with RNAP2 Ser2ph-mintbody–enriched regions compared with $D$ values in the bottom 25% fraction (Fig. 8 A). Consistent with the higher $D$ values, molecules in the top 25% showed larger areas of confinement and lower effective spring coefficients than the bottom 25% (Figs. 8 B and S4 B). Significant differences between median $D$ values of the top and bottom fractions were robustly observed over a wide range of definitions for the top and bottom fractions based on RNAP2 Ser2ph-mintbody intensity (from 5% to 45%) and also when different values of $D_{thr}$ were used to define the bound fraction (from 0.045 to 0.169 $\mu m^2/s$; Fig. S5). These data suggest that the bound molecules that are highly associated with RNAP2 Ser2ph-enriched regions are more mobile than lowly associated molecules.

### Mobility of RNAP2 Ser2ph foci

To further investigate the dynamics of the transcription elongation sites over several seconds, we analyzed the mobility of

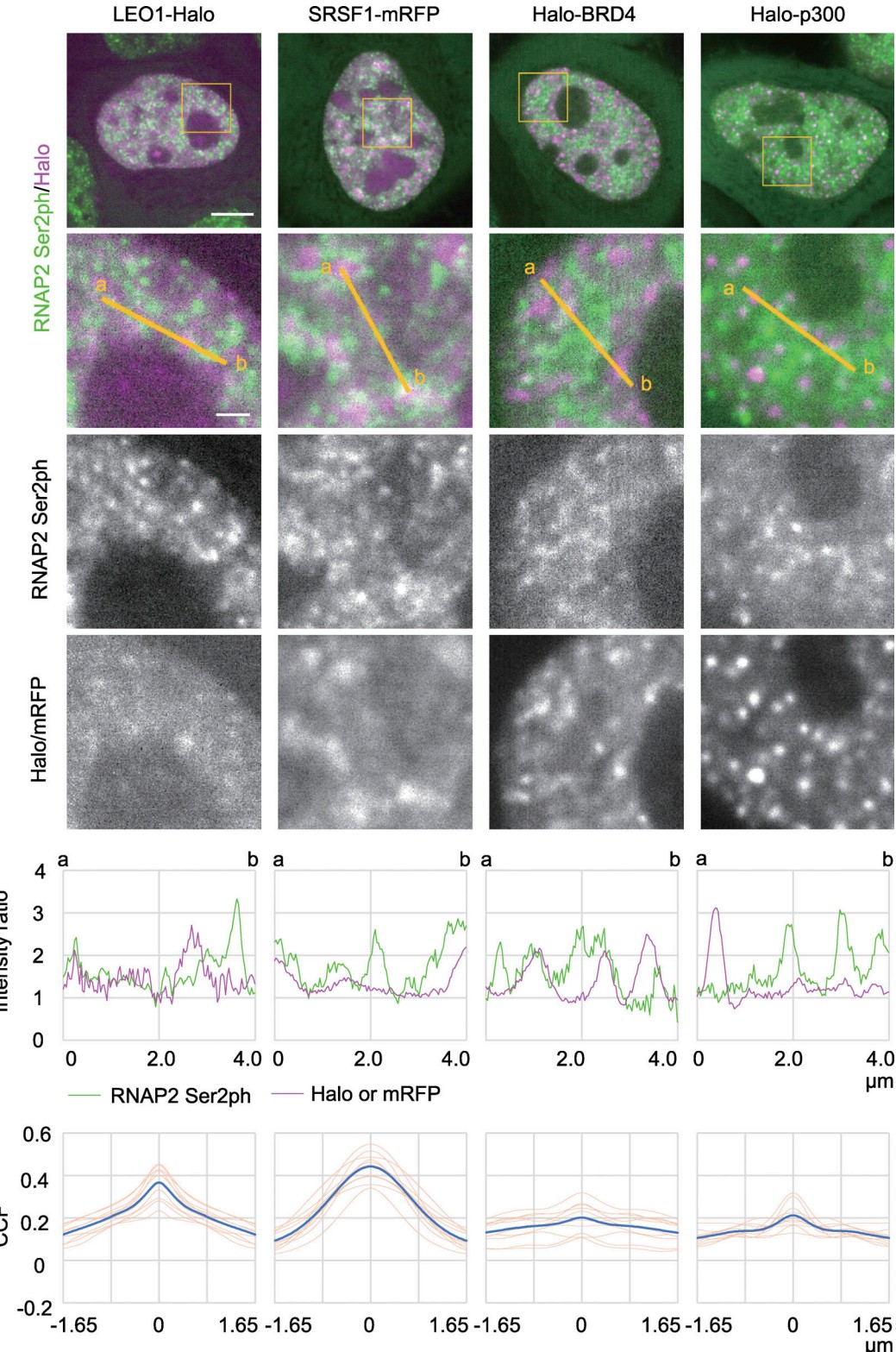

Figure 6. **Localization of LEO1, SRSF1, BRD4, and p300 with respect to RNAP2 Ser2ph-mintbody.** HeLa cells stably expressing RNAP2 Ser2ph-mintbody were transfected with expression vectors for HaloTag- or mRFP-tagged proteins, and confocal images were acquired as described in Fig. 5. The averaged images of 10 consecutive frames (551 ms/frame) are shown with magnified views of areas indicated in orange (middle three rows). Line intensity profiles of 4-µm lines in the second row are shown (fifth row) by normalizing against the average nuclear intensity to yield relative intensity ratios. The CCFs of 10 cells are shown at the bottom (blue, average; orange, individual cells). Scale bars, 5 µm; 1 µm for magnified views.

**Figure 7. Single-molecule analysis of protein mobility relative to RNAP2 Ser2ph–enriched regions.** The mobility of proteins that are highly and lowly associated with RNAP2 Ser2ph-mintbody–enriched regions was quantified using single-molecule trajectories of HaloTag-tagged proteins (RPB3, BRD4, CDK9, p300, H2B, and PCNA) stained with HaloTag TMR ligand recorded at 33.33 ms/frame, which were superimposed upon high-pass-filtered RNAP2 Ser2ph-mintbody images in living HeLa cells. **(A)** Representative single-molecule trajectories of RPB3 superimposed upon RNAP2 Ser2ph-mintbody–enriched regions. RNAP2 Ser2ph-mintbody images were time averaged, high-pass filtered, and normalized to define locally enriched areas. The corresponding histograms of pixel intensities are shown. Right: Trajectories of tracked single molecules are overlaid on the normalized grayscale image. Green and magenta lines indicate

trajectories of mobile and bound molecules, respectively. Scale bar, 5 µm. **(B)** Magnified views of RNAP2 Ser2ph-mintbody images for 0.5 s averaging (left), 16.67 s averaging (middle), and processed with trajectories (right). Scale bars, 1 µm. **(C)** Representative trajectories and MSD curves of bound RPB3 molecules that were highly associated (i) and lowly associated (ii) with RNAP2 Ser2ph-mintbody–enriched regions. Yellow points represent the tracking points with the top two RNAP2 Ser2ph-mintbody intensities, which were averaged and used as the relative RNAP2 Ser2ph-mintbody intensity ($I_{rel\_Ser2ph}$) of the trajectory. The $D$ value (µm²/s) was obtained by linear fitting of the first six steps of MSD (≤0.2 s). Scale bars, 200 nm. **(D)** The distributions of $D$ of the HaloTag-tagged proteins (top) and schematic representation (bottom) showing how to classify the mobile (right) and bound (left) molecules using two-component Gaussian fitting of the distribution of $\log_{10}(D)$. Fitted lines for two-component Gaussian (solid line) and each component (green broken and magenta dotted lines) are indicated. A blue vertical line indicates the threshold $D_{thr}$ (0.065 µm²/s) by which 97.5% of mobile molecules fall into the mobile fraction. Numbers of analyzed trajectories were H2B, 4,290; p300, 5,655; BRD4, 8,298; CDK9, 4,343; RPB3, 4,097; and PCNA, 6,144. **(E and F)** Relative intensities (int.) of RNAP2 Ser2ph-mintbody in a locally selected area at the position of tracked single HaloTag-tagged protein are plotted during the tracked period. Frequencies of trajectories that showed the local relative intensity >0.5 (cyan lines) are indicated. **(E)** Graphs corresponding to the molecules, (i) and (ii), shown in C. **(F)** Trajectories moving into and out of RNAP2 Ser2ph-minbody–enriched regions are shown with the relative intensity by time. Scale bars, 200 nm. **(G)** $D$ and $I_{rel\_Ser2ph}$ of RPB3 are plotted. Each dot represents a single trajectory (molecule). The whole plot (top; 4,097 trajectories) and only the bound fraction (bottom, left; 619 trajectories) are shown with the histogram of $I_{rel\_Ser2ph}$ (bottom, right). Dots (i) and (ii) correspond to molecules shown in C. The top and bottom 25% $I_{rel\_Ser2ph}$ are shown in magenta and green, respectively. **(H)** Frequencies of trajectories that showed local relative RNAP2 Ser2ph-minbody intensity >0.5 in the first to fourth quarters from the highest $I_{rel\_Ser2ph}$ are box plotted (n = 619). Center lines show the medians; box limits indicate the 25th and 75th percentiles; whiskers extend 1.5 times the interquartile range from the 25th and 75th percentiles; means are indicated by ×; gray dots indicate individual data points.

RNAP2 Ser2ph-mintbody foci by tracking their center of mass in each focus with 551-ms imaging intervals using a confocal microscope. The mobility of RNAP2 Ser2ph-mintbody foci was first compared with replication foci marked by Halo-PCNA (Fig. 9, A–C). PCNA foci appear throughout euchromatin in early S phase and later redistribute to heterochromatin at the nuclear periphery and in late S phase (Leonhardt et al., 2000; Fig. S2). RNAP2 Ser2ph-mintbody foci exhibited modestly constrained diffusional motion, with a $D$ of 0.0029 µm²/s and an anomalous exponent ($\alpha$) of 0.69 (such that the MSD = $4Dt^{\alpha}$). This mobility ($\alpha < 1$ and $D \sim 10^{-3}$ µm²/s) is consistent with other measurements of chromatin motion in mammalian cells ($D \sim 10^{-4}$–$10^{-2}$ µm²/s depending on the gene locus and methods; Levi et al., 2005; Chen et al., 2013; Lucas et al., 2014; Germier et al., 2017; Gu et al., 2018; Ma et al., 2019). The anomalous exponents appeared constant during the observation period (Fig. 9 C, bottom). Compared with RNAP2 Ser2ph-mintbody foci, both euchromatic and heterochromatic PCNA foci were less mobile ($D = 0.0023$ µm²/s, $\alpha = 0.69$; and $D = 0.0009$ µm²/s, $\alpha = 0.64$, respectively; Fig. 9, A–C). This result suggests that over a period of several seconds, transcription elongation foci are more mobile than replication foci in which many replication forks in the same chromatin domains are clustered (Jackson and Pombo, 1998).

To compare the mobility of RNAP2 Ser2ph-mintbody foci with chromatin domains that share the same replication timing, we labeled replication domains by DNA replication–mediated cyanine 3 (Cy3)-dUTP incorporation (Manders et al., 1999; Nozaki et al., 2017). The Cy3-labeled replication domains have been shown to behave similarly to individual nucleosomes in the same domains (Nozaki et al., 2017). HeLa cells expressing RNAP2 Ser2ph-mintbody were loaded with Cy3-dUTP to pulse-label replicated chromatin and were further grown for 2 d. As a limited amount of Cy3-dUTP was loaded into cells, DNA regions that were replicated just after the loading were labeled, and then Cy3 signals exhibited characteristic DNA replication foci depending on the stage in S phase (Manders et al., 1999). Once incorporated, Cy3 on DNA persisted after cell divisions, so the replication domains could be tracked in living cells. We classified Cy3-DNA patterns into the early and late replication domains based on the number and intranuclear distribution of Cy3-DNA foci, similar to what we did for Halo-PCNA (Fig. 5, second column; and Fig. 9, A and B). If there were many foci in nuclear interior regions, then we classified them as early, euchromatic domains (Fig. 9 D). If there were fewer foci at the nuclear periphery and around nucleoli, then we classified them as late, heterochromatic domains (Fig. 9 E). Consistent with Halo-PCNA and nucleosome mobility (Nozaki et al., 2017; Shaban et al., 2020), euchromatic domains were more mobile than heterochromatic domains, and RNAP2 Ser2ph-mintbody foci were further mobile than euchromatic domains (Fig. 9 F).

Finally, we measured the mobility of Halo-p300–enriched foci, which are likely to be associated with enhancers (Heintzman et al., 2009; Visel et al., 2009), together with RNAP2 Ser2ph-mintbody foci. Again, RNAP2 Ser2ph-mintbody foci were more mobile than p300-enriched foci, whose mobility was lower than typical euchromatic replication foci, or chromatin domains (Fig. 10). Taken together with the single-molecule tracking data (Figs. 7 and 8), these results suggest that the elongating RNAP2 complexes and molecules therein can move together with DNA templates that are more mobile than chromatin replication domains and p300-enriched foci.

## Discussion

### RNAP2 Ser2ph-mintbody probe

In this study, we developed a genetically encoded probe for visualizing Ser2ph RNAP2 in living cells based on the specific antibody. The expression of functional scFv depends on the folding and stability of the framework regions (Cattaneo and Biocca, 1999; Ewert et al., 2004; Sato et al., 2016; Tjalsma et al., 2021). The original 42B3 clone showed a moderate level of nuclear enrichment, an indication of the functionality of scFv that targets a nuclear protein or its modification. Introducing several amino acid substitutions improved the nuclear enrichment. Although no universal framework is currently available (Kabayama et al., 2020), some framework sequences are more stable than others, and a substitution of the Met residue in a hydrophobic core with a more hydrophobic amino acid, such as Leu and Ile, often increases scFv stability (Tjalsma et al., 2021). In 42B3, Met95 substitution to Val or Ile also improved the nuclear accumulation,

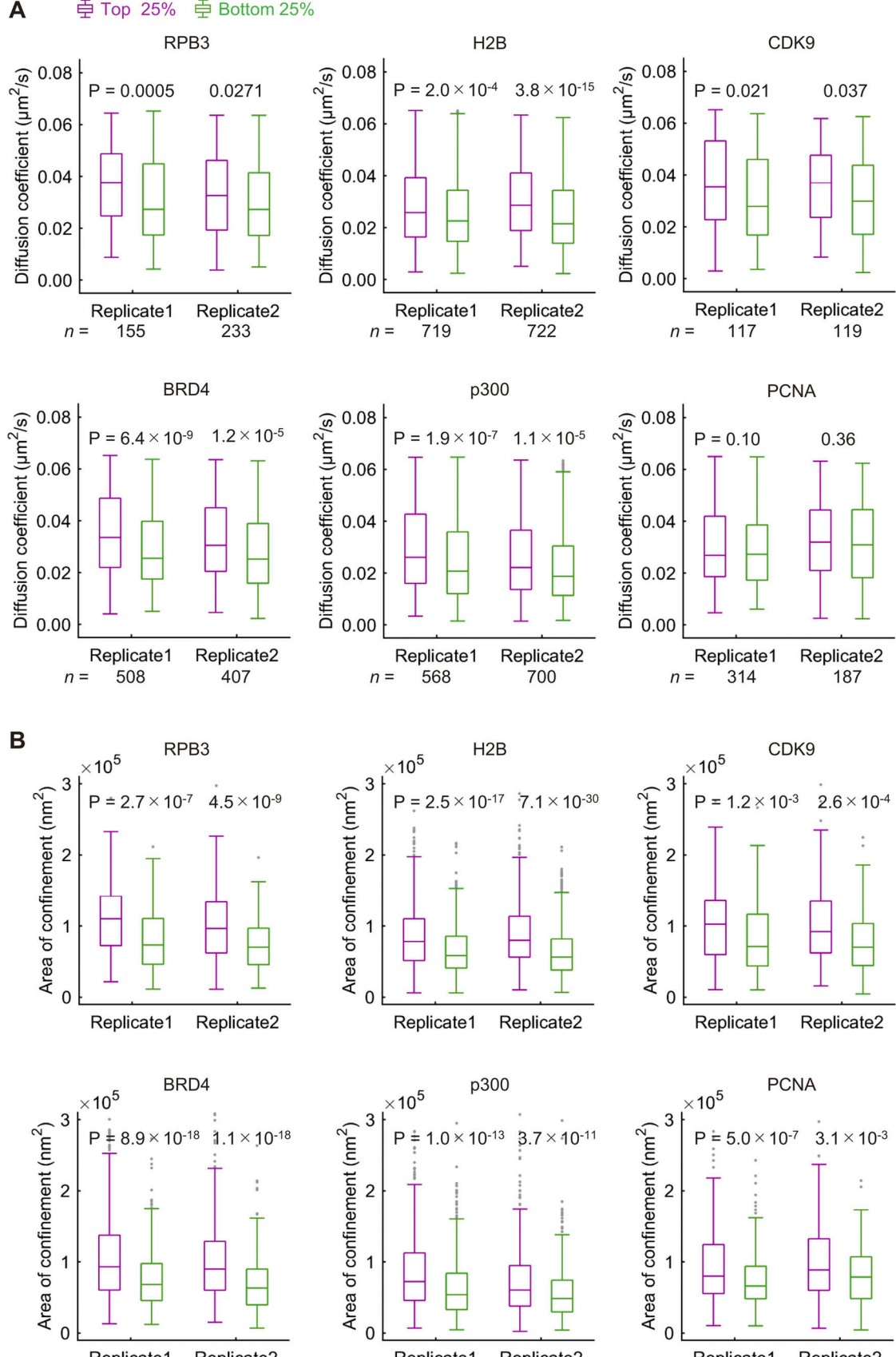

Figure 8. **Diffusion coefficients and confinement areas of HaloTag-tagged proteins that are associated or unassociated with RNAP2 Ser2ph-enriched regions. (A and B)** *D* values and confinement areas of HaloTag-tagged proteins that are associated or unassociated with RNAP2 Ser2ph-enriched regions (A)

and the area of confinement (B) of bound HaloTag-tagged proteins compared between the molecules that were highly associated (magenta, top 25% $I_{rel\_Ser2ph}$) and lowly associated (green, bottom 25% $I_{rel\_Ser2ph}$) with RNAP2 Ser2ph-enriched regions, with P values derived by Mann-Whitney $U$ test and the numbers of analyzed bound molecules ($n$). Box plots from two independent replicate experiments are shown. Center lines show the medians; box limits indicate the 25th and 75th percentiles; whiskers extend 1.5 times the interquartile range from the 25th and 75th percentiles; outliers are represented by gray dots. The corresponding scatterplots with the median $D$ values and effective spring coefficients are shown in Fig. S4.

implying that such substitutions are a good option to improve the folding and stability of scFv in general. Amino acid substitutions that could affect hydrophilic interactions also improved the 42B3 scFv, but less effectively than the Met substitution in the hydrophobic core. These amino acid substitution results are beneficial for the future development of intracellular antibodies for research and therapeutic purposes (Stocks, 2005; Messer and Butler, 2020).

The resulting RNAP2 Ser2ph-mintbody was concentrated in numerous foci in the nucleus depending on RNAP2 Ser2ph, making it possible to detect and track RNAP2 transcription elongation sites in living cells. Even though the binding residence time of the mintbody was less than a few seconds, keeping the expression level low is important so as not to disturb the turnover of phosphorylation. Therefore, we selected cells that expressed relatively low levels of the mintbody and confirmed the sensitivity to flavopiridol.

### RNAP2 Ser2ph-mintbody foci
Ser2ph is a hallmark of an elongating form of RNAP2 (Eick and Geyer, 2013; Zaborowska et al., 2016; Harlen and Churchman, 2017). Lines of evidence support the view that RNAP2 Ser2ph-mintbody foci in living cells represent the sites of RNA-engaged Ser2ph RNAP2. First, the mintbody foci disappeared and reappeared during the prophase to prometaphase and during the telophase to $G_1$, respectively, which is consistent with the substantial repression of RNAP2 transcription during mitosis (Parsons and Spencer, 1997; Liang et al., 2015). Second, the mintbody foci were observed depending on RNAP2 and Ser2ph, demonstrated by inhibitor treatments. The photobleaching assay confirmed the increased mobility of the mintbody due to a loss of the binding target by inhibitors. Third, the binding specificity was validated by an in vitro ELISA using purified protein. Thus, the location of RNAP2 Ser2ph is likely to be detected by the specific mintbody.

As the brightness of single mintbody foci was not at the single-molecule level, multiple mintbody molecules were concentrated in a space less than the ~200-nm diffraction limit. As Ser2 in the CTD repeat can be highly phosphorylated (Schüller et al., 2016), a single RNAP2 could have ~50 mintbody binding targets, although it is not known whether the variations at the distal C-terminal repeats, such as Tyr1-Ser2-Pro3-Thr4-Ser5-Pro6-Lys7 and Tyr1-Ser2-Pro3-Thr4-Ser5-Pro6-Thr7, are still recognized. However, it is unlikely that 50 mintbody molecules are housed on a single RNAP2 because of steric hindrance. In addition, if all Ser2ph is occupied by mintbodies, the binding of endogenous proteins can be blocked, and dephosphorylation may be inhibited. Although it is still possible to have tens of mintbodies per RNAP2 exhibit a fluorescence focus, the number of foci in a HeLa cell does not correspond to the number of total

active RNAP2 molecules, which is estimated to be tens of thousands (Jackson et al., 1998; Kimura et al., 1999). The maximum number of foci detected (~5,000) was rather similar to several thousands of transcription sites containing multiple transcription units, suggesting that the mintbody foci likely correspond to transcription factories (Iborra et al., 1996; Pombo et al., 1999; Cook, 1999).

### Motion of RNAP2 Ser2ph foci
We observed that protein molecules associated with RNAP2 Ser2ph-rich regions were more mobile than those outside the foci. Similarly, RNAP2 Ser2ph foci were more mobile than replication foci and chromatin domains. This observation is consistent with the results of studies that showed the higher mobility of transcribed loci (Gu et al., 2018) and decreased RNAP2 mobility with a CDK9 inhibitor (Shaban et al., 2020) but not with those of other studies that showed that RNAP2 transcription constrains chromatin motion (Ochiai et al., 2015; Germier et al., 2017; Nagashima et al., 2019). Single-molecule analyses have also revealed that transcription constrains chromatin motion (Nagashima et al., 2019; Shaban et al., 2020). The lower mobility of chromatin is still observed in cells treated with CDK9 inhibitors (Germier et al., 2017), implying that transcription-dependent chromatin constraint is mediated through transcription initiation rather than elongation (Babokhov et al., 2020). As only a small fraction of RNAP2 Ser2ph foci was associated with CDK9 (also demonstrated by Ghamari et al., 2013), most mintbody foci corresponded to RNAP2 Ser2ph that were already elongating on the gene body but not at the transcription start sites. In addition, BRD4 foci were even farther apart from RNAP2 Ser2ph or transcripts (Li et al., 2020). From the distinct localization and dynamics of the factors involved in transcription initiation from RNAP2 Ser2ph, it is possible that initiating and elongating RNAP2 complexes are organized differently in space. We anticipate that RNAP2 Ser2ph-mintbody will help address the question whether new initiation events occur at or distal to preexisting elongation RNAP2 foci.

Typical reporter genes that are short and highly transcribed (Ochiai et al., 2015; Germier et al., 2017; Forero-Quintero et al., 2021) are likely dominated by the initiating form of RNAP2, even though elongating RNAP2 is also present, which may result in confined chromatin motion. The endogenous genes are often periodically transcribed by bursting (Suter et al., 2011, Fukaya et al., 2016; Larsson et al., 2019; Ochiai et al., 2020), and the elongating RNAP2 can be separated from the transcription start sites and enhancers (Li et al., 2020). In this case, the elongating Ser2-phosphorylated RNAP2 clusters on a single gene by transcription bursting, or different genes become free of the confined initiating complexes and exhibit more diffuse motion. The chromatin structure needs to be extensively loosened during

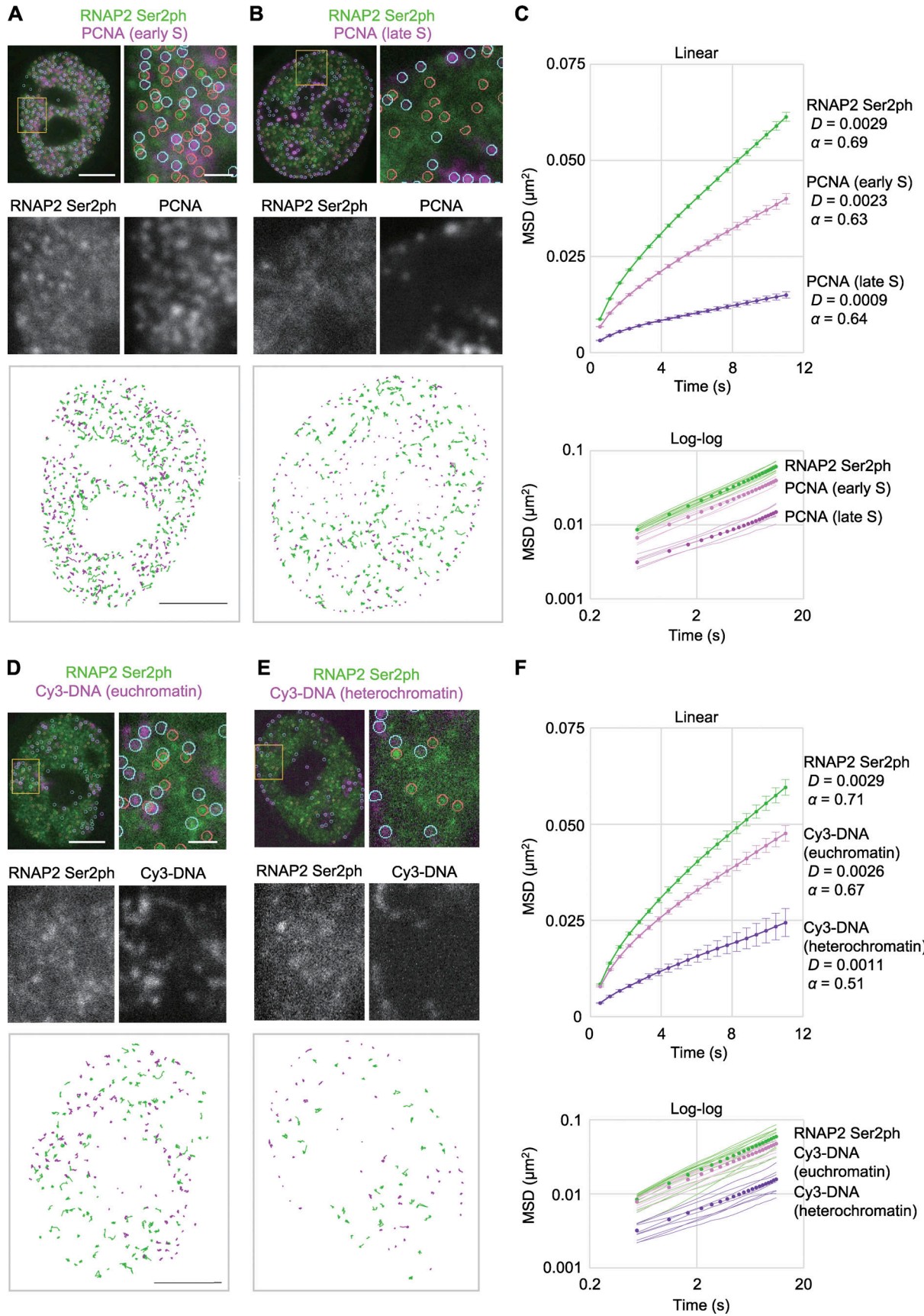

Figure 9. **Mobility of RNAP2 Ser2ph foci compared with PCNA foci and replication domains. (A–C)** RNAP2 Ser2ph-mintbody and Halo-PCNA were expressed in HeLa cells and fluorescence images were acquired at 551 ms/frame using a high-resolution confocal system. RNAP2 Ser2ph-mintbody and Halo-

PCNA foci in cells that exhibited euchromatic (A) and heterochromatic (B) PCNA patterns at the early and late S stages, respectively, were tracked to measure their mobilities. PCNA is concentrated in euchromatic foci in the nuclear interior region in the early S phase (A) and then in heterochromatic foci at the nuclear periphery and around nucleoli in the late S phase (B); see also Fig. 5 A. The whole nucleus, magnified views with indications of tracked foci (orange and cyan circles for mintbody and PCNA, respectively), and traces of individual foci are shown. **(C)** MSDs (means of all foci) are shown in linear scale (top) with SEM, $D$ values ($\mu m^2/s$), and anomalous exponents ($\alpha$), and in double logarithmic scale (bottom) with means in single cells (thin lines). Numbers of analyzed foci are 3,993 (RNAP2 Ser2ph), 1,226 (euchromatic PCNA), and 925 (heterochromatic PCNA), from 12 cells for RNAP2 Ser2ph (143–468 foci/cell) and 6 cells each for euchromatic and heterochromatic PCNA foci (31–291 and 72–249 foci/cell, respectively). **(D–F)** HeLa cells stably expressing RNAP2 Ser2ph-mintbody were loaded with Cy3-dUTP to pulse-label chromatin domains that replicated just after loading, and cells were further grown for 2 d. Fluorescence images were acquired, foci were tracked, and MSDs were plotted, as in A–C. Cells were classified into euchromatic (D) and heterochromatic (E) Cy3-DNA patterns, just as were PCNA-expressing cells. **(D)** Numbers of analyzed foci are 1,329 (RNAP2 Ser2ph), 951 (euchromatic Cy3), and 623 (heterochromatic Cy3) from 15 cells for RNAP2 Ser2ph (41–194 foci/cell) and 9 cells each for euchromatic and heterochromatic Cy3-DNA foci (49–149 and 37–123 foci/cell, respectively). Scale bars, 5 μm; 1 μm for magnified views.

ongoing transcription, and the elongating complex on such loosened chromatin fibers can become less constrained. The higher mobility at the single-molecule level may also be associated with the dynamic structure of RNAP2 clusters. It is thus tempting to speculate that clustered RNAP2 molecules that transcribe different genes are pulled by chromatin motions from different angles at different time points, resulting in more dynamic mobility than euchromatin domains.

## Materials and methods
### Cells and transfection
HeLa cells were grown in DMEM high-glucose medium (Nacalai Tesque) containing 10% FBS (Gibco) and 1% L-glutamine–penicillin–streptomycin solution (GPS; Sigma-Aldrich) at 37°C in a 5% $CO_2$ atmosphere. For transfection, FuGENE HD (Promega) was used according to the manufacturer's instructions. Briefly, 2 μg DNA was mixed with 6 μl of FuGENE HD in 100 μl

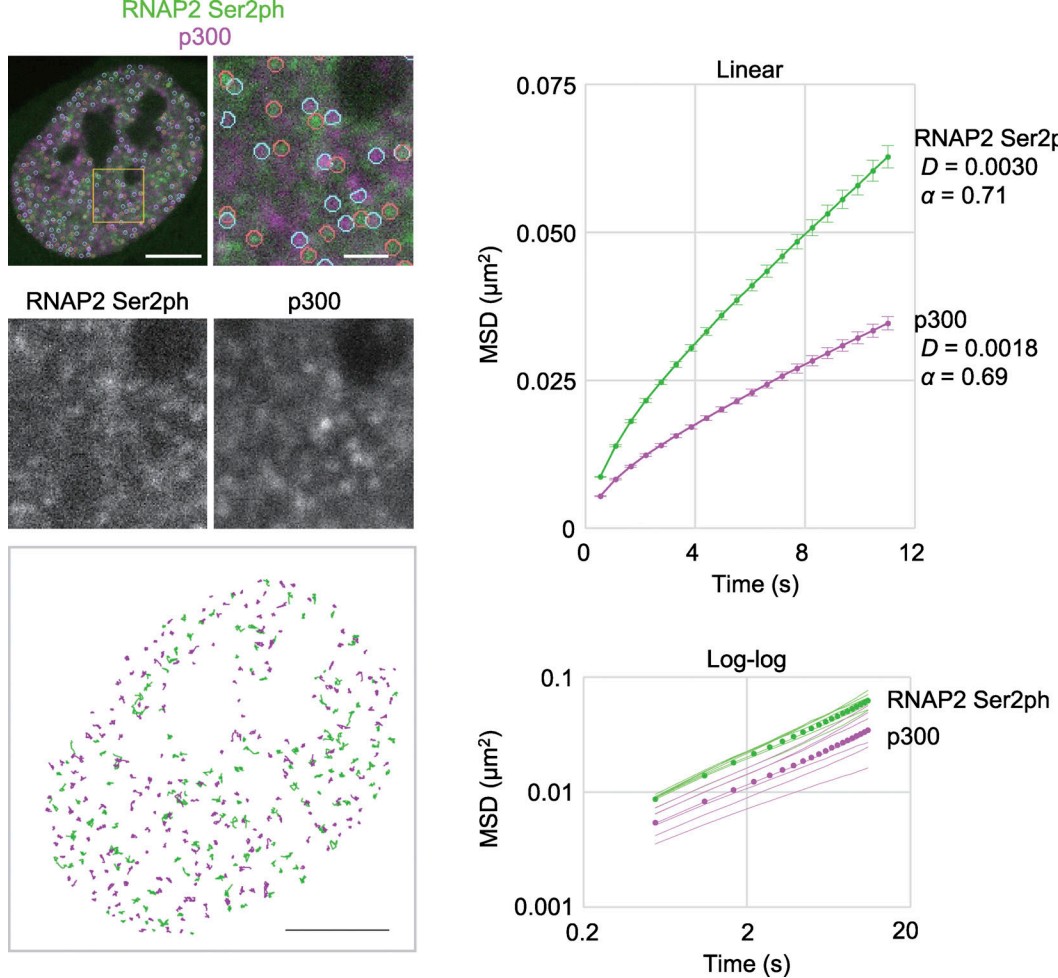

Figure 10. **Mobility of RNAP2 Ser2ph foci compared with p300-enriched foci.** RNAP2 Ser2ph-mintbody and Halo-p300 were expressed in HeLa cells, and fluorescence images were acquired, foci were tracked, and MSDs with SEM. were plotted, as in Fig. 9. Numbers of analyzed foci are 1,554 (RNAP2 Ser2ph) and 2,026 (p300) from 7 cells each for RNAP2 Ser2ph (111–344 foci/cell) and p300 (242–387 foci/cell). Scale bars, 5 μm; 1 μm for magnified views.

of Opti-MEM (Thermo Fisher Scientific) and incubated at RT for 30 min before being added to HeLa cells grown in 35-mm glass-bottomed dishes (AGC Technology Solutions) to 40–70% confluence. To establish stable cell lines, 1.6 µg PB533- or PB510-based PiggyBac plasmid and 0.4 µg transposase expression vector (System Biosciences) were used, and cells were selected in 1 mg/ml G418 (Nacalai Tesque) for RNAP2 Ser2ph-mintbody and H2B-Halo or 1 µg/ml puromycin (InvivoGen) for mCherry-PCNA. Cells expressing RNAP2 Ser2ph-mintbody at low levels were collected using a cell sorter (SH800; Sony). Before live-cell imaging, the medium was replaced with FluoroBrite (Thermo Fisher Scientific) containing 10% FBS and 1% GPS.

## Cloning and evaluation of antibody variable fragments encoding 42B3 and its mutants

Mouse hybridoma cell lines expressing antibodies specific to RNAP2 Ser2ph were generated and screened by Mab Institute Inc. (Nagano, Japan), as previously described (Kimura et al., 2008). Clone 42B3 was originally generated against a peptide, Ser-Tyr-Ser-Pro-Thr-Ser-Pro-phosphoSer-Tyr-phosphoSer-Pro-Thr-Ser-Pro-Ser-Tyr-Ser-Pro-Cys, harboring Ser7ph and Ser2ph at the CTD sequence, but it was reacted with peptides containing Ser2ph, regardless of phosphorylation state of Ser7 (Fig. 3). RNA was extracted from the hybridoma cells using TRIzol (Thermo Fisher Scientific) and subjected to RNA sequencing to determine the DNA sequence encoding the IgG heavy and light chains, as described previously (Kuniyoshi et al., 2016). The variable regions of heavy ($V_H$) and light ($V_L$) chains of 42B3 were amplified and connected (Sato et al., 2018; Tjalsma et al., 2021) using sets of primers specific for $V_H$ ($V_H\_s$, $V_H\_as$), $V_L$ ($V_L\_s$, $V_L\_as$), and the linker (LINK primer1 and LINK primer2), which are listed in Table S1. The resulting scFv fragments were further amplified using the 5′ and 3′ primers (scFv primer_s and scFv primer_as) and cloned into the sfGFP-N1 vector (Addgene 54737, deposited by Michael Davidson and Geoffrey Waldo; Pédelacq et al., 2006) to generate a 42B3-sfGFP expression vector.

Point mutation primers were designed (Table S1) for PCR using 42B3-scFv as a template with PrimeSTAR (Takara Bio). KOD One PCR Master Mix Blue (Toyobo) was used for the M95V mutation. After PCRs, the amplified products were treated with DpnI (0.4 U/µl; 1 h) to digest the template before mixing with competent cells. The nucleotide sequence of the resulting plasmids was verified by Sanger sequencing. 42B3 scFv mutants were then subcloned into a PB533-based vector (System Biosciences) by restriction enzyme digestion (EcoRI and NotI) and ligation.

To screen mintbodies and mutants, a point-scan confocal microscope (FluoView FV1000; Olympus) operated by the built-in software FluoView version 4.2, featuring a heated stage (Tokai Hit; 37°C, 5% $CO_2$), was used with a UPlanApoN 60× OSC oil immersion lens (NA 1.4) using the following settings: 512 × 512 pixels, pixel dwell time 4.0 µs, pinhole 100 µm, zoom ×3.0, line averaging ×3, with a 543-nm (50%) laser line (Fig. 1 B; for HeLa cells that stably express mCherry-tagged mintbodies), or zoom ×5.0, line averaging was ×4, with a 488-nm (3%) laser line (Fig. S1; for HeLa cells that transiently express sfGFP-tagged mintbodies).

To determine the N/C ratio of scFv-mCherry (Fig. 1 C), the whole-cell, nucleus, and cell-free background regions were selected manually, and the intensities and areas were measured using ImageJ Fiji 1.52d software (https://imagej.net/Fiji). Net intensities of the whole cell and nucleus were obtained by subtracting background intensity, and the cytoplasmic intensity was calculated by dividing the total cytoplasmic intensity ([whole cell intensity] × [whole cell area] – [nuclear intensity] × [nuclear area]) by the cytoplasmic area ([whole cell area] – [nuclear area]). The N/C ratio was obtained by dividing the cytoplasmic intensity by the nuclear intensity. Box plots were created using BoxPlotR (http://shiny.chemgrid.org/boxplotr/).

## Purification of RNAP2 Ser2ph-mintbody

The RNAP2 Ser2ph-mintbody sequence was inserted into pTrcHis-C (Thermo Fisher Scientific) to harbor the His-tag at the N-terminus. *E. coli* BL21 (DE3) harboring the expression plasmid was grown in 100 ml of YTG (1% tryptone, 0.5% yeast extract, 0.5% NaCl, and 0.2% glucose) for 20 h at 18°C to reach the early stationary phase, diluted to a final volume of 2 liters of YTG, and further grown for 8 h at 15°C. IPTG (Nacalai Tesque) was added at a final concentration of 1 mM, and *E. coli* culture was further incubated for 12 h at 15°C before collecting cells by centrifugation at 4,000 × $g$ for 10 min at 4°C. After the supernatant was discarded, the pellet was frozen at –80°C. The pellet was thawed on ice and then suspended in 20 ml of Buffer L (50 mM Tris-HCl, pH 8.0, 300 mM NaCl, 5% glycerol) containing 1 mg/ml lysozyme (Nacalai Tesque) and 1% proteinase inhibitor cocktail (Nacalai Tesque). Cells were lysed in icy water by sonication using a Sonifier 250 (Branson; duty cycle 30%, output 1.2; 60 s; 30 times with 60-s intervals). The supernatant was collected after centrifugation at 15,000 ×$g$ for 15 min at 4°C and applied to a 20-ml open column (Poly-Prep Chromatography Columns; Bio-Rad Laboratories) containing 500 µl of Ni-nitrilotriacetic acid agarose (Qiagen) preequilibrated with Buffer L at 4°C. After the column was washed twice with 10 ml of Buffer L, bound proteins were eluted three times with 1 ml of elution buffer (Buffer L containing 150 mM imidazole, pH 8.0) and then dialyzed overnight in 1 liter of starting buffer (10 mM Tris-HCl, pH 7.0, 50 mM NaCl). After replacing with new buffer, dialysis was continued for a further 20 h. The dialyzed sample was filtered and applied to a HiTrap Q column (GE Healthcare) and fractionated by linear gradient with end buffer (10 mM Tris-HCl, 1 M NaCl) using an AKTAprime plus system (GE Healthcare) at 4°C. Proteins in each fraction were analyzed by 10–20% SDS-PAGE, and fractions exclusively containing a protein with the expected size of His-RNAP2 Ser2ph-mintbody were pooled. His-tag was removed using an Enterokinase Cleavage Capture kit (Novagen), according to the manufacturer's instructions. Diluted recombinant enterokinase (10 µl) and 2 µl of 1 M $CaCl_2$ were added to 1 ml of purified protein sample. After overnight incubation at RT, enterokinase was removed with Ekapture Agarose. The concentration of the purified RNAP2 Ser2ph-mintbody was determined using a Bradford protein assay (Bio-Rad Laboratories) with BSA as the standard.

## ELISA

Microtiter ELISA plates (Greiner Bio-One) were coated with 1 µg/ml BSA conjugated with RNAP2 CTD repeat peptides

containing phosphorylated amino acids (Mab Institute Inc.; Fig. 4 B) overnight at 4°C, and they were washed three times with PBS. Each well was incubated with 100 µl of Blocking One-P (Nacalai Tesque) for 20 min at RT, washed with PBS, and then incubated with 50 µl of a 1:3 dilution series of purified RNAP2 Ser2ph-mintbody and IgG antibodies specific for Ser2ph (CMA602 and 42B3) and Ser5ph (CMA603) in PBS overnight. RNAP2 Ser2ph-mintbody and antibodies were diluted from 2.7 µg/ml and 300 ng/ml, respectively. Microtiter plates were washed three times with PBS and incubated with HRP-conjugated anti-GFP (MBL; 1:2,000) and HRP-conjugated antimouse IgG (Jackson ImmunoResearch; 1:10,000) for RNAP2 Ser2ph-mintbody and IgG, respectively, for 90 min at RT. After washing four times with PBS, 100 µl of o-phenylenediamine solution (freshly prepared by dissolving one tablet containing 13 mg o-phenylenediamine-2HCl into 50 ml of 0.1 M sodium citrate buffer, pH 5.0, and 15 µl 30% hydrogen peroxide) was added. After incubating at RT until a yellow color was observed, the absorbance was measured at 490 nm, with a reference at 600 nm, using a Varioskan (Thermo Fisher Scientific).

### Live imaging of RNAP2 Ser2ph-mintbody and inhibitor treatments

Live-cell imaging of HeLa cells expressing RNAP2 Ser2ph-mintbody was performed using a high-resolution spinning-disk confocal microscope featuring microlens-associated pinholes and a 3.2× tube lens to theoretically improve the optical resolution 1.4-fold (Ixplore SpinSR; Olympus) equipped with a UplanApo 60× OHR (NA 1.5) and a UplanApoN 60× OSC2 (NA 1.4) objective lens for single-color and multicolor fluorescence imaging, respectively, using a 488-nm laser (OBIS 488 LS, 100 mW; Coherent), a 561-nm laser (OBIS 561 LS, 100 mW; Coherent), a scientific complementary metal oxide semiconductor camera (ORCA Flash 4; Hamamatsu Photonics), and a heated stage (Tokai Hit; 37°C, 5% $CO_2$), under the operation of cellSens Dimension 3.1 software (Olympus).

To measure the number and size of RNAP2 Ser2ph-mintbody foci (Fig. 1, D–F), z-stack images were acquired using a 488-nm laser (100% transmission; 300 ms) at 0.2-µm intervals. Using ImageJ Fiji 1.52d software, line profiles of single foci were drawn and fit using a Gaussian distribution, $Y = a + b*exp(-[x - c]^2/[2d^2])$, to calculate the full width at half maximum. To measure the number of foci per nucleus, foci were detected using NIS Elements (Nikon) with the spot detection function (binary > spot detection > bright spots with a typical diameter of 7 pixels) in all z-sections. The contrast in the spot detection function was adjusted to maximally detect nuclear spots without cytoplasmic spots. As one focus can be detected in three consecutive sections at the maximum, the total number of detected foci in the whole nucleus was divided by 3.

For live-cell imaging of HeLa cells that stably expressed both RNAP2 Ser2ph-minbody and H2B-Halo during mitosis (Fig. 2), cells were incubated in 50 nM HaloTag TMR ligand (Promega) for 30 min, and the medium was replaced with FluoroBrite (Thermo Fisher Scientific) with supplements. Images were collected sequentially with a 488-nm laser (100% transmission; 300 ms) and a 561-nm laser (1% transmission; 300 ms) every 1 min.

To demonstrate the relative localization of condensed chromosomes and RNAP2 Ser2ph-mintbody, the relative intensity profiles of 3-pixel-thick 2-µm lines were drawn using Fiji 1.52d. The intensities were normalized by setting the maximum and minimum line-pixel intensities to 1 and 0, respectively.

For inhibitor treatments, after 10 images were acquired using a 488-nm laser (10% transmission; 200 ms) without intervals, flavopiridol (at a final concentration of 1 µM; Sigma-Aldrich), triptolide (at a final concentration of 5 µM; Tocris), or the vehicle (DMSO; 1:10,000 dilution) was added, and 10 images without intervals were acquired every 30 min for 2.5 h. The number of foci was counted using NIS Elements (Nikon) as describe above.

### FRAP

HeLa cells stably expressing RNAP2 Ser2ph-mintbody were grown on a 35-mm dish, and FRAP was performed in inhibitors that were added 2–4 h before the experiment. A dish was set on a heated stage (Tokai Hit) at 37°C under 5% $CO_2$ (Tokken) on a point-scan confocal microscope (FV1000; Olympus) with a Uplan Apo 60× OSC (NA 1.4) lens under the operation of built-in ASW version 4.2 software (Olympus). Overall, 250 confocal images were collected every 47.8 ms (24 × 128 pixels; 2-µs pixel dwell time; zoom ×8.0; pinhole 800 µm; 0.2% transmission for a 488-nm line from a 20-mW multi-Ar+ ion laser), a 2-µm diameter circle was bleached using a 488-nm laser line with 100% transmission for 55 ms, and a further 50 images were collected using the original settings. The intensity of the bleached, nuclear reference, and background areas was measured using ASW version 4.2 (Olympus). After subtracting the background from the bleached and nuclear reference areas, the relative fluorescence in the bleached area was obtained by two-step normalization. The intensity in the bleached area at each time point was obtained by dividing by that in the nuclear reference area, and the resulting relative intensity was then normalized using the average before bleaching.

### Expression and imaging of fluorescently tagged proteins

To construct expression vectors for HaloTag-tagged proteins (CDK9, CDK12, and LEO1), total RNA was prepared from HeLa cells using TRIzol (Thermo Fisher Scientific) according to the manufacturer's instructions, and cDNA was synthesized using the SuperScript III First-Strand Synthesis System for RT-PCR (Thermo Fisher Scientific). To the total RNA (2.3 µg in 1 µl), 7 µl of nuclease-free water, 1 µl of 50 µM oligo(dT)$_{20}$, and 4 µl of 2.5 mM deoxynucleotide triphosphate were added, and the mixture was incubated at 65°C for 5 min and chilled on ice for 1 min. After 4 µl of 5× First-Strand buffer, 1 µl of 0.1 M DTT, 1 µl of Rnase OUT recombinant Rnase inhibitor, and 1 µl of SuperScript III RT were added, the mixture was incubated for 60 min at 50°C and then for 15 min at 70°C before the addition of 1 µl of Rnase H (2 U/µl) and incubation for 15 min at 37°C. cDNA was purified using a PCR purification kit (Qiagen), and the concentration was measured using NanoDrop (Thermo Fisher Scientific). To amplify specific protein sequences, PCR was performed in a reaction mixture containing 5 µl of 5× Q5 buffer, 2 µl of 2.5 mM deoxynucleotide triphosphates, 0.6 µl each of forward

and reverse primers (Table S1), 1 µl of cDNA, 15.55 µl of water, and 0.25 µl of Q5 DNA Polymerase (New England Biolabs), with the following PCR cycle: 1 min at 98°C, 35 cycles of 10 s at 98°C, 5 s at 72°C, 30 s at 72°C, and 2 min at 72°C. The PCR products were cloned into the HaloTag vector based on PB533 using an infusion system (Takara Bio). The expression vectors of H2B-Halo and Halo-PCNA were constructed by inserting their coding sequence (Kanda et al., 1998; Leonhardt et al., 2000) into the same HaloTag PB533 vector (Sato and Kimura, 2021). The expression vectors of Halo-BRD4 and Halo-p300 were Kazusa clones (Promega). The expression vector of SF2(SRSF1/ASF)-mRFP has been described previously (Yomoda et al., 2008).

HeLa cells stably expressing both RNAP2 Ser2ph-mintbody and mCherry-PCNA and cells expressing RNAP2 Ser2ph-mintbody were established by transfection of mCherry-PCNA expression vector based on PB510 (System Biosciences; Imada et al., 2021). After selecting in 1 µg/ml puromycin (InvivoGen), mCherry-positive cells were collected using a cell sorter (SH800; Sony). Fluorescence images of RNAP2 Ser2ph-mintbody and mCherry-PCNA were sequentially acquired using an Ixplore SpinSR (Olympus) with a 488-nm laser (30% transmission; 300 ms) and a 561-nm laser (30% transmission; 200 ms) every 30 min for 36 h.

HeLa cells stably expressing RNAP2 Ser2ph-mintbody were grown on glass-bottomed dishes, transfected with the expression vectors of HaloTag-tagged proteins, and, the next day, cells were incubated in 50 nM HaloTag TMR Ligand (Promega) for 30 min before washing with FluoroBrite (Thermo Fisher Scientific) containing supplements. To evaluate the effect of Halo-p300 expression levels on the localization, HeLa cells were cotransfected with the expression vectors for RNAP2 Ser2ph-mintbody and Halo-p300 and left for 2 d before labeling with HaloTag TMR Ligand and imaging. Fluorescence images of sfGFP and TMR or mRFP1 were sequentially acquired using an Ixplore SpinSR (Olympus) with a 488-nm laser (50% transmission; 300 ms) and a 561-nm laser (50% transmission for RPB3, BRD4, p300, SRSF1, PCNA, and H2B; 100% for CDK9, CDK12, and LEO1; 200 ms).

ImageJ Fiji software was used for colocalization analyses (Figs. 5 and 6). To draw a line profile, a 4-pixel-thick line was drawn and expressed as the relative intensity to the average nuclear intensity after subtracting cytoplasmic intensity. CCF was calculated using Just Another Colocalisation Plugin for ImageJ software. The Gaussian filter (radius 2.0 pixels) was applied to diminish the pseudocorrelation caused by pixel-to-pixel sensitivity variation, and the average fluorescence intensity of the nucleoplasm was subtracted from the image for thresholding. The CCF was measured with the x-shift set to 100 pixels (3.3 µm) without rotation and with rotation by 90°, 180°, and 270°, and the resulting four graphs were averaged and plotted in the range of −50 ≤ x ≤ 50 pixels. Pearson's correlation of the nucleus was obtained using NIS Elements (Nikon). Graphs were created using Excel 2019 (Microsoft).

### Single-molecule analysis

HeLa cells stably expressing RNAP2 Ser2ph-mintbody were transfected with expression vectors for HaloTag-tagged proteins

(RPB3, BRD4, CDK9, p300, H2B, and PCNA) and were stained with 24 pM HaloTag TMR ligand for 30 min just before live-cell imaging. Highly expressing cells, which were brightly labeled even with the low ligand concentration, were not used for the analysis. The medium was then replaced with FluoroBrite-containing supplements. Single-molecule imaging was performed on a custom-build inverted microscope (IX83; Olympus) equipped with a 488-nm laser (OBIS 488 LS, 100 mW; Coherent), a 561-nm laser (OBIS 561 LS, 100 mW; Coherent), a 100× objective lens (PlanApo NA 1.45 total internal refluorescence microscope; Olympus), and an intermediate magnification lens (2×; Olympus; Tokunaga et al., 2008; Lim et al., 2018). HILO illuminations of both laser lines were achieved using a CellTIRF system (Olympus). Images were recorded at 33.33 ms per frame for 500 frames using two EM charge-coupled device cameras (C9100-13; Hamamatsu Photonics) and AQUACOSMOS software (Hamamatsu Photonics). The spatial shift between the two camera images was corrected using the images of a 10-µm square lattice with ImageConverter software (Olympus). Cells were maintained at 37°C during observation using a stagetop incubator and objective heater (Tokai Hit). Then, 15 cells were observed for each sample, and two independent experiments were performed.

Single-molecule tracking was performed using modified u-track MATLAB code (Jaqaman et al., 2008). Briefly, single-molecule positions were detected using the detectSubResFeatures2D function in u-track and were precisely determined by 2D Gaussian fitting. The localizations were linked using the trackCloseGapsKalmanSparse function with a maximum step length of 720 nm. Neither gap closing nor Kalman filtering in the u-track were used. The trajectories that lasted at least six steps (200 ms) were used to calculate MSD (Ito et al., 2017). The first six steps of MSD were fit with MSD = $4Dt$ to determine the $D$ of an individual trajectory; as the period of analysis was very short (200 ms), the simple diffusion model was used. The maximum threshold of the diffusion coefficient $D_{thr}$ for bound molecules was determined from the distribution of $\log_{10}(D)$. A stacked histogram from all samples was fitted to a double Gaussian function, and the mean – 2 SD of the fast fraction was used as $D_{thr}$. Similar values for $D_{thr}$ (0.065 and 0.063 µm²/s) were obtained from two independent replicate experiments.

The relative intensity of RNAP2 Ser2ph-mintbody was obtained from a 500-frame (16.67 s) averaged image that was recorded simultaneously with single-molecule imaging. Nonuniform lighting of the average image was minimized by a 2-µm high-pass image filter. The range of the intensity from −2 to +2 SD from the mean was normalized to that from 0 to 1. The normalized intensity of the mintbody at each pixel point on the trajectory of HaloTag-tagged protein was calculated. The pixel number ratio at the intensity >0.5 was determined as the frequency of $I_{>0.5}$. Since each trajectory traverses various mintbody intensity regions, the average of the top two intensities of the points on each trajectory was used as the relative RNAP2 Ser2ph-mintbody intensity ($I_{rel\_Ser2ph}$) of the trajectory. The difference between the $D$ values of bound molecules ($\leq D_{thr}$) that were highly associated (the top $I_{rel\_Ser2ph}$) and lowly associated (the bottom $I_{rel\_Ser2ph}$) with RNAP2 Ser2ph-mintbody–enriched

regions was analyzed using the Mann-Whitney $U$ test. The confinement area of each trajectory was calculated by drawing a confidence ellipse that contains 95% of trajectory points (Germier et al., 2017). The effective spring coefficient was calculated using linear regression of the centripetal displacement on the distance from the center of mass (Shukron et al., 2019).

### Tracking RNAP2 Ser2ph-mintbody with Halo-PCNA, Halo-p300, and Cy3-DNA foci

HeLa cells or those stably expressing RNAP2 Ser2ph-mintbody were transfected with expression vectors for Halo-PCNA and Halo-p300, and, 2 d later, cells were labeled with HaloTag TMR Ligand (Promega) and prepared for live imaging as described above. For tracking chromatin domains, HeLa cells expressing RNAP2 Ser2ph-mintbody were bead loaded with 0.1 mM Cy3-dUTP (PerkinElmer; Manders et al., 1999; Sato et al., 2018) and cultured for 2 d. On the day of imaging, the medium was replaced with FluoroBrite (Thermo Fisher Scientific) containing supplements. Fluorescence images of sfGFP and TMR, or Cy3, were sequentially acquired at 551 ms/frame using an Ixplore SpinSR (Olympus) with a 488-nm laser (50% transmission; 300 ms) and a 561-nm laser (50% and 10% transmission for TMR and Cy3, respectively; 200 ms).

For tracking foci, RNAP2 Ser2ph-mintbody, Halo-PCNA, Halo-p300, and Cy3 foci were detected with NIS Elements (Nikon) using binary > spot detection > bright spots with the following parameters: typical diameter pixels 10 (RNAP2 Ser2-mintbody, PCNA, and p300) and 12 (Cy3); contrast 4.94 (RNAP2 Ser2ph) or values adjusted to maximally detect nuclear spots without (PCNA and p300) or with little (Cy3) cytoplasmic spots; SD multiplication factor 2 SD; maximum object speed 200 µm/s; maximum gap size 3; and delete tracks <30 frames. In the case where cytoplasmic spots were detected (Cy3), they were removed manually. MSDs up to 20 steps (11 s) were used for fitting using MSD = $4Dt^\alpha$, where $D$ is the diffusion coefficient; $t$ is the elapsed time; and $\alpha$ is the anomalous exponent ($0 < \alpha < 2$). MATLAB codes are available at GitHub (https://github.com/Kimura-Lab/Uchino-et-al.-2021).

### Online supplemental material

Fig. S1 shows the initial screening data of various RNAP2-specific scFvs, the amino acid sequence of 42B3 scFv, microscopic images of 42B3 scFv mutants expressed in HeLa cells, and the structural models of RNAP2 Ser2ph-mintbody, related to Fig. 1. Fig. S2 shows the dynamic redistribution of Halo-PCNA in HeLa cells expressing RNAP2 Ser2ph-mintbody during the cell cycle, related to Figs. 5 and 9. Fig. S3 shows the effect of expression levels of p300 on the focus formation and the CCF with RNAP2 Ser2ph-mintbody, related to Fig. 6. Fig. S4 and Fig. S5 show supporting data for single-molecule tracking analyses, related to Figs. 7 and 8. Fig. S4 shows $D$ values and effective spring coefficients of HaloTag-tagged proteins associated with different RNAP2 Ser2ph-mintbody intensities, related to Figs. 7 and 8. Fig. S5 validates the significance of the difference between the median $D$ values in bound molecules that are highly and lowly associated with RNAP Ser2ph-mintbody–enriched regions, related to Fig. 7. Video 1 and Video 2 present the dynamics of RNAP2 Ser2ph-mintbody foci with respect to condensing and decondensing chromosomes during the entry into and exit from mitosis, respectively, related to Fig. 2. Table S1 lists reagents and resources used in the study.

### Data availability

The nucleotide sequence data of 42B3(R78K/A80T/M95V)-scFv are available in several public databases (DNA Data Bank of Japan/EMBL/GenBank) under accession number LC628457.

## Acknowledgments

We thank T.J. Stasevich and M. Saxton for valuable comments and English-language proofreading; H. Ochiai for valuable comments; M.C. Cardoso for the PCNA construct; members of the Kimura laboratory for constructive discussion; and the Biomaterials Analysis Division, Open Facility Center, Tokyo Institute of Technology for DNA-sequencing analysis.

This work was supported by Ministry of Education, Culture, Sports, Science and Technology/Japan Society for the Promotion of Science KAKENHI (JP20K15755 to Y. Ito; JP17KK0143 and JP20K06484 to Y. Sato; JP17K17719 to T. Handa; JP20H04846, JP20H00456, and JP21H00232 to Y. Ohkawa; JP19H03192 to M. Tokunaga; JP17H01417 and JP21H04764 to H. Kimura; and JP18H05527 to Y. Ito, Y. Ohkawa, and H. Kimura) and Japan Science and Technology Agency CREST (JPMJCR20S6 to Y. Sato and JPMJCR16G1 to Y. Ohkawa and H. Kimura).

The authors declare no competing financial interests.

Author contributions: Conceptualization, H. Kimura; investigation, S. Uchino, Y. Ito., Y. Sato, and H. Kimura; formal analysis, S. Uchino and Y. Ito; methodology, Y. Ito and Y. Sato; resources, Y. Sato, T. Handa, Y. Ohkawa, and M. Tokunaga; writing – original draft, S. Uchino, Y. Ito, and H. Kimura; writing – review and editing, Y. Sato, T. Handa, Y. Ohkawa, and M. Tokunaga; funding acquisition, Y. Ito, Y. Sato, Y. Ohkawa, T. Handa, M. Tokunaga, and H. Kimura; supervision, H. Kimura.

Submitted: 29 April 2021

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

# Supplemental material

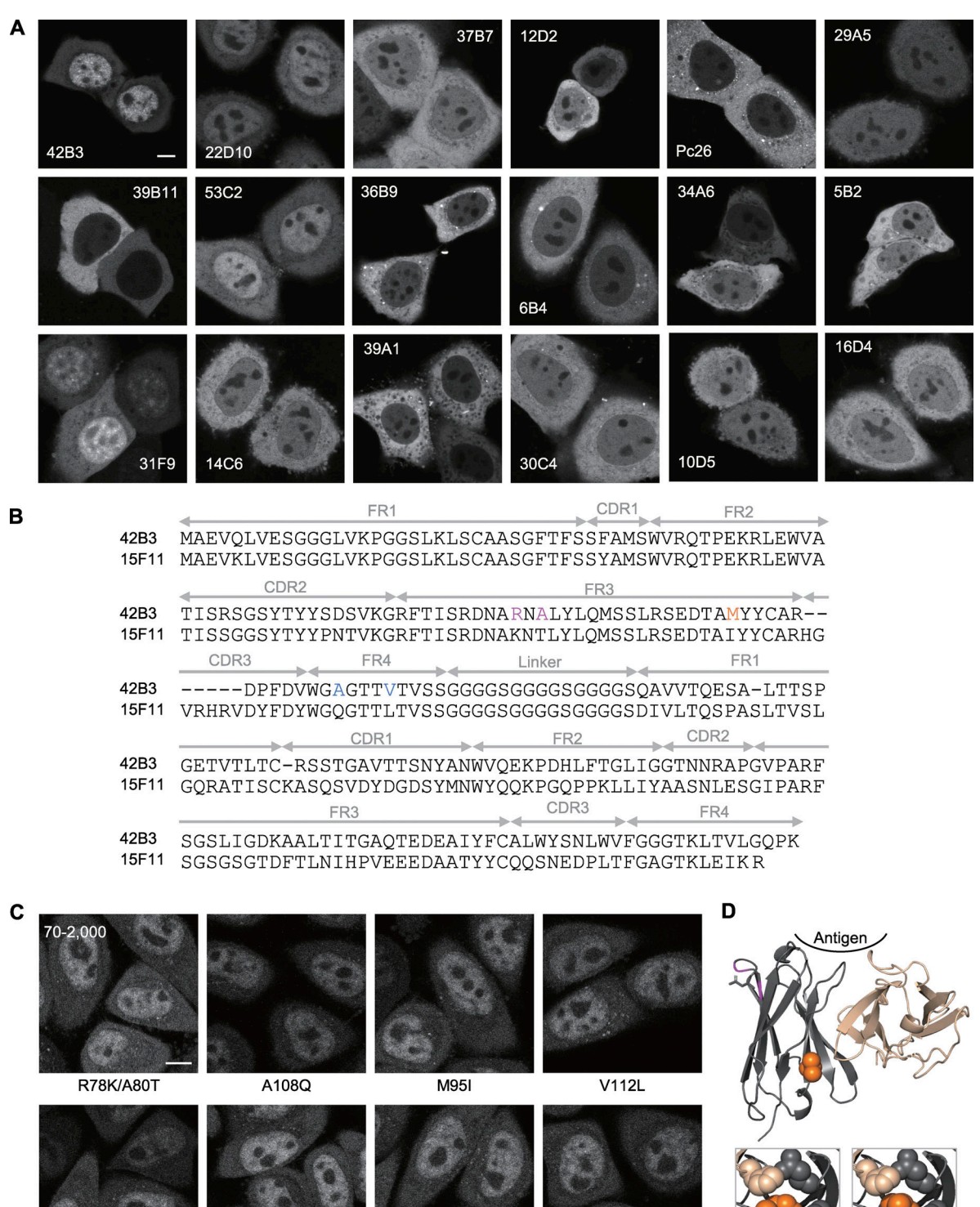

Figure S1. **Screening of RNAP2 Ser2ph-specific scFv-sfGFP and development of RNAP2 Ser2ph-mintbody. (A)** scFvs tagged with sfGFP (scFv-sfGFP) were transiently expressed in HeLa cells, and the fluorescence patterns were imaged using a confocal microscope. Single confocal sections are shown. Among the 18 different clones tested, 42B3 was most accumulated in the nucleus, where the target RNAP2 Ser2ph is present. **(B)** Amino acid sequence of RNAP2 Ser2ph (42B3) and H4K20me1 (15F11) scFvs. Framework regions (FRs), complementarity-determining regions (CDRs), and the linker region are indicated. Mutated sites in 42B3 are indicated in colors. The final construct, named "RNAP2 Ser2ph-mintbody," contains R78K, A80T, and M95V substitutions. **(C)** Example images of mutant scFv-mCherry that were stably expressed in HeLa cells. The image acquisition setting and contrast adjustment (70–2,000) are the same. **(D)** Comparison of model structures of 42B3 and 42B3(R78K/A80T/M95V). The models were generated using 15F11 (Protein Data Bank accession no. 5B3N). The M95V mutation appears to strengthen the hydrophobic core more than the original 42B3. Scale bars, 5 µm.

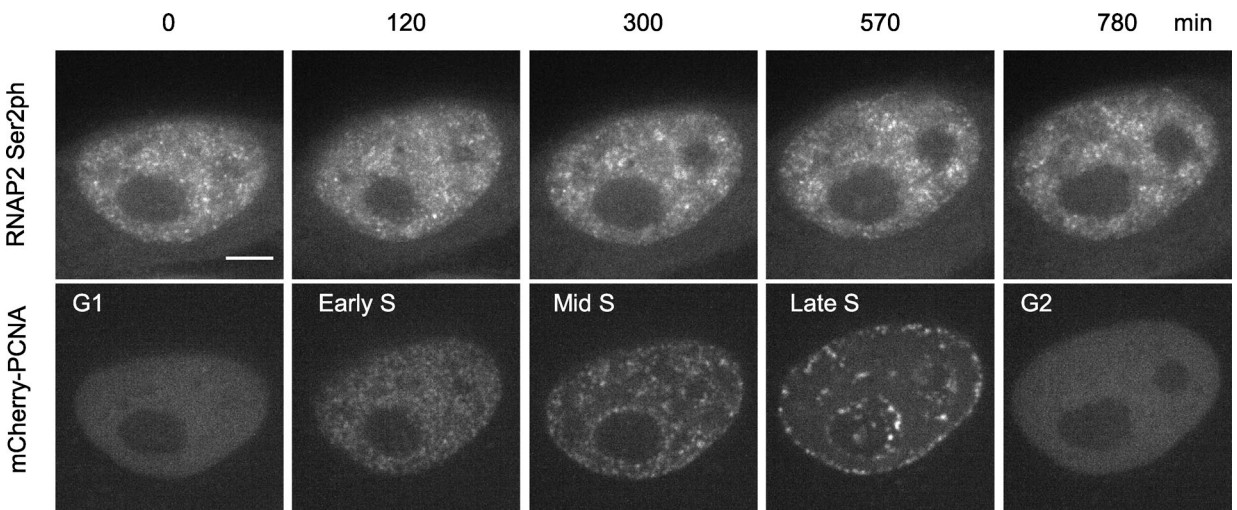

Figure S2. **PCNA dynamics during the cell cycle.** Time-lapse images of HeLa cells stably expressing RNAP2 Ser2ph-mintbody and mCherry-PCNA were acquired. Single confocal sections are shown with the elapsed time (min) and the cell cycle stage. PCNA foci scattered throughout euchromatin in the early S phase became more concentrated with fewer in number in the middle S phase and located at the nuclear periphery and around the nucleolus in the late S phase. Scale bar, 5 µm.

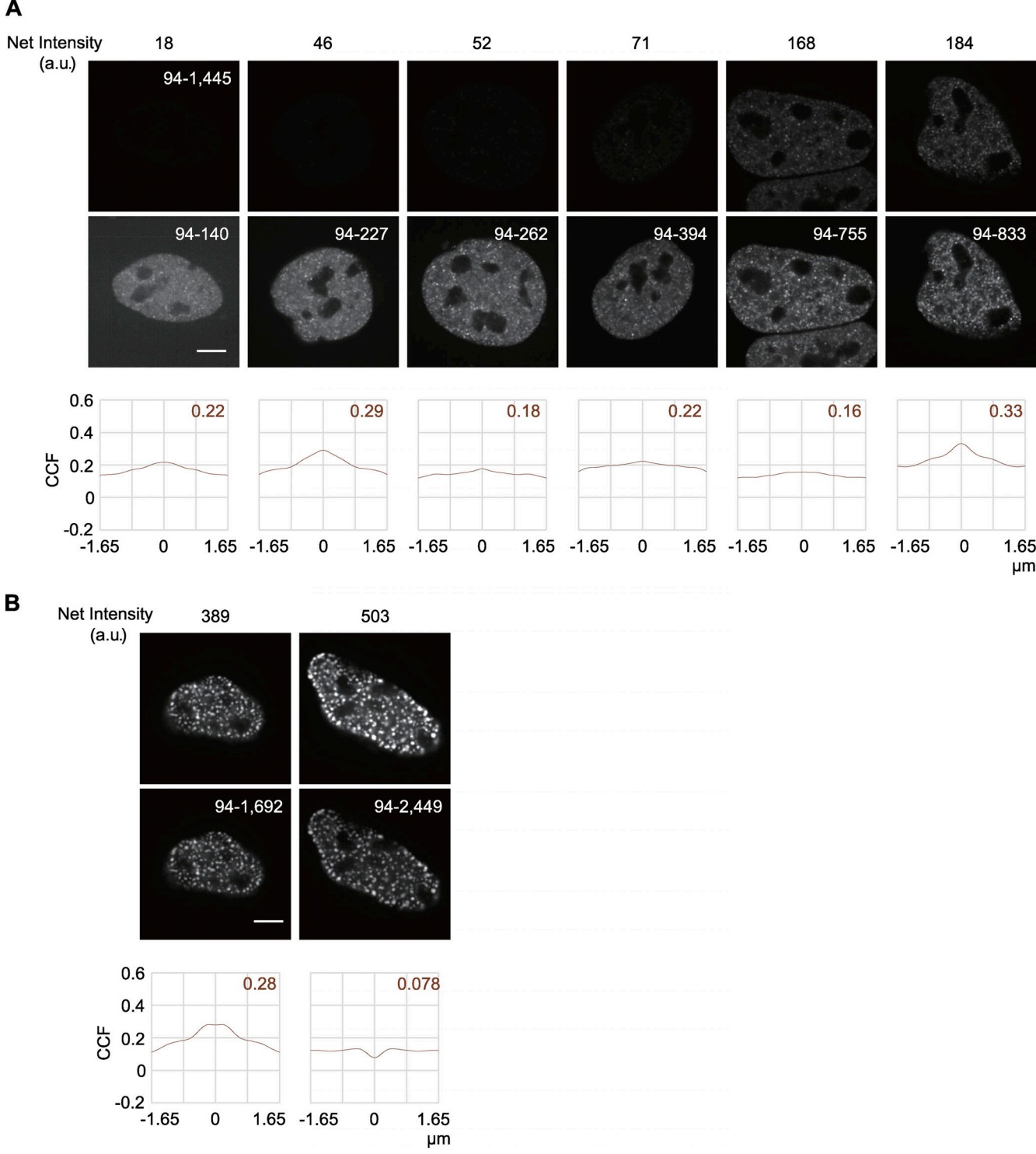

Figure S3. **Localization of Halo-p300 with different expression levels. (A and B)** Fluorescence images of Halo-p300 and RNAP2 Ser2ph-mintbody expressed in HeLa cells were acquired using a confocal microscope. Cells with low to modest (A) and high (B) expression are shown. Images of Halo-p300 are shown with fixed contrast adjusted to the highly expressed one (top; range, 94–1,445) and with individually adjusted contrast (middle; range indicated). Net nuclear fluorescence intensities, after subtracting background intensities from mean nuclear intensities, are indicated on top. The CCFs to RNAP2 Ser2ph-mintbody are shown at the bottom. Cells with larger Halo-p300 foci (B) were excluded from our analysis. Scale bars, 5 μm.

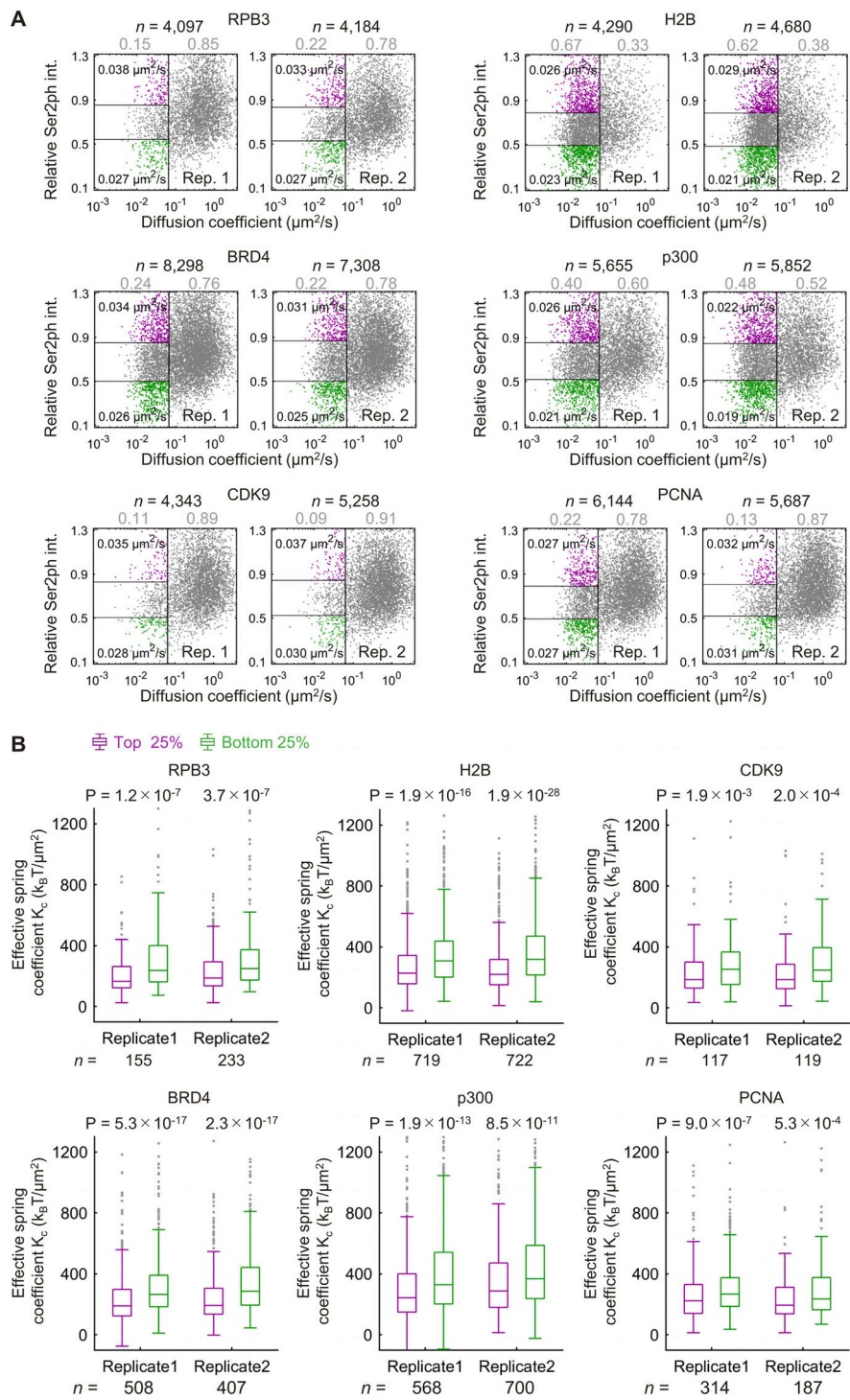

Figure S4. **Diffusion coefficient and effective spring coefficient of HaloTag-fusion proteins and RNAP2 Ser2ph-mintbody intensity. (A)** Scatter plots of the $D$ value and the relative RNAP2 Ser2ph-mintbody intensity (int.). The diffusion coefficients ($D$) of proteins were obtained using single-molecule trajectories of HaloTag-tagged proteins (RPB3, BRD4, CDK9, p300, H2B, and PCNA) stained with HaloTag TMR Ligand recorded at 33.33 ms/frame, which were superimposed upon high-pass-filtered RNAP2 Ser2ph-mintbody images in living Hela cells (Fig. 7). Scatter plots of $D$ and the relative RNAP2 Ser2ph-mintbody intensity ($I_{rel\_Ser2ph}$) from two independent experiments are shown. Each dot represents a single trajectory (molecule). A vertical line indicates $D_{thr}$. The ratio of the bound ($D \le D_{thr}$) and mobile ($D_{thr} < D$) fractions is shown in gray above the plots. The top and bottom 25% $I_{rel\_Ser2ph}$ of bound fractions are shown in magenta and green dots, respectively. The median $D$ values of bound molecules of the top and bottom 25% $I_{rel\_Ser2ph}$ are also shown. Numbers of trajectories ($n$) and the rates of bound and mobile fractions are indicated above dot plots. **(B)** The effective spring coefficient of bound HaloTag-tagged proteins, compared between the molecules that were highly associated (magenta, top 25% $I_{rel\_Ser2ph}$) and lowly associated (green, bottom 25% $I_{rel\_Ser2ph}$) with RNAP2 Ser2ph-enriched regions, with P values derived by Mann-Whitney $U$ test. Box plots from two independent replicate experiments are shown. Center lines show the medians; box limits indicate the 25th and 75th percentiles; whiskers extend 1.5 times the interquartile range from the 25th and 75th percentiles; outliers are represented by gray dots.

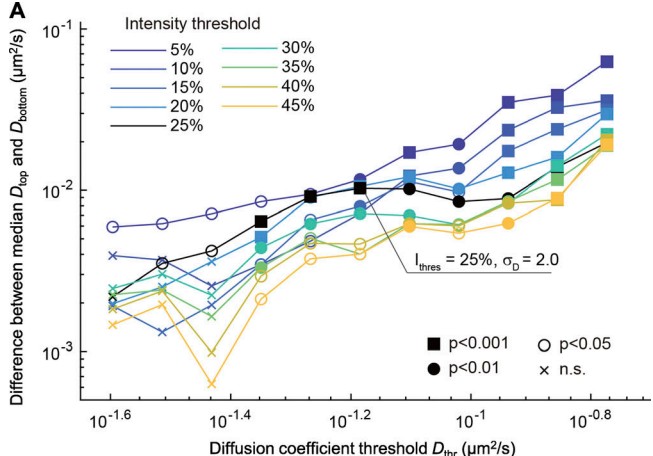

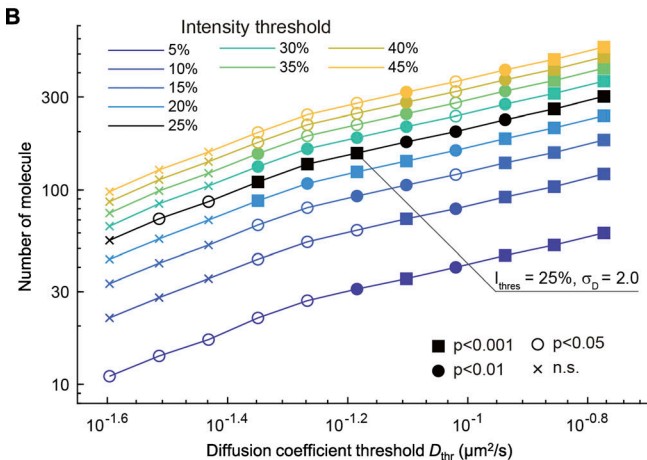

Figure S5. **Effects of thresholding parameters on the difference between median *D* values in molecules highly and lowly associated with RNAP2 Ser2ph-mintbody-enriched regions. (A and B)** The difference between median *D* values in HaloTag-tagged RPB3 molecules that were highly and lowly associated with RNAP2 Ser2ph-mintbody–enriched regions were analyzed by altering the thresholding parameters to define the fractions. The percentages of molecules to yield the top and bottom fractions of RNAP2 Ser2ph-mintbody intensity were changed from 5% to 45%; note that 25% was used for the detailed analysis. The $D_{thr}$ to define bound fraction was also changed from 0.025 to 0.169 µm²/s; note that 0.065 µm²/s was used for the detailed analysis. $\sigma_D = 2.0$ indicates $D_{thr}$ with the mean and −2 SD of the mobile fraction. After applying the thresholds, the difference between median *D* values of fractions with the top and bottom RNAP2 Ser2ph-mintbody intensity ($D_{top}$ and $D_{bottom}$, respectively; A) and the number of molecules (B) are plotted with the P values derived by Mann-Whitney *U* test and the numbers of analyzed bound molecules (*n*). In a broad range of $D_{thr}$ (0.045–0.169 µm²/s), significant differences between median $D_{top}$ and $D_{bottom}$ were observed in any percentages from 5% to 45%, with the lowest P values in the range of 20–30%.

Video 1. **RNAP2 Ser2ph-mintbody and H2B-Halo during prophase to prometaphase.** Single confocal sections of HeLa cells that stably express RNAP2 Ser2ph-mintbody (green) and H2B-Halo (magenta) were acquired every 1 min. RNAP2 Ser2ph-mintbody foci that were observed in the early prophase disappeared as the prophase progressed. RNAP2 Ser2ph-mintbody diffused into the cytoplasm after the nuclear membrane broke down. Video shows 4 frames/s.

Video 2. **RNAP2 Ser2ph-mintbody and H2B-Halo during telophase to G₁ phase.** Single confocal sections of HeLa cells that stably expressed RNAP2 Ser2ph-mintbody (green) and H2B-Halo (magenta) were acquired every 1 min. RNAP2 Ser2ph-mintbody was excluded from condensed chromosomes during the telophase but became concentrated in small foci as the nucleus was formed. The number of foci increased as the G₁ phase progressed. Video shows 4 frames/s.

**Table S1 lists the reagents and resources used in the study.**

