## [Peer Review File · The Journal of Cell Biology]

Live imaging of transcription sites using an elongating RNA polymerase II-specific probe

Satoshi Uchino, Yuma Ito, Yuko Sato, Tetsuya Handa, Yasuyuki Ohkawa, Makio Tokunaga, and Hiroshi Kimura

Corresponding Author(s): Hiroshi Kimura, Tokyo Institute of Technology

Review Timeline:

Submission Date:	2021-04-29
Editorial Decision:	2021-06-24
Revision Received:	2021-10-12
Editorial Decision:	2021-11-02
Revision Received:	2021-11-08

Monitoring Editor: Ana Pombo

Scientific Editor: Lucia Morgado-Palacin

Transaction Report:

DOI: <https://doi.org/10.1083/jcb.202104134>

June 24, 2021

Re: JCB manuscript #202104134

Prof. Hiroshi Kimura
Tokyo Institute of Technology
Cell Biology Center, Institute of Innovative Research
4259 Nagatsuta-cho
Midori-ku
Yokohama, Kanagawa 226-8503
Japan

Dear Prof. Kimura,

Thank you for submitting your manuscript entitled "Visualizing transcription sites in living cells using an elongating RNA polymerase II-specific probe". The manuscript was assessed by expert reviewers, whose comments are appended to this letter. We sincerely apologize for the delay in communicating our decision to you and thank you for your patience while waiting for the reviewers' reports. We invite you to submit a revision if you can address the reviewers' key concerns, as outlined here.

You will see that the reviewers were overall enthusiastic about your work but raised a number of concerns that will need to be addressed before the paper would be deemed appropriate for publication in JCB. While all of them found that the Ser2p mintbody is a useful system for the nuclear organization field to track elongating Pol II at endogenous genes, they requested some additional quantifications and clarifications on the parameters assessed, as well as on the logic behind some experiments. We hope that you will be able to address each of these concerns in full.

GENERAL GUIDELINES:

Text limits: Character count for an Tools is < 40,000, not including spaces. Count includes title page, abstract, introduction, results, discussion, acknowledgments, and figure legends. Count does not include materials and methods, references, tables, or supplemental legends.

Figures: Tools may have up to 10 main text figures. Figures must be prepared according to the policies outlined in our Instructions to Authors, under Data Presentation, <https://jcb.rupress.org/site/misc/ifora.xhtml>. All figures in accepted manuscripts will be screened prior to publication.

*****IMPORTANT:** It is JCB policy that if requested, original data images must be made available. Failure to provide original images upon request will result in unavoidable delays in publication. Please ensure that you have access to all original microscopy and blot data images before submitting your revision. ***

Supplemental information: There are strict limits on the allowable amount of supplemental data. Tools may have up to 5 supplemental figures. Up to 10 supplemental videos or flash animations are allowed. A summary of all supplemental material should appear at the end of the Materials and methods section.

As you may know, the typical timeframe for revisions is three to four months. However, we at JCB realize that the implementation of social distancing and shelter in place measures that limit spread of COVID-19 also pose challenges to scientific researchers. Lab closures especially are preventing scientists from conducting experiments to further their research. Therefore, JCB has waived the revision time limit. We recommend that you reach out to the editors once your lab has reopened to decide on an appropriate time frame for resubmission. Please note that papers are generally considered through only one revision cycle, so any revised manuscript will likely be either accepted or rejected.

Thank you for this interesting contribution to Journal of Cell Biology. You can contact us at the journal office with any questions, cellbio@rockefeller.edu or call (212) 327-8588.

Sincerely,

Ana Pombo
Monitoring Editor
Journal of Cell Biology

Lucia Morgado Palacin, PhD
Scientific Editor
Journal of Cell Biology

Reviewer #1 (Comments to the Authors (Required)):

The manuscript by Uchino et al. describe the development of a new genetically encoded antibody fragment (mintbody) that can probe the a form of RNA Polymerase II phosphorylated at Serine 2 of its large subunit (Ser2ph Pol II) in live cells. This is a very interesting and important development, since Ser2ph Pol II corresponds to the elongating form of Pol II, and although previous probes existed (developed by the same authors) for total and Ser5ph (the initiating form) Pol II, Ser2ph probes had not been described. A particularly exciting feature of the new Ser2ph mintbody is that it enables visualizing small foci throughout the nucleus, which presumably correspond to site of nascent Pol II transcription, of endogenous genes. Previously Ser2ph Pol II could only be tracked in an artificial gene array (Stasevich et al 2014), so the development of Ser2ph Pol II probe that can track elongating Pol II at endogenous genes is a significant advance.

The authors further validate the new Ser2ph Pol II probe with transcription inhibition experiments, and present some new results on the relative organization of Ser2ph Pol II (visualized with the mintbody) and various protein factors, as well as results on the mobility of protein factors in and out of regions of high Ser2ph, and the mobility of Ser2ph foci relative to Cy3-dUTP tagged chromatin.

Overall, the study of Uchino et al is very good and deserves publication. The text needs significant improvements to make it more accessible to a general readership, and also to clarify the rationale of some experiments and the logic behind some of the conclusions.

Please see following list of comments for specific points that could be improved.

Major points:

1. Single-particle tracking analysis.

The authors classify the trajectories as "mobile" and "bound" based on D calculated from the whole trajectory. However, in Fig. 6A-B it is clear that for a given trajectory the character of the motion might change, e.g. Fig. 6B(i) shows both confinement and larger jumps. The authors' data seem to contain much more rich information that can be extracted with more detailed analysis. What is the relationship between the regions of high Ser2ph intensity in the particle-tracking experiments (Fig. 6A-B) and the Ser2ph clusters (Fig. 2)? The authors state that in the single-particle tracking "Under the conditions used, individual RNAP2-mintbody foci were difficult to resolve, thus, mintbody-enriched regions were defined by intensity.". I think the authors want to understand the movement of single molecules inside the Ser2ph clusters. However in Fig. 6 they might only be looking at regions of the nucleus that are more accessible to the mintbody (because of chromatin compactness, presence of nuclear "bodies" or other physical effects). The relationship of the Ser2ph clusters (Fig. 2) and the regions of high Ser2ph intensity (Fig. 6) should be more accurately defined.

2. Fig. 10 "All proteins, except PCNA, showed higher D inside RNAP2 Ser2ph-mintbody-enriched regions compared to that outside, suggesting that the elongating RNAP2 and/or the molecules in the clusters were more mobile than chromatin." How did the authors reach the conclusion that elongating Pol II and/or the molecules in the clusters were more mobile than chromatin? It is not clear what the logic of this argument is, since there is not measurement of the mobility of chromatin to compare to, only measurements in and out of regions of high Ser2ph intensity.

3. The differences in D inside and outside of regions of high Ser2ph are very small (Fig. 6E). How robust are the results on the parameters selected in the analysis (e.g. Dthr, 25-75% cutoffs etc.)?

4. Motion of Ser2ph foci vs. Cy3-dUTP

The authors state "We compared the mobility of RNAP2 Ser2ph-mintbody foci and Cy3-DNA domains that showed euchromatic or heterochromatic patterns". What is "euchromatic" and "heterochromatic" patterns? I could not find any explanation of how such patterns were classified. Also, why would Ser2ph (elongating Pol II) have "heterochromatic" pattern? How big are the domains tagged with Cy3-dUTP? If Cy3-dUTP forms structures larger than diffraction limit, what does the movement of the centroid represent? What is the purpose of comparing Cy3-dUTP domains and Ser2ph foci? Is it to compare the movement of active genes (elongating Pol II) with random chromatin regions? Then the two should at least be of equal size (e.g. X kb). Another caveat is that the authors compare directly labeled DNA with mintbody tagged Pol II CTD, but the structure of the CTD and the nature of the linkage to the DNA is not well characterized. Perhaps the authors can better explain the rationale of the experiment and the possible caveats.

More minor points:

5. Pg. 4 "standard confocal microscopy cannot resolve the small foci"

- Confocal and STED super-resolution (Li et al, Cell 2019) resolved Pol II clusters in live cells and at active transcription sites.
6. Pg. 4 "High-resolution single-molecule analyses using photo-convertible or -activatable RNAP2 have indicated the transient clustering of RNAP2 during initiation (Cisse et al., 2013; Cho et al., 2016) in association with activators (Boehning et al., 2018) and mediator condensates (Cho et al., 2018)."
- Boehning et al, 2018 only looked at clustering of Dendra2-Pol II, not its association with activators.
7. Pg. 4 "The dynamics of elongating RNAP2 have yet to be documented."
- Previous works had addressed the dynamics of elongating Pol II (Li et al. Cell 2019).
8. Pg. 5 "proteins that facilitate elongation near the transcription start site, such as CDK9, BRD4, and p300."
- Can the authors provide references to previous work defining the roles of these proteins (facilitating elongation near the transcription site)? Also, Ckd9 does facilitate pause release (elongation near the transcription site), but Brd4 and p300 might have more multi-faceted roles.
9. Pg. 5 (and also abstract) "These results suggest that elongating RNAP2 foci are quite mobile compared to the (pre)initiating RNAP2 foci that constrain chromatin motion in living cells."
- Where was the mobility of (pre)initiating Pol II foci measured? Please give reference.
10. Pg. 5 "This suggested that 42B3 scFv was functionally folded and bound to RNAP2 Ser2ph in the nucleus, whereas the majority of scFvs failed to properly fold in the cytoplasm"
- Why would the scFVs not fold but the sfGFP does fold correctly? Is it possible that the scFVs do fold properly but do not bind to Pol II Ser2ph with high affinity and are thus retained in the cytoplasm. Do the scFVs have a Nuclear Localization Signal (NLS)?
11. Pg. 6 Estimation of Pol II Ser2ph foci size "The size of the foci was estimated to be ~200 nm (Fig. 2B), which is the diffraction limit of confocal microscopy, suggesting that the actual focus size is smaller than 200 nm."
- This appears to be a rather crude estimate of foci size. Can the authors perform super-resolution imaging (or perhaps 2D or 3D Structured Illumination Microscopy) to obtain a better measurement of the foci sizes?
12. Pg. 7 "depletion of RNAP2 with triptolide resulted in the rapid diminishment of nuclear foci."
- Can the authors exclude that Ser2ph foci diminishment could be because no new initiation occurs (thus no new Ser2ph is deposited)? What is the time-scale of Pol II degradation vs. Ser2ph dephosphorylation?
13. Pg. 7 "Flavopiridol also induced the diminishment of nuclear foci, but slowly, which is reasonable because elongating RNAP2 remains phosphorylated at Ser2 until it terminates."
- Is it known if Pol II Ser2ph requires continuous kinase activity for maintenance during elongation or not? For instance, has one done a Pol II Ser2ph ChIP time-lapse experiment after adding flavopiridol and does such an experiment show a wave of Ser2ph traveling down the gene?
14. Pg. 7 "RNAP2 Ser2ph-mintbody repeats binding and unbinding to RNAP2 Ser2ph in living cells."
- RNAP2 Ser2ph-mintbody repeatedly binds and unbinds to RNAP2 Ser2ph in living cells.
15. Pg. 9 "Histone H3 acetylation is involved in transcription activation through chromatin decondensation and recruiting the activators and bromodomain proteins (Stasevich et al., 2014; Cho et al., 2016; Boehning et al., 2018; Cho et al., 2018)."
- The references "Cho et al., 2016; Boehning et al., 2018; Cho et al., 2018" do not address anything about histone H3 acetylation and recruitment of Brd4! Please provide appropriate references on H3 acetylation, chromatin decondensation, recruitment of activators and bromodomain proteins.
16. Pg. 9. ""we used bulk mintbody fluorescence" - more appropriate "we used the fluorescence intensity in the mintbody images"
17. Fig. 6A,B - What kind of processing was done to the Ser2ph images? Are the raw data shown? There are white saturated pixels and clipped background (black solid area), which makes it hard to assess the distribution of Ser2ph visually. Perhaps the authors can show a grayscale image in the full dynamic range of the images, or use an alternative colormap.
18. Fig. 6D - this figure is hard to understand, particularly the solid lines. There seems to be inconsistencies with the descriptions in the figure legend. For instance "The average Ds of bound molecules of the top and bottom 25% Irel_Ser2ph are shown in magenta and green, respectively." Are the authors referring to the horizontal lines? These lines would indicate a value of Irel_Ser2ph, not D! Perhaps the figure can be made more intuitive and more details and explanations can be included in the figure legend and in the text.
19. Fig. 6D - the authors state "Trajectories of the top 25% and the bottom 25% in the Irel_Ser2ph distribution (bound) were predominantly located inside and outside RNAP2 Ser2ph-enriched regions, respectively, as shown in B."
- Can the authors show the quantification that leads to this conclusion? What is "predominantly"?
20. Pg. 10 "Taken together with the single-molecule tracking data (Fig. 6), these results suggest that the elongating RNAP2 complexes and molecules therein are not fixed to a single location in the nucleus but rather are more mobile than chromatin domains."
- I cannot understand the argument the authors try to make. The elongating Pol II complexes are presumably fixed on the gene body.
21. Pg. 12 "Therefore, it is possible that the initiating and elongating RNAP2 clusters are differently organized in space.". What is

the evidence for initiating Pol II clusters?

22. Pg. 13 "It is thus tempting to speculate that clustered RNAP2 molecules that transcribe different genes are pulled through chromatin motions from different angles, resulting in more dynamic mobility than euchromatin domains." This argument needs more clarification. Pulling at the same time from different angles would result in a net sum force closer to zero, thus movement would be smaller and mobility less dynamic.

Reviewer #2 (Comments to the Authors (Required)):

This is an enterprising and much desired resource for the nuclear organisation field, with strong implications for broader fields of biology such as development, tissue homeostasis, regeneration. The ability to look at global Pol2 activity in living cells opens many doors for understanding the relationships between signalling and biosynthesis, in these and other contexts. To the best of my understanding, the work is very well performed, with some beautiful imaging and the work is very clearly presented.

Specific comments:

1. "Because the folding of scFv is affected by the fusion partner protein or short peptide (Liu et al., 2019), it is likely that sfGFP, but not mCherry, assisted in 42B3 scFv folding."

A small issue, but I would actually suggest that Cherry is causing the problems here, not the sfGFP solving them. GFP tagged proteins are often much better behaved than Cherry ones, because there is still an innate self-assembly feature of the red proteins, despite all the engineering they underwent.

2. "In cells that started chromosome condensation at the onset of prophase, RNAP2 Ser2ph-mintbody foci were observed around the edge or outside the condensing chromosomes"

Fig 2D is very attractive and well presented, but I wouldn't summarise the distribution of S2P like this without some form of quantification. It is not a distribution the eye sees immediately, if at all, although there seems to be some sort of pattern here. Perhaps show some example parallel line scans of the different fluorescent channels.

3. Fig 3b vertical axis- this needs to be more specific. Can you use a "real value" eg, the cytoplasmic level.

4. Figure 7- is the replication labelling a fair comparison with the S2P labelling? If each spot is a transcribing gene (based on a few thousand spots, this seems reasonable), then would a single operator array insertion be a more realistic comparison? The literature has numbers on this you can almost certainly use, in many different systems- these do have a range of values, depending on locus/cell type etc- so perhaps it would be better to use these data as an additional benchmark to the mobility.

Reviewer #3 (Comments to the Authors (Required)):

In this manuscript, Uchino et al. develop genetically encoded antibodies for observing RNAP2 phosphorylation status. Because phosphorylation of the RNAP2 CTD correlates with different stages of the transcription cycle (initiation, elongation, etc.), the ability to dynamically visualize this post-translational modification in living cells at single genes would be a considerable breakthrough. The authors isolated single chain variable fragments from hybridomas which were then mutagenized to optimize stability and nuclear compartmentalization. The resulting 'mintbody' binds transiently to targets and reveals the dynamics of CTD phosphorylation. The authors don't quite get to this single-gene limit - which was my hope when I first read the abstract -- but they do study RNAP2 Ser2p across the cell cycle and look at dynamics of the transcription machinery on and off Ser2p foci. Having a genetically-encoded probe is a substantial technical advance. That said, the main achievement is development of a new promising reagent and not a convincing biological finding per se. Overall, the images are high quality, and the analysis is thoughtful. My compliments to the authors on a nice study.

Comments:

1. One of the main biological experiments is single-molecule tracking of H2B, RPB3, CDK9, BRD4, followed by MSD analysis, and binning of the resulting diffusion coefficients. The finding that most proteins involved in gene regulation showed higher mobility in the Ser2p foci is one finding the authors pursue further by measuring the mobility of these foci directly. I must admit I find the results a little underwhelming. Looking at Fig. 6D, it is not clear to me there is a relationship between the extent of foci formation (represented by 'Relative Ser2ph intensity' vs. diffusion coefficient), nor do I see separation between populations with differing diffusion coefficients.

Similarly, the conclusion that elongating complexes are more mobile than chromatin domains perhaps indicates the absence of stable transcription 'factories?' This area has been a contentious one in the literature, and the authors summarize some competing views in the discussion. However, I found the results in Fig. 6 and 7 somewhat untethered from recent developments in the field.

2. Similarly, using Halo tagging for the correlation measurements strikes me as problematic since there will always be a dark fraction of un-labeled molecules. I suspect that at some point, someone will have to show that this reagent shows a ChIP profile which is consistent with Ser2p for the mintbody to gain broad acceptance. However, that experiment is not necessary for

publication in JCB.

3. These parts require clarification:

"ELISA plates were coated with RNAP2 CTD peptides that harbor phosphorylation at different Ser residues." How were the CTD peptides prepared/ validated?

"Directional movements of RNAP2 Ser2ph-mintbody foci were hardly observed during this period." In the larger context of this paragraph, it is a confusing statement. Does it mean the authors were somehow expecting to see RNAP2 moving directionally on a template? Translocating to a transcription factory? Have such things been seen before? I just need a little more to go on. I'm assuming Fig. 7 fits are anomalous diffusion exponents? That statement needs to be visible in the text somewhere.

1st Revision - Authors' Response to Reviewers: October 12, 2021

Point-by-point response to reviewers' comments:

In the revised version, we added more data and reorganized figures to follow the JCB guideline (up to 10 figures and up to 5 supplementary materials). The change from the original to the revised version are listed below.

Original	Revised	New contents
Fig. 1	Fig. 1 (A-C)	
Fig. 2 (A-C)	Fig. 1 (D-F)	
Fig. 2 (D-F)	Fig. 2 (A-C)	
Fig. 3	Fig. 3	
Fig. 4	Fig. 4	
Fig. 5	Fig. 5	
Fig. S3	Fig. 6	
Fig. 6 (A-D)	Fig. 7 (B-D, G)	
	Fig. 7 (A, E, F, H) New	Details for the single molecule analysis
Fig. 6 (E)	Fig. 8 (A)	
	Fig. 8 (B) New	Areas of confinement by single molecule tracking
	Fig. 9 (A-C) New	Mobility of Ser2ph-mintbody and PCNA foci
Fig. 7	Fig. 9 (D-F)	
	Fig. 10 New	Mobility of Ser2ph-mintbody and p300 foci
Fig. S1	Fig. S1 (A)	
Fig. S2	Fig. S1 (B)	
	Fig. S2 New	Dynamics of PCNA foci
	Fig. S3 New	p300 localization with different expression levels
Fig. S4	Fig. S4 (A)	
	Fig. S4 (B) New	Effective spring coefficients by single molecule tracking
	Fig. S5 New	Robustness of the difference in median D_s

The reviewers' comments are indicated in blue and the changed parts in the text are marked in red.

Reviewer #1

The manuscript by Uchino et al. describe the development of a new genetically encoded antibody fragment (mintbody) that can probe the a form of RNA Polymerase II phosphorylated at Serine 2 of its large subunit (Ser2ph Pol II) in live cells. This is a very interesting and important development, since Ser2ph Pol II corresponds to the elongating form of Pol II, and although previous probes existed (developed by the same authors) for total and Ser5ph (the initiating form) Pol II, Ser2ph probes had not been described. A particularly exciting feature of the new Ser2ph mintbody is that it enables visualizing small foci throughout the nucleus, which presumably correspond to site of nascent Pol II transcription, of endogenous genes. Previously Ser2ph Pol II could only be tracked in an artificial gene array (Stasevich et al 2014), so the development of Ser2ph Pol II probe that can track elongating Pol II at endogenous genes is a significant advance.

The authors further validate the new Ser2ph Pol II probe with transcription inhibition experiments, and present some new results on the relative organization of Ser2ph Pol II (visualized with the mintbody) and various protein factors, as well as results on the mobility of protein factors in and out of regions of high Ser2ph, and the mobility of Ser2ph foci relative to Cy3-dUTP tagged chromatin.

Overall, the study of Uchino et al is very good and deserves publication. The text needs significant improvements to make it more accessible to a general readership, and also to clarify the rationale of some experiments and the logic behind some of the conclusions.

Please see following list of comments for specific points that could be improved.

We appreciate your positive and constructive comments.

Major points:

1. Single-particle tracking analysis.

The authors classify the trajectories as "mobile" and "bound" based on D calculated from the whole trajectory. However, in Fig. 6A-B it is clear that for a given trajectory the character of the motion might change, e.g. Fig. 6B(i) shows both confinement and larger jumps. The authors' data seem to contain much more rich information that can be extracted with more detailed analysis. What is the relationship between the regions of high Ser2ph intensity in the particle-tracking experiments (Fig. 6A-B) and the Ser2ph clusters (Fig. 2)? The authors state that in the single-particle tracking "Under the conditions used, individual RNAP2-mintbody foci were difficult to resolve, thus, mintbody-enriched regions were defined by intensity.". I think the authors want to understand the movement of single molecules inside the Ser2ph clusters. However in Fig. 6 they might only be looking at regions of the nucleus that are more accessible to the mintbody (because of chromatin compactness, presence of nuclear "bodies" or other physical effects). The relationship of the Ser2ph clusters (Fig. 2) and the regions of high Ser2ph intensity (Fig. 6) should be more accurately defined.

Thank you for the comment. In the revised version, we now show the original images by HILO microscopy and the filtered images that we used to define RNAP2 Ser2ph-mintbody-enriched regions (Fig. 7A-7C). Because our use of "inside" and "outside" of RNAP2 Ser2ph-mintbody-enriched regions sounded like a molecule being "inside" or "outside" throughout the entire observation period, we now instead use "highly associated" and "lowly associated" to more accurately reflect our data. We have also now analyzed the properties of molecules that are highly and lowly associated with RNAP2 Ser2ph-mintbody-enriched regions in more detail, including the frequency of the association during the tracked duration (Fig. 7E, F, and H), the area of confinement (Fig. 8B), and the effective spring coefficient (Fig. S4B). The results are not so surprising; when the diffusion coefficient was larger, the area of confinement was larger and the effective spring coefficient was smaller. We now describe these results:

p10, the third paragraph

"Single HaloTag-tagged proteins were simultaneously imaged with RNAP2 Ser2ph-mintbody **fluorescence** at 33.33 ms per frame for 16.67 s (Fig. 7A, B). The diffusion coefficients (D s) of the individual molecules were determined by **linear fitting to the first six steps (~200 ms) of Mean Squared Displacement (MSD)** (Fig. 7C). The histograms of D represent the bimodal distribution of slow and fast fractions (Fig. 7D). As most H2B-Halo molecules were classified into slow fractions **by setting a threshold of D_{thr} 0.065 $\mu\text{m}^2/\text{s}$** (Fig. 7D), this fraction represented the molecules bound to chromatin and/or other large structures. We called this the bound fraction. Relatively large proportions of BRD4 and p300 were also found in the bound fraction compared to RPB3 and

CDK9 (Fig. 7D and S4A). This probably reflects the small fraction of total cellular RNAP2 that are elongating at any given time (Kimura et al., 2002) and the transient interactions of CDK9 with RNAP2.

We then analyzed the mobility of bound molecules that were highly and lowly associated with RNAP2 Ser2ph-mintbody-enriched regions. As an example, the molecule tracked in (i) in Fig. 7C was located most (88%) of the time in an area of the nucleus that was enriched with RNAP2 Ser2ph-mintbody (Fig. 7E, top). By contrast, the molecule tracked in (ii) was located exclusively in an area of the nucleus that was not enriched with RNAP2 Ser2ph-mintbody (Fig. 7E, bottom). Molecules that went in and out of RNAP2 Ser2ph-mintbody-enriched regions were also observed (Fig. 7F). After plotting D and the underlying RNAP2 Ser2ph-mintbody intensity from each tracked molecule (Fig. 7G, top; and S4A), the bound fraction as determined above (Fig. 7D) was subjected to further analyses (Fig. 7G, bottom). Based on the RNAP2 Ser2ph-mintbody intensity, the top 25% and bottom 25% were classified as fractions that were highly and lowly associated with RNAP2 Ser2ph-mintbody-enriched regions (Fig. 7G, bottom; and S4A). Molecules in the top 25% and bottom 25% stayed in local RNAP2 Ser2ph-mintbody-enriched regions for on average 88% and 5% of the trajectory durations, respectively (Fig. 7H). When compared the D s in the top (magenta) and bottom (green) fractions of bound molecules, all proteins, except PCNA, had higher D s in the top 25% highly associated fraction with RNAP2 Ser2ph-mintbody-enriched regions compared to D in the bottom 25% fraction (Fig. 8A). Consistent with the higher D s, molecules in the top 25% showed larger areas of confinement and lower effective spring coefficients than the bottom 25% (Fig. 8B and S4B). Significant differences between median D s of the top and bottom fractions were robustly observed over a wide range of definitions for the top and bottom fractions based on RNAP2 Ser2ph-mintbody intensity (from 5% to 45%) and also when different values of D_{thr} were used to define the bound fraction (from 0.045 to 0.169 $\mu\text{m}^2/\text{s}$) (Fig. S5). These data suggest that the bound molecules that are highly associated with RNAP2 Ser2ph-enriched regions are more mobile than lowly associated molecules.”

2. Fig. 6 "All proteins, except PCNA, showed higher D inside RNAP2 Ser2ph-mintbody-enriched regions compared to that outside, suggesting that the elongating RNAP2 and/or the molecules in the clusters were more mobile than chromatin." How did the authors reach the conclusion that elongating Pol II and/or the molecules in the clusters were more mobile than chromatin? It is not clear what the logic of this argument is, since there is not measurement of the mobility of chromatin to compare to, only measurements in and out of regions of high Ser2ph intensity.

Thank you for this comment. We agree that from this experiment we do not know if the molecules outside the foci are associated with chromatin. We have therefore carefully rewritten the paragraph so it is no longer misleading.

p11, the first paragraph

“When compared the D s in the top (magenta) and bottom (green) fractions of bound molecules, all proteins, except PCNA, had higher D s in the top 25% highly associated fraction with RNAP2 Ser2ph-mintbody-enriched regions compared to D in the bottom 25% fraction (Fig. 8A). ... These data suggest that the bound molecules that are highly associated with RNAP2 Ser2ph-enriched regions are more mobile than lowly associated molecules.”

3. The differences in D inside and outside of regions of high Ser2ph are very small (Fig. 6E). How robust are the results on the parameters selected in the analysis (e.g. D_{thr} , 25-75% cutoffs etc.)?

We have now analyzed the differences in D (D_{top} vs D_{bottom}) using different values for the diffusion coefficient threshold (D_{thr}) that defines bound molecules (Fig. S5). Over a broad range of D_{thr} , significant differences were observed. Thus, we believe our results are quite robust with respect to changing the definition of the top and bottom fractions and D_{thr} .

p11, the first paragraph

“Significant differences between median D s of the top and bottom fractions were robustly observed over a wide range of definitions for the top and bottom fractions based on RNAP2 Ser2ph-mintbody intensity (from 5% to 45%) and also when different values of D_{thr} were used to define the bound fraction (from 0.045 to 0.169 $\mu\text{m}^2/\text{s}$) (Fig. S5).”

4. Motion of Ser2ph foci vs. Cy3-dUTP

The authors state "We compared the mobility of RNAP2 Ser2ph-mintbody foci and Cy3-DNA domains that showed euchromatic or heterochromatic patterns". What is "euchromatic" and "heterochromatic" patterns? I could not find any explanation of how such patterns were classified. Also, why would Ser2ph (elongating Pol II) have "heterochromatic" pattern? How big are the domains tagged with Cy3-dUTP? If Cy3-dUTP forms structures larger than diffraction limit, what does the movement of the centroid represent? What is the purpose of comparing Cy3-dUTP domains and Ser2ph foci? Is it to compare the movement of active genes (elongating Pol II) with random chromatin regions? Then the two should at least be of equal size (e.g. X kb). Another caveat is that the authors compare directly labeled DNA with mintbody tagged Pol II CTD, but the structure of the CTD and the nature of the linkage to the DNA is not well characterized. Perhaps the authors can better explain the rationale of the experiment and the possible caveats.

We have now analyzed the mobility of Halo-PCNA and Halo-p300 foci. The rationale of the Halo-PCNA experiment is to compare the mobility of transcription elongation (Ser2ph) foci and replication (PCNA) foci. Replication foci are known to contain clustered replicons. These replicons, which share the same replication timing and can be pulse-labeled with Cy3-dUTP, can form a domain that is stably inherited over cell generations. The mobility of Cy3-incorporated replication domains has been shown to behave similarly to single nucleosomes within the domains (Nozaki et al., 2017), which we now cite. Therefore, the movement of the centroid of PCNA and Cy3-DNA represents the mobility of the chromatin domains.

To clarify things, we have made several changes. First, to better illustrate the euchromatic and heterochromatic patterns we used for classification, we have now added images showing the dynamic change in the distribution of Halo-PCNA, which moves from euchromatin to heterochromatin during the S phase (Fig. S2). Second, we describe how these patterns were used to classify tracks of Halo-PCNA foci into euchromatin (early S pattern) or heterochromatin (late S pattern), respectively (Fig. 9A-C). Third, we describe how the same classification scheme was used for Cy3-labeled DNA, which also marks replication domains like PCNA foci. The Cy3-DNA mobility was confirmatory of Halo-PCNA data (Fig. 9D-F). Fourth, we analyzed more data. This changed the figures slightly, but the results are essentially the same.

Regarding Ser2ph foci in heterochromatic regions, we apologize for the confusion. What we meant to say is Ser2ph in cells showing heterochromatic Cy3-DNA patterns. To clarify, we combined graphs to show the mobility of Ser2ph, Cy3-euchromatin, and Cy3-heterochromatin (Fig. 9F).

Finally, we analyzed the mobility of p300-bound chromatin to compare the mobility of elongation foci (Ser2ph) and enhancers/promoters (Fig. 10).

p11, the second paragraph

"To further investigate the dynamics of the transcription elongation sites over several seconds, we analyzed the mobility of RNAP2 Ser2ph-mintbody foci by tracking their center of mass in each focus with 551 ms imaging intervals using a confocal microscope. The mobility of RNAP2 Ser2ph-mintbody foci was first compared with replication foci marked by Halo-PCNA (Fig. 9A-C). PCNA foci appear throughout euchromatin in early S and then later redistribute to heterochromatin at the nuclear periphery and in late S (Fig. S2) (Leonhardt et al., 2000). RNAP2 Ser2ph-mintbody foci exhibited modestly constrained diffusional motion, with a D of $0.0029 \mu\text{m}^2/\text{s}$ and an anomalous exponent (α) of 0.69 (such that the $\text{MSD} = 4Dt^\alpha$). This mobility ($\alpha < 1$ and $D \sim 10^{-3} \mu\text{m}^2/\text{s}$) is consistent with other measurements of chromatin motion in mammalian cells ($D \sim 10^{-4} - 10^{-2} \mu\text{m}^2/\text{s}$ depending on the gene locus and methods; Levi et al., 2005; Chen et al., 2013; Lucas et al., 2014; Germier et al., 2017; Gu et al., 2018; Ma et al., 2019). The anomalous exponents appeared constant during the observation period (Fig. 9C, bottom). Compared with RNAP2 Ser2ph-mintbody foci, both euchromatic and heterochromatic PCNA foci were less mobile (D $0.0023 \mu\text{m}^2/\text{s}$, α 0.69; and D $0.0009 \mu\text{m}^2/\text{s}$, α 0.64, respectively) (Fig. 9A-C). This result suggests that over a period of several seconds, transcription elongation foci are more mobile than replication foci in which many replication forks in the same chromatin domains are clustered (Jackson and Pombo, 1998).

To compare the mobility of RNAP2 Ser2ph-mintbody foci with chromatin domains that share the same replication timing, we labeled replication domains by DNA replication-mediated Cy3-dUTP incorporation (Manders et al., 1999; Nozaki et al., 2017). The Cy3-labeled replication domains have been shown to behave similarly to individual nucleosomes in the same domains (Nozaki et al., 2017). HeLa cells expressing RNAP2 Ser2ph-mintbody were loaded with Cy3-dUTP to pulse-label replicated chromatin and were further grown for two days. As a limited amount of Cy3-dUTP was loaded into cells, DNA regions that were replicated just after

the loading were labeled and then Cy3 signals exhibited characteristic DNA replication foci depending on the stage in S-phase (Manders et al., 1999). Once incorporated, Cy3 on DNA persisted after cell divisions, so the replication domains could be tracked in living cells. We classified Cy3-DNA patterns into the early and late replication domains based on the number and intranuclear distribution of Cy3-DNA foci, similar to what we did for Halo-PCNA (Fig. 5, second column; 9A, B). If there were many foci in nuclear interior regions, then we classified them as early, euchromatic domains (Fig. 9D). If there were fewer foci at the nuclear periphery and around nucleoli, then we classified them as late, heterochromatic domains (Fig. 9E). Consistent with Halo-PCNA and nucleosome mobility (Nozaki et al., 2017; Shaban et al., 2020), euchromatic domains were more mobile than heterochromatic domains, and RNAP2 Ser2ph-mintbody foci were further mobile than euchromatic domains (Fig. 9C).

Finally, we measured the mobility of Halo-p300-enriched foci, which are likely to be associated with enhancers (Heintzman et al., 2009; Visel et al., 2009), together with RNAP2 Ser2ph-mintbody foci. Again, RNAP2 Ser2ph-mintbody foci were more mobile compared to p300-enriched foci, whose mobility was lower than typical euchromatic replication foci, or chromatin domains (Fig. 10). Taken together with the single-molecule tracking data (Fig. 7 and 8), these results suggest that the elongating RNAP2 complexes and molecules therein can move together with DNA templates that are more mobile than chromatin replication domains and p300-enriched foci.”

More minor points:

5. Pg. 4 "standard confocal microscopy cannot resolve the small foci"

Confocal and STED super-resolution (Li et al, Cell 2019) resolved Pol II clusters in live cells and at active transcription sites.

Thank you for pointing this out. We have updated the text to include a more proper description along with the suggested citation.

p4, the second paragraph

“As more than 100,000 RNAP2 molecules are present in a cell (Kimura et al., 1999; Stasevich et al., 2014), it has been difficult to resolve the small foci by standard confocal microscopy (Sugaya et al., 2000; Imada et al., 2021), but recent confocal and 3D stimulated emission depletion (3D-STED) microscopy has enabled detecting single elongating RNAP2 foci in living cells (Li et al., 2019).”

6. Pg. 4 "High-resolution single-molecule analyses using photo-convertible or -activatable RNAP2 have indicated the transient clustering of RNAP2 during initiation (Cisse et al., 2013; Cho et al., 2016) in association with activators (Boehning et al., 2018) and mediator condensates (Cho et al., 2018)."

Boehning et al, 2018 only looked at clustering of Dendra2-Pol II, not its association with activators.

Thank you. We corrected.

p4, the second paragraph

“High-resolution single-molecule analyses using photo-convertible or -activatable RNAP2 have indicated the transient clustering of RNAP2 during initiation (Cisse et al., 2013; Cho et al., 2016; Boehning et al., 2018) in association with mediator condensates (Cho et al., 2018).”

7. Pg. 4 "The dynamics of elongating RNAP2 have yet to be documented."

Previous works had addressed the dynamics of elongating Pol II (Li et al. Cell 2019).

We rephrased the text and cited the literature.

p4, the second paragraph

“The dynamics of elongating RNAP2 have not been well documented, except RNAP2 foci on some specific genes including *Nanog* in mouse embryonic stem cells (Li et al., 2019).”

8. Pg. 5 "proteins that facilitate elongation near the transcription start site, such as CDK9, BRD4, and p300."

Can the authors provide references to previous work defining the roles of these proteins (facilitating elongation near the transcription site)? Also, Cdk9 does facilitate pause release (elongation near the transcription site), but Brd4 and p300 might have more multi-faceted roles.

We rephrased the text and cited the literature.

p5, the first paragraph

“Using high-resolution microscopy, we analyzed the relative localization of RNAP2 Ser2ph-mintbody with proteins involved in RNAP2 phosphorylation, elongation, and transcription activation, such as CDK9, CDK12, a Paf1 complex component LEO1, splicing factor SRRF1/SF2/ASF, bromodomain containing protein 4 (BRD4) (Dey et al., 2003), and p300 histone acetyltransferase (Heintzman et al., 2009; Visel et al., 2009). RNAP2 Ser2ph showed more colocalization with CDK12 and LEO1 than CDK9, which facilitates elongation near the transcription start site (Price, 2000), and enhancer-associated proteins that can also facilitate elongation, such as BRD4 and p300 (Zhang et al., 2012; Li et al., 2019; Hsu et al., 2021).”

9. Pg. 5 (and also abstract) "These results suggest that elongating RNAP2 foci are quite mobile compared to the (pre)initiating RNAP2 foci that constrain chromatin motion in living cells."

Where was the mobility of (pre)initiating Pol II foci measured? Please give reference.

We rephrase the text so as not to mention (pre)initiating RNAP2. The previous sentence was also changed to reflect the new data using PCNA and p300.

p5, the first paragraph

“RNAP2 Ser2ph foci as such were also more mobile than both euchromatic and heterochromatic DNA replication foci and p300-enriched foci. These results suggest that elongating RNAP2 foci are quite mobile compared to typical euchromatin and heterochromatin domains.”

p2, Abstract

“RNAP2 Ser2ph-mintbody foci showed constrained diffusional motion like chromatin, but was more mobile compared to DNA replication foci and p300-enriched foci, suggesting that the elongating RNAP2 complexes are separated from the more confined chromatin domains.”

10. Pg. 5 "This suggested that 42B3 scFv was functionally folded and bound to RNAP2 Ser2ph in the nucleus, whereas the majority of scFvs failed to properly fold in the cytoplasm"

Why would the scFVs not fold but the sfGFP does fold correctly?

Is it possible that the scFvs do fold properly but do not bind to Pol II Ser2ph with high affinity and are thus retained in the cytoplasm. Do the scFvs have a Nuclear Localization Signal (NLS)?

Sorry for the confusion. We rephrased the text, including the possibility of low binding affinity and mentioning that the scFv-sfGFP does not have an NLS.

p5, the second paragraph

“As the scFv-sfGFP does not harbor a nuclear localization signal, this suggested that 42B3 scFv was functionally folded and bound to RNAP2 Ser2ph in the nucleus, whereas the other scFvs had a much lower antigen-binding affinity or failed to properly fold in the cytoplasm (Wörn et al., 2001; Sato et al., 2013; Sato et al., 2016; Zhao et al., 2019).”

11. Pg. 6 Estimation of Pol II Ser2ph foci size "The size of the foci was estimated to be ~200 nm (Fig. 2B), which is the diffraction limit of confocal microscopy, suggesting that the actual focus size is smaller than 200 nm."

This appears to be a rather crude estimate of foci size. Can the authors perform super-resolution imaging (or perhaps 2D or 3D Structured Illumination Microscopy) to obtain a better measurement of the foci sizes?

To better estimate the focus size, RNAP2 Ser2ph-mintbody images were acquired using a TauSTED superresolution microscope (Leica Microsystems). It was, however, still difficult to determine the actual size

because the signal was noisy and the results depended on the parameters to define mintbody foci (See Figure for referees below). As we had a limited access to the STED microscope in another institution, it was not possible to optimize the acquisition condition; in fact, none of the original authors were allowed to enter the microscope facility room under the COVID19 pandemic situation. We would like to leave the crude estimate as is, and leave more precise measurements for a future study.

Figure for referees. RNAP2 Ser2ph-mintbody in HeLa cells analyzed by STED microscopy. HeLa cells expressing RNAP2 Ser2ph-mintbody (Venus-tagged version) were imaged using a TauSTED microscope (Leica Microsystems; Stellaris; gate 0.5; 18 nm/pixel). A typical fluorescence image and a magnified view of an indicated area are shown in (A). Fluorescence spots in the nucleus were detected using NIS Elements (Nikon). The following image processing analysis were performed using MATLAB. A 21 x 21 pixel square containing a spot was cropped, the intensity was normalized, and the averaged image of all spots from a single was generated (B). By fitting to 2D Gaussian distribution (C), the full width at half maximum (FWHM) was obtained. By using different parameters for typical diameter and contrast to detect spots, different number of spots were detected. FWHM with > 100 detected spots in the nucleus are plotted in (D), showing the range from ~70 to ~170 nm depending on the parameters. Setting the typical diameter pixels to 4 or less was not reliable because of noise.

12. Pg. 7 "depletion of RNAP2 with triptolide resulted in the rapid diminishment of nuclear foci." Can the authors exclude that Ser2ph foci diminishment could be because no new initiation occurs (thus no new Ser2ph is deposited)? What is the time-scale of Pol II degradation vs. Ser2ph dephosphorylation?

Please see the response to 13 just below.

13. Pg. 7 "Flavopiridol also induced the diminishment of nuclear foci, but slowly, which is reasonable because elongating RNAP2 remains phosphorylated at Ser2 until it terminates."

Is it known if Pol II Ser2ph requires continuous kinase activity for maintenance during elongation or not? For instance, has one done a Pol II Ser2ph ChIP time-lapse experiment after adding flavopiridol and does such an experiment show a wave of Ser2ph traveling down the gene?

As triptolide inhibits initiation (Titov et al., 2011; Jonkers et al., 2014), we changed the text not to exclude its effect on inhibition. We also cited a reference that showed a wave of transcripts and Ser2ph ChIP-seq signals towards 3' end after treatment with flavopiridol (Jonkers et al., 2014) and DRB, which also inhibit P-TEFb (Lavigne et al., 2017), respectively.

p7, the second paragraph

"The number of RNAP2 Ser2ph-mintbody foci was **not decreased** in cells treated with DMSO for 2 h. In contrast, the inhibition of the transcription initiation and depletion of RNAP2 with 5 μM triptolide (Forero-Quintero et al., 2021) resulted in the rapid diminishment of nuclear foci. Flavopiridol also induced the diminishment of nuclear foci, but more mildly because only the productive elongation of newly initiated RNAP2 is inhibited, leaving elongating RNAP2 complexes until their termination (Jonkers et al., 2014); RNAP2 Ser2ph in elongating complexes has also been shown to remain in DRB treatment, which also inhibit P-TEFb (Lavigne et al., 2017)."

14. Pg. 7 "RNAP2 Ser2ph-mintbody repeats binding and unbinding to RNAP2 Ser2ph in living cells." RNAP2

Ser2ph-mintbody repeatedly binds and unbinds to RNAP2 Ser2ph in living cells.

Thank you for the suggestion, which we have now included.

p8, the first paragraph

“Thus, kinetic analysis supported the view that RNAP2 Ser2ph-mintbody repeatedly binds and unbinds to RNAP2 Ser2ph in living cells.”

15. Pg. 9 "Histone H3 acetylation is involved in transcription activation through chromatin decondensation and recruiting the activators and bromodomain proteins (Stasevich et al., 2014; Cho et al., 2016; Boehning et al., 2018; Cho et al., 2018)."

The references "Cho et al., 2016; Boehning et al., 2018; Cho et al., 2018" do not address anything about histone H3 acetylation and recruitment of Brd4! Please provide appropriate references on H3 acetylation, chromatin decondensation, recruitment of activators and bromodomain proteins.

We cited appropriate references in this and the following sentences to introduce these proteins.

p9, the third paragraph

“Histone H3 acetylation is involved in transcription activation through chromatin decondensation and recruiting the co-activators and bromodomain proteins (Dhalluin et al., 1999; Tóth et al., 2004; Stasevich et al., 2014).”

16. Pg. 9. ""we used bulk mintbody fluorescence" - more appropriate "we used the fluorescence intensity in the mintbody images"

We have changed accordingly.

p10, the second paragraph

“Instead, we used the fluorescence intensity in the mintbody images to define RNAP2 Ser2ph-enriched regions using a much lower laser power than that used in the single-molecule analysis.”

17. Fig. 6A,B - What kind of processing was done to the Ser2ph images? Are the raw data shown? There are white saturated pixels and clipped background (black solid area), which makes it hard to assess the distribution of Ser2ph visually. Perhaps the authors can show a grayscale image in the full dynamic range of the images, or use an alternative colormap.

We have made new Fig. 7A to show the details of image processing, including the raw data. We also explained the processing in the main text.

p10, the second paragraph

“Under the conditions used, individual RNAP2-mintbody foci were difficult to resolve. Thus, mintbody-enriched regions were defined by areas with locally high intensity after high-pass filtration (Fig. 7A, B).”

18. Fig. 6D - this figure is hard to understand, particularly the solid lines. There seems to be inconsistencies with the descriptions in the figure legend. For instance "The average Ds of bound molecules of the top and bottom 25% Irel_Ser2ph are shown in magenta and green, respectively." Are the authors referring to the horizontal lines? These lines would indicate a value of Irel_Ser2ph, not D! Perhaps the figure can be made more intuitive and more details and explanations can be included in the figure legend and in the text.

Sorry for the confusion. We have now rearranged the figures to be more easily understandable (Fig. 7D-7G) and explained the details in the main text. Please find the text in the response to point 19 below

19. Fig. 6D - the authors state "Trajectories of the top 25% and the bottom 25% in the Irel_Ser2ph distribution (bound) were predominantly located inside and outside RNAP2 Ser2ph-enriched regions, respectively, as shown in B."

Can the authors show the quantification that leads to this conclusion?

We now show the frequency of molecular association with RNAP2 Ser2ph-mintbody-enriched regions (Fig. 7E and 7F). The top 25% and bottom 25% showed respectively higher and lower frequencies of association with Ser2ph regions (Fig 7H).

p10, the fourth paragraph

“We then analyzed the mobility of bound molecules that were highly and lowly associated with RNAP2 Ser2ph-mintbody-enriched regions. As an example, the molecule tracked in (i) in Fig. 7C was located most (88%) of the time in an area of the nucleus that was enriched with RNAP2 Ser2ph-mintbody (Fig. 7E, top). By contrast, the molecule tracked in (ii) was located exclusively in an area of the nucleus that was not enriched with RNAP2 Ser2ph-mintbody (Fig. 7E, bottom). Molecules that went in and out of RNAP2 Ser2ph-mintbody-enriched regions were also observed (Fig. 7F). After plotting D and the underlying RNAP2 Ser2ph-mintbody intensity from each tracked molecule (Fig. 7G, top; and S4A), the bound fraction as determined above (Fig. 7D) was subjected to further analyses (Fig. 7G, bottom). Based on the RNAP2 Ser2ph-mintbody intensity, the top 25% and bottom 25% were classified as fractions that were highly and lowly associated with RNAP2 Ser2ph-mintbody-enriched regions (Fig. 7G, bottom; and S4A). Molecules in the top 25% and bottom 25% stayed in local RNAP2 Ser2ph-mintbody-enriched regions for on average 88% and 5% of the trajectory durations, respectively (Fig. 7H).”

20. Pg. 10 "Taken together with the single-molecule tracking data (Fig. 6), these results suggest that the elongating RNAP2 complexes and molecules therein are not fixed to a single location in the nucleus but rather are more mobile than chromatin domains."

I cannot understand the argument the authors try to make. The elongating Pol II complexes are presumably fixed on the gene body.

The elongating RNAP2 complexes are fixed to the template DNA indeed, but the RNAP2-bound DNA can be moved. To avoid the confusion, we rephrased.

p12, the third paragraph

“Taken together with the single-molecule tracking data (Fig. 7 and 8), these results suggest that the elongating RNAP2 complexes and molecules therein **can move together with DNA templates that** are more mobile than chromatin **replication domains and p300-enriched foci.**”

21. Pg. 12 "Therefore, it is possible that the initiating and elongating RNAP2 clusters are differently organized in space.". What is the evidence for initiating Pol II clusters?

We added a reason for the possibility and a future work to test this. We also changed “clusters” to “complexes”.

p14, the second paragraph

“From the distinct localization and dynamics of the factors involved in transcription initiation from RNAP2 Ser2ph, it is possible that initiating and elongating RNAP2 complexes are organized differently in space. We anticipate that RNAP2 Ser2ph-mintbody will help address the question of whether or not new initiation events occurs at or distal to preexisting elongation RNAP2 foci.”

22. Pg. 13 "It is thus tempting to speculate that clustered RNAP2 molecules that transcribe different genes are pulled through chromatin motions from different angles, resulting in more dynamic mobility than euchromatin domains." This argument needs more clarification. Pulling at the same time from different angles would result in a net sum force closer to zero, thus movement would be smaller and mobility less dynamic.

Thank you. We added “at different time points” to make the point clearer.

p15, the first paragraph

“It is thus tempting to speculate that clustered RNAP2 molecules that transcribe different genes are pulled by chromatin motions from different angles **at different time points**, resulting in more dynamic mobility than euchromatin domains.”

Reviewer #2

This is an enterprising and much desired resource for the nuclear organisation field, with strong implications for broader fields of biology such as development, tissue homeostasis, regeneration. The ability to look at global Pol2 activity in living cells opens many doors for understanding the relationships between signalling and biosynthesis, in these and other contexts. To the best of my understanding, the work is very well performed, with some beautiful imaging and the work is very clearly presented.

We appreciate your positive and constructive comments.

1. "Because the folding of scFv is affected by the fusion partner protein or short peptide (Liu et al., 2019), it is likely that sfGFP, but not mCherry, assisted in 42B3 scFv folding."

A small issue, but I would actually suggest that Cherry is causing the problems here, not the sfGFP solving them. GFP tagged proteins are often much better behaved than Cherry ones, because there is still an innate self-assembly feature of the red proteins, despite all the engineering they underwent.

Thank you for this comment. We rephrased the text to consider the effects of sfGFP and mCherry in either way, by citing a reference for the aggregation issue of mCherry-fusion proteins (Landgraf et al., 2012).

p5, the second paragraph

"Because the folding of scFv is affected by the fusion partner protein or short peptide (Kabayama et al., 2020), 42B3 scFv folding can be assisted by sfGFP and/or be disturbed by mCherry, which could cause cytoplasmic aggregations of fusion proteins (Landgraf et al., 2012). Thus, 42B3 scFv did not appear to have a particularly stable framework, unlike another mintbodies specific for histone H4 Lys20 monomethylation (H4K20me1), which is functional even when fused with mCherry and whose framework has been used to generate a stable chimeric scFv by implanting the complementary determining regions from a different antibody (Sato et al., 2016; Zhao et al., 2019; Liu et al., 2021)."

2. "In cells that started chromosome condensation at the onset of prophase, RNAP2 Ser2ph-mintbody foci were observed around the edge or outside the condensing chromosomes"

Fig 2D is very attractive and well presented, but I wouldn't summarise the distribution of S2P like this without some form of quantification. It is not a distribution the eye sees immediately, if at all, although there seems to be some sort of pattern here. Perhaps show some example parallel line scans of the different fluorescent channels.

We rearranged the figure (Fig. 2A) to show magnified views with line scans to clearly see the relationship between RNAP2 Ser2ph-mintbody foci and condensing chromosomes.

p7, the first paragraph

"In cells that started chromosome condensation at the onset of prophase, RNAP2 Ser2ph-mintbody foci were observed around the edge or outside the condensing chromosomes (Fig. 2A, 0 and 1 min; magnified views and line scan profiles), which is consistent with the results of a previous study using RNA in situ hybridization to detect nascent transcripts (Liang et al., 2015). The number of foci decreased as the prophase progressed (Fig. 2A, 2 and 3 min) and almost disappeared in the late prophase (Fig. 2A, 4 min; Movie 1)."

3. Fig 3b vertical axis- this needs to be more specific. Can you use a "real value" eg, the cytoplasmic level.

In the original manuscript we plotted the area occupied by foci. In the revised figure (Fig. 3B), we have now plotted the number of foci relative to the starting time point, which should be easier to understand. The results are very similar to the previous data.

p7, the second paragraph

"Time-lapse images were collected using a confocal microscope and the numbers of the RNAP2 Ser2ph-mintbody foci in single sections were measured (Fig. 3A, B). The number of RNAP2 Ser2ph-mintbody foci was not decreased in cells treated with DMSO for 2 h. In contrast, the inhibition of the transcription initiation and depletion of RNAP2 with 5 μ M triptolide (Forero-Quintero et al., 2021) resulted in the rapid diminishment of

nuclear foci. Flavopiridol also induced the diminishment of nuclear foci, but more mildly because only the productive elongation of newly initiated RNAP2 is inhibited, leaving elongating RNAP2 complexes until their termination (Jonkers et al., 2014)”

4. Figure 7- is the replication labelling a fair comparison with the S2P labelling? If each spot is a transcribing gene (based on a few thousand spots, this seems reasonable), then would a single operator array insertion be a more realistic comparison? The literature has numbers on this you can almost certainly use, in many different systems- these do have a range of values, depending on locus/cell type etc- so perhaps it would be better to use these data as an additional benchmark to the mobility.

Thank you for this comment. There was a huge range of diffusion coefficients reported by different systems, although the diffusion coefficients obtained here were well fit in the range, and so it was difficult to compare the actual numbers between different studies. We now compared the mobility of transcription elongation (Ser2ph) foci with replication (PCNA) foci (Fig. 9A) and p300-enriched foci (Fig. 10), and described more in detail with citations of papers reporting chromatin mobility in a similar time scale.

p11, the second paragraph

“To further investigate the dynamics of the transcription elongation sites over several seconds, we analyzed the mobility of RNAP2 Ser2ph-mintbody foci by tracking their center of mass in each focus with 551 ms imaging intervals using a confocal microscope. The mobility of RNAP2 Ser2ph-mintbody foci was first compared with replication foci marked by Halo-PCNA (Fig. 9A-C). PCNA foci appear throughout euchromatin in early S and then later redistribute to heterochromatin at the nuclear periphery and in late S (Fig. S2) (Leonhardt et al., 2000). RNAP2 Ser2ph-mintbody foci exhibited modestly constrained diffusional motion, with a D of $0.0029 \mu\text{m}^2/\text{s}$ and an anomalous exponent (α) of 0.69 (such that the $\text{MSD} = 4D\alpha t$). This mobility ($\alpha < 1$ and $D \sim 10^{-3} \mu\text{m}^2/\text{s}$) is consistent with other measurements of chromatin motion in mammalian cells ($D \sim 10^{-4} - 10^{-2} \mu\text{m}^2/\text{s}$ depending on the gene locus and methods; Levi et al., 2005; Chen et al., 2013; Lucas et al., 2014; Germier et al., 2017; Gu et al., 2018; Ma et al., 2019). The anomalous exponents appeared constant during the observation period (Fig. 9C, bottom). Compared with RNAP2 Ser2ph-mintbody foci, both euchromatic and heterochromatic PCNA foci were less mobile (D $0.0023 \mu\text{m}^2/\text{s}$, α 0.69; and D $0.0009 \mu\text{m}^2/\text{s}$, α 0.64, respectively) (Fig. 9A-C). This result suggests that over a period of several seconds, transcription elongation foci are more mobile than replication foci in which many replication forks in the same chromatin domains are clustered (Jackson and Pombo, 1998).

To compare the mobility of RNAP2 Ser2ph-mintbody foci with chromatin domains that share the same replication timing, we labeled replication domains by DNA replication-mediated Cy3-dUTP incorporation (Manders et al., 1999; Nozaki et al., 2017). The Cy3-labeled replication domains have been shown to behave similarly to individual nucleosomes in the same domains (Nozaki et al., 2017). HeLa cells expressing RNAP2 Ser2ph-mintbody were loaded with Cy3-dUTP to pulse-label replicated chromatin and were further grown for two days. As a limited amount of Cy3-dUTP was loaded into cells, DNA regions that were replicated just after the loading were labeled and then Cy3 signals exhibited characteristic DNA replication foci depending on the stage in S-phase (Manders et al., 1999). Once incorporated, Cy3 on DNA persisted after cell divisions, so the replication domains could be tracked in living cells. We classified Cy3-DNA patterns into the early and late replication domains based on the number and intranuclear distribution of Cy3-DNA foci, similar to what we did for Halo-PCNA (Fig. 5, second column; 9A, B). If there were many foci in nuclear interior regions, then we classified them as early, euchromatic domains (Fig. 9D). If there were fewer foci at the nuclear periphery and around nucleoli, then we classified them as late, heterochromatic domains (Fig. 9E). Consistent with Halo-PCNA and nucleosome mobility (Nozaki et al., 2017; Shaban et al., 2020), euchromatic domains were more mobile than heterochromatic domains, and RNAP2 Ser2ph-mintbody foci were further mobile than euchromatic domains (Fig. 9C).

Finally, we measured the mobility of Halo-p300-enriched foci, which are likely to be associated with enhancers (Heintzman et al., 2009; Visel et al., 2009), together with RNAP2 Ser2ph-mintbody foci. Again, RNAP2 Ser2ph-mintbody foci were more mobile compared to p300-enriched foci, whose mobility was lower than typical euchromatic replication foci, or chromatin domains (Fig. 10). Taken together with the single-molecule tracking data (Fig. 7 and 8), these results suggest that the elongating RNAP2 complexes and molecules therein can move together with DNA templates that are more mobile than chromatin replication domains and p300-enriched foci.”

Reviewer #3

In this manuscript, Uchino et al. develop genetically encoded antibodies for observing RNAP2 phosphorylation status. Because phosphorylation of the RNAP2 CTD correlates with different stages of the transcription cycle (initiation, elongation, etc.), the ability to dynamically visualize this post-translational modification in living cells at single genes would be a considerable breakthrough. The authors isolated single chain variable fragments from hybridomas which were then mutagenized to optimize stability and nuclear compartmentalization. The resulting 'mintbody' binds transiently to targets and reveals the dynamics of CTD phosphorylation. The authors don't quite get to this single-gene limit - which was my hope when I first read the abstract -- but they do study RNAP2 Ser2p across the cell cycle and look at dynamics of the transcription machinery on and off Ser2p foci. Having a genetically-encoded probe is a substantial technical advance. That said, the main achievement is development of a new promising reagent and not a convincing biological finding per se. Overall, the images are high quality, and the analysis is thoughtful. My compliments to the authors on a nice study.

We appreciate your positive and constructive comments.

1. One of the main biological experiments is single-molecule tracking of H2B, RPB3, CDK9, BRD4, followed by MSD analysis, and binning of the resulting diffusion coefficients. The finding that most proteins involved in gene regulation showed higher mobility in the Ser2p foci is one finding the authors pursue further by measuring the mobility of these foci directly. I must admit I find the results a little underwhelming. Looking at Fig. 6D, it is not clear to me there is a relationship between the extent of foci formation (represented by 'Relative Ser2ph intensity' vs. diffusion coefficient), nor do I see separation between populations with differing diffusion coefficients.

Thank you for this comment. We performed more rigorous analyses for the tracking data by changing the threshold of RNAP2 Ser2ph-mintbody intensity and diffusion coefficient, and confirmed that the conclusions were quite robust (Fig. S5). We describe the analytical procedure and the validation in more detail with new figure panels (Fig. 7).

p10, the second paragraph

“We next analyzed the mobility of single protein molecules that were highly and lowly associated with RNAP2 Ser2ph-mintbody-enriched regions, by labeling HaloTag-tagged proteins with a low concentration of the HaloTag TMR ligand using a highly inclined and laminated optical sheet (HILO) microscope (Tokunaga et al., 2008; Lim et al., 2018). As the individual RNAP2 Ser2ph-mintbody molecules bound to the target in less than a few seconds, we did not perform single-molecule analysis of the mintbody. Instead, we used the fluorescence intensity in the mintbody images to define RNAP2 Ser2ph-enriched regions using a much lower laser power than that used in the single-molecule analysis. Under the conditions used, individual RNAP2-mintbody foci were difficult to resolve. Thus, mintbody-enriched regions were defined by areas with locally high intensity after high-pass filtration (Fig. 7A, B).

Single HaloTag-tagged proteins were simultaneously imaged with RNAP2 Ser2ph-mintbody fluorescence at 33.33 ms per frame for 16.67 s (Fig. 7A, B). The diffusion coefficients (D s) of the individual molecules were determined by linear fitting to the first six steps (~200 ms) of Mean Squared Displacement (MSD) (Fig. 7C). The histograms of D represent the bimodal distribution of slow and fast fractions (Fig. 7D). As most H2B-Halo molecules were classified into slow fractions by setting a threshold of $D_{thr} 0.065 \mu\text{m}^2/\text{s}$ (Fig. 7D), this fraction represented the molecules bound to chromatin and/or other large structures. We called this the bound fraction. Relatively large proportions of BRD4 and p300 were also found in the bound fraction compared to RPB3 and CDK9 (Fig. 7D and S4A). This probably reflects the small fraction of total cellular RNAP2 that are elongating at any given time (Kimura et al., 2002) and the transient interactions of CDK9 with RNAP2.

We then analyzed the mobility of bound molecules that were highly and lowly associated with RNAP2 Ser2ph-mintbody-enriched regions. As an example, the molecule tracked in (i) in Fig. 7C was located most (88%) of the time in an area of the nucleus that was enriched with RNAP2 Ser2ph-mintbody (Fig. 7E, top). By contrast, the molecule tracked in (ii) was located exclusively in an area of the nucleus that was not enriched with RNAP2 Ser2ph-mintbody (Fig. 7E, bottom). Molecules that went in and out of RNAP2 Ser2ph-mintbody-enriched regions were also observed (Fig. 7F). After plotting D and the underlying RNAP2 Ser2ph-mintbody intensity from each tracked molecule (Fig. 7G, top; and S4A), the bound fraction as determined above (Fig. 7D)

was subjected to further analyses (Fig. 7G, bottom). Based on the RNAP2 Ser2ph-mintbody intensity, the top 25% and bottom 25% were classified as fractions that were highly and lowly associated with RNAP2 Ser2ph-mintbody-enriched regions (Fig. 7G, bottom; and S4A). Molecules in the top 25% and bottom 25% stayed in local RNAP2 Ser2ph-mintbody-enriched regions for on average 88% and 5% of the trajectory durations, respectively (Fig. 7H). When compared the D s in the top (magenta) and bottom (green) fractions of bound molecules, all proteins, except PCNA, had higher D s in the top 25% highly associated fraction with RNAP2 Ser2ph-mintbody-enriched regions compared to D in the bottom 25% fraction (Fig. 8A). Consistent with the higher D s, molecules in the top 25% showed larger areas of confinement and lower effective spring coefficients than the bottom 25% (Fig. 8B and S4B). Significant differences between median D s of the top and bottom fractions were robustly observed over a wide range of definitions for the top and bottom fractions based on RNAP2 Ser2ph-mintbody intensity (from 5% to 45%) and also when different values of D_{thr} were used to define the bound fraction (from 0.045 to 0.169 $\mu\text{m}^2/\text{s}$) (Fig. S5). These data suggest that the bound molecules that are highly associated with RNAP2 Ser2ph-enriched regions are more mobile than lowly associated molecules.”

Similarly, the conclusion that elongating complexes are more mobile than chromatin domains perhaps indicates the absence of stable transcription 'factories?' This area has been a contentious one in the literature, and the authors summarize some competing views in the discussion. However, I found the results in Fig. 6 and 7 somewhat untethered from recent developments in the field.

There are some contradictory observations for RNAP2 organization and chromatin mobility in living cells, and integrating all the data into a single model has been extremely difficult. We discuss a plausible explanation of our findings as much as possible to fit with the literature. We now added a question that can be addressed by using the RNAP2 Ser2ph-mintbody.

p14, the second paragraph

“We observed that protein molecules associated with RNAP2 Ser2ph-rich regions were more mobile than those outside the foci. Similarly, RNAP2 Ser2ph foci were more mobile than replication foci and chromatin domains. This observation is consistent with the results of studies that showed the higher mobility of transcribed loci (Gu et al., 2018) and decreased RNAP2 mobility with a CDK9 inhibitor (Shaban et al., 2020) but not with those of other studies that showed that RNAP2 transcription constrains chromatin motion (Ochiai et al., 2015; Germier et al., 2017; Nagashima et al., 2019). Single-molecule analyses have also revealed that transcription constrains chromatin motion (Nagashima et al., 2019; Shaban et al., 2020). The lower mobility of chromatin is still observed in cells treated with CDK9 inhibitors (Germier et al., 2017; Nagashima et al., 2019), implying that transcription-dependent chromatin constraint is mediated through transcription initiation rather than elongation (Babokhov et al., 2020). As only a small fraction of RNAP2 Ser2ph foci was associated with CDK9 (also demonstrated by Ghamari et al., 2013), most mintbody foci corresponded to RNAP2 Ser2ph that were already elongating on the gene body but not at the transcription start sites. In addition, BRD4 foci were even further apart from RNAP2 Ser2ph or transcripts (Li et al., 2019; Li et al., 2020). From the distinct localization and dynamics of the factors involved in transcription initiation from RNAP2 Ser2ph, it is possible that initiating and elongating RNAP2 complexes are organized differently in space. We anticipate that RNAP2 Ser2ph-mintbody will help address the question of whether or not new initiation events occurs at or distal to preexisting elongation RNAP2 foci.”

2. Similarly, using Halo tagging for the correlation measurements strikes me as problematic since there will always be a dark fraction of un-labeled molecules. I suspect that at some point, someone will have to show that this reagent shows a CHIP profile which is consistent with Ser2p for the mintbody to gain broad acceptance. However, that experiment is not necessary for publication in JCB.

We now mention that we did not use highly expressed cells. For Halo-p300, we analyzed cells with different expression levels (Fig. S3). The profiles (CCF) did not appear to correlate with the expression levels in cells with low to moderate intensity. The highest expression lead to the formation of bigger foci, but we did not use those cells. We are planning to perform ChIP-seq using the mintbody, but we have not achieved this yet. Thank you for your understanding.

p20, the fourth paragraph

“Highly expressed cells, which were brightly labeled even with the low ligand concentration, were not used for the analysis.”

p10, the first paragraph

“Halo-p300 exhibited a number of small foci with a few bright condensates, as seen when monomeric enhanced GFP was knocked into p300 locus (Ma et al., 2021). Although p300 could form larger condensates when highly expressed (McManus and Hendzel, 2003) (Fig. S3B), we only analyzed cells in which Halo-p300 was expressed at much lower levels than those with large condensates (Fig. S3A). In these cells with low to moderate expression, CCF profiles of Halo-p300 respect to RNAP Ser2ph-mintbody did not depend on the expression levels (Fig. S3A).”

3. These parts require clarification:

"ELISA plates were coated with RNAP2 CTD peptides that harbor phosphorylation at different Ser residues." How were the CTD peptides prepared/ validated?

We purchased the BSA-conjugated peptides from a company (as described in Materials and Methods). Ser5ph peptide was validated by its reactivity with anti-Ser5ph antibody. This point is now mentioned.

p8, the second paragraph

“Ser5ph-containing peptides were reacted with CMA603, assuring that the Ser5ph peptides that were not reacted with RNAP2 Ser2ph-mintbody were properly coated on the plates.”

"Directional movements of RNAP2 Ser2ph-mintbody foci were hardly observed during this period." In the larger context of this paragraph, it is a confusing statement. Does it mean the authors were somehow expecting to see RNAP2 moving directionally on a template? Translocating to a transcription factory? Have such things been seen before? I just need a little more to go on.

Thank you for this comment. To avoid confusion, we removed this sentence. We initially thought someone might expect tracking RNAP2 moving linearly along with DNA, but everyone knows that DNA in the nucleus is packed non-linearly in the nucleus.

I'm assuming Fig. 7 fits are anomalous diffusion exponents? That statement needs to be visible in the text somewhere.

Indeed, an anomalous diffusion model was used to fit the MSD data. We now described the mobility of RNAP2 Ser2ph-mintbody foci in more detail, with the diffusion coefficient and anomalous exponent.

p11, the second paragraph

“To further investigate the dynamics of the transcription elongation sites over several seconds, we analyzed the mobility of RNAP2 Ser2ph-mintbody foci by tracking their center of mass in each focus with 551 ms imaging intervals using a confocal microscope. The mobility of RNAP2 Ser2ph-mintbody foci was first compared with replication foci marked by Halo-PCNA (Fig. 9A-C). PCNA foci appear throughout euchromatin in early S and then later redistribute to heterochromatin at the nuclear periphery and in late S (Fig. S2) (Leonhardt et al., 2000). RNAP2 Ser2ph-mintbody foci exhibited modestly constrained diffusional motion, with a D of $0.0029 \mu\text{m}^2/\text{s}$ and an anomalous exponent (α) of 0.69 (such that the $\text{MSD} = 4Dt^\alpha$). This mobility ($\alpha < 1$ and $D \sim 10^{-3} \mu\text{m}^2/\text{s}$) is consistent with other measurements of chromatin motion in mammalian cells ($D \sim 10^{-4} - 10^{-2} \mu\text{m}^2/\text{s}$ depending on the gene locus and methods; Levi et al., 2005; Chen et al., 2013; Lucas et al., 2014; Germier et al., 2017; Gu et al., 2018; Ma et al., 2019). The anomalous exponents appeared constant during the observation period (Fig. 9C, bottom). Compared with RNAP2 Ser2ph-mintbody foci, both euchromatic and heterochromatic PCNA foci were less mobile (D $0.0023 \mu\text{m}^2/\text{s}$, α 0.69; and D $0.0009 \mu\text{m}^2/\text{s}$, α 0.64, respectively) (Fig. 9A-C). This result suggests that over a period of several seconds, transcription elongation foci are more mobile than replication foci in which many replication forks in the same chromatin domains are clustered (Jackson and Pombo, 1998).”

November 2, 2021

RE: JCB Manuscript #202104134R

Prof. Hiroshi Kimura
Tokyo Institute of Technology
Cell Biology Center, Institute of Innovative Research
4259 Nagatsuta-cho
Midori-ku
Yokohama, Kanagawa 226-8503
Japan

Dear Prof. Kimura:

Thank you for submitting your revised manuscript entitled "Visualizing transcription sites in living cells using an elongating RNA polymerase II-specific probe". We have now assessed your revised manuscript and we would be happy to publish your paper in JCB pending revisions to address some minor final text edits indicated below and those necessary to meet our formatting guidelines (see details below).

Final minor edits:

- p. 6 "one CTD harbours 52 Thy1-Ser2-... repeats". It would be more precise to say "52 heptapeptide repeats", as not all heptapeptide repeats have the listed canonical aminoacid sequence.
- There is a small typo in page 14 ('occurs at' instead of 'occur at')
- Panel 9C is mentioned after 9D and 9E, so the authors could consider reorganizing the figure or relabeling the panels, or otherwise reorganizing the text slightly.
- p16. Chery instead of Cherry
- p18. Florescence recovery instead of Fluorescence rec..
- p20. Highly expressed cells should be H.. expressing cells
- Fig 4A, shows kD instead of kDa

A. MANUSCRIPT ORGANIZATION AND FORMATTING:

Full guidelines are available on our Instructions for Authors page, <https://jcb.rupress.org/submission-guidelines#revised>.
Submission of a paper that does not conform to JCB guidelines will delay the acceptance of your manuscript.

1) Text limits: Character count for Articles and Tools is < 40,000, not including spaces. Count includes title page, abstract, introduction, results, discussion, and acknowledgments. Count does not include materials and methods, figure legends, references, tables, or supplemental legends.

2) Figures limits: Articles and Tools may have up to 10 main text figures.

3) Figure formatting:

Molecular weight or nucleic acid size markers must be included on all gel electrophoresis. Scale bars must be present on all microscopy images, including inset magnifications. Also, please avoid pairing red and green for images and graphs to ensure legibility for color-blind readers.

4) Statistical analysis:

Error bars on graphic representations of numerical data must be clearly described in the figure legend. The number of independent data points (n) represented in a graph must be indicated in the legend. Statistical methods should be explained in full in the materials and methods. For figures presenting pooled data the statistical measure should be defined in the figure legends. Please also be sure to indicate the statistical tests used in each of your experiments (both in the figure legend itself and in a separate methods section) as well as the parameters of the test (for example, if you ran a t-test, please indicate if it was one- or two-sided, etc.). If you use parametric tests in your study (i.e. t-tests), you should have first determined whether the data was normally distributed before selecting that test. In the stats section of the methods, please indicate how you tested for normality. If you did not test for normality, you must state something to the effect that "Data distribution was assumed to be normal but this was not formally tested."

5) Abstract and title: The abstract should be no longer than 160 words and should communicate the significance of the paper for

a general audience. The title should be less than 100 characters including spaces. Make the title concise but accessible to a general readership. **Although your title is fine, we would like to suggest the following "Live imaging of transcription sites using an elongating RNA polymerase II-specific probe" as we think it is a bit more clear.**

6) Materials and methods:

Should be comprehensive and not simply reference a previous publication for details on how an experiment was performed. Please provide full descriptions (at least in brief) in the text for readers who may not have access to referenced manuscripts. The text should not refer to methods "...as previously described."

7) Please be sure to provide the sequences for all of your primers/oligos and RNAi constructs in the materials and methods. You must also indicate in the methods the source, species, and catalog numbers (where appropriate) for all of your antibodies.

8) Microscope image acquisition: The following information must be provided about the acquisition and processing of images:

- a. Make and model of microscope
- b. Type, magnification, and numerical aperture of the objective lenses
- c. Temperature
- d. imaging medium
- e. Fluorochromes
- f. Camera make and model
- g. Acquisition software
- h. Any software used for image processing subsequent to data acquisition. Please include details and types of operations involved (e.g., type of deconvolution, 3D reconstitutions, surface or volume rendering, gamma adjustments, etc.).

10) Supplemental materials: There are strict limits on the allowable amount of supplemental data. Articles/Tools may have up to 5 supplemental figures. A summary of all supplemental material (that is in addition to the supplementary figure/table legends) should appear at the end of the Materials and methods section. We would need a bit more of detail accompanying the description of the supplementary material.

11) eTOC summary: A ~40-50 word summary that describes the context and significance of the findings for a general readership should be included on the title page. The statement should be written in the present tense and refer to the work in the third person. *** It should begin with "First author name(s) et al..." to match our preferred style.

12) Conflict of interest statement:

JCB requires inclusion of a statement in the acknowledgements regarding competing financial interests. If no competing financial interests exist, please include the following statement: "The authors declare no competing financial interests." If competing interests are declared, please follow your statement of these competing interests with the following statement: "The authors declare no further competing financial interests."

13) A separate author contribution section is required following the Acknowledgments in all research manuscripts. **All authors should be mentioned and designated by their first and middle initials and full surnames.** We encourage use of the CRediT nomenclature (<https://casrai.org/credit/>).

14) ORCID IDs: ORCID IDs are unique identifiers allowing researchers to create a record of their various scholarly contributions in a single place. At resubmission of your final files, please consider providing an ORCID ID for as many contributing authors as possible.

15) Materials and data sharing: All datasets included in the manuscript must be available from the date of online publication, and the source code for all custom computational methods, apart from commercial software programs, must be made available either in a publicly available database or as supplemental materials hosted on the journal website. Numerous resources exist for data storage and sharing (see Data Deposition: <https://rupress.org/jcb/pages/data-deposition>), and you should choose the most appropriate venue based on your data type and/or community standard. If no appropriate specific database exists, please deposit your data to an appropriate publicly available database.

B. FINAL FILES:

Please contact the journal office with any questions, cellbio@rockefeller.edu.

Thank you for this interesting contribution, we look forward to publishing your paper in Journal of Cell Biology.

Sincerely,

Ana Pombo
Monitoring Editor
Journal of Cell Biology

Lucia Morgado-Palacin, PhD
Scientific Editor
Journal of Cell Biology